# How does unlabeled data improve generalization in self-training? A one-hidden-layer theoretical analysis

**Shuai Zhang**
Rensselaer Polytechnic Institute
Troy, NY, USA 12180
`zhangs29@rpi.edu`

**Meng Wang**
Rensselaer Polytechnic Institute
Troy, NY, USA 12180
`wangm7@rpi.edu`

**Sijia Liu**
Michigan State University
East Lansing, MI, USA 48824
MIT-IBM Watson AI Lab, IBM Research
`liusiji5@msu.edu`

**Pin-Yu Chen**
IBM Research
Yorktown Heights, NY, USA 10562
`Pin-Yu.Chen@ibm.com`

**Jinjun Xiong**
University at Buffalo
Buffalo NY, USA 14260
`jinjun@buffalo.edu`

## Abstract

Self-training, a semi-supervised learning algorithm, leverages a large amount of unlabeled data to improve learning when the labeled data are limited. Despite empirical successes, its theoretical characterization remains elusive. To the best of our knowledge, this work establishes the first theoretical analysis for the known iterative self-training paradigm and proves the benefits of unlabeled data in both training convergence and generalization ability. To make our theoretical analysis feasible, we focus on the case of one-hidden-layer neural networks. However, theoretical understanding of iterative self-training is non-trivial even for a shallow neural network. One of the key challenges is that existing neural network landscape analysis built upon supervised learning no longer holds in the (semi-supervised) self-training paradigm. We address this challenge and prove that iterative self-training converges linearly with both convergence rate and generalization accuracy improved in the order of $1/\sqrt{M}$, where $M$ is the number of unlabeled samples. Experiments from shallow neural networks to deep neural networks are also provided to justify the correctness of our established theoretical insights on self-training.

## 1 Introduction

Self-training (Scudder, 1965; Yarowsky, 1995; Lee et al., 2013; Han et al., 2019), one of the most powerful semi-supervised learning (SemiSL) algorithms, augments a limited number of labeled data with unlabeled data so as to achieve improved generalization performance on test data, compared with the model trained by supervised learning using the labeled data only. Self-training has shown empirical success in diversified applications such as few-shot image classification (Su et al., 2020; Xie et al., 2020; Chen et al., 2020a; Yalniz et al., 2019; Zoph et al., 2020), objective detection (Rosenberg et al., 2005), robustness-aware model training against adversarial attacks (Carmon et al., 2019), continual lifelong learning (Lee et al., 2019), and natural language processing (He et al., 2019; Kahn et al., 2020). The terminology "self-training" has been used to describe various SemiSL

algorithms in the literature, while this paper is centered on the commonly-used iterative self-training method in particular. In this setup, an initial teacher model (learned from the labeled data) is applied to the unlabeled data to generate pseudo labels. One then trains a student model by minimizing the weighted empirical risk of both the labeled and unlabeled data. The student model is then used as the new teacher to update the pseudo labels of the unlabeled data. This process is repeated multiple times to improve the eventual student model. We refer readers to Section 2 for algorithmic details.

Despite the empirical achievement of self-training methods with neural networks, the theoretical justification of such success is very limited, even in the field of SemiSL. The majority of the theoretical results on general SemiSL are limited to linear networks (Chen et al., 2020b; Raghunathan et al., 2020; Oymak & Gulcu, 2020; Oneto et al., 2011). The authors in (Balcan & Blum, 2010) show that unlabeled data can improve the generalization bound if the unlabeled data distribution and target model are compatible. For instance, the unlabeled data need to be well-chosen such that the target function for labeled data can separate the unlabeled data clusters, which, however, may not be able to be verified ahead. Moreover, (Rigollet, 2007; Singh et al., 2008) proves that unlabeled data can improve the convergence rate and generalization error under a similar clustering assumption, where the data contains clusters that have homogeneous labels. A recent work by Wei et al. (2020) analyzes SemiSL on nonlinear neural networks and proves that an infinite number of unlabeled data can improve the generalization compared with training with labeled data only. However, Wei et al. (2020) considers single shot rather than iterative SemiSL, and the training problem aims to minimize the consistency regularization rather than the risk function in the conventional self-training method (Lee et al., 2013). Moreover, Wei et al. (2020) directly analyzes the global optimum of the nonconvex training problem without any discussion about how to achieve the global optimum. To the best of our knowledge, there exists no analytical characterization of how the unlabeled data affect the generalization of the learned model by iterative self-training on nonlinear neural networks.

**Contributions.** This paper provides the first theoretical study of iterative self-training on nonlinear neural networks. Focusing on one-hidden-layer neural networks, this paper provides a quantitative analysis of the generalization performance of iterative self-training as a function of the number of labeled and unlabeled samples. Specifically, our contributions include

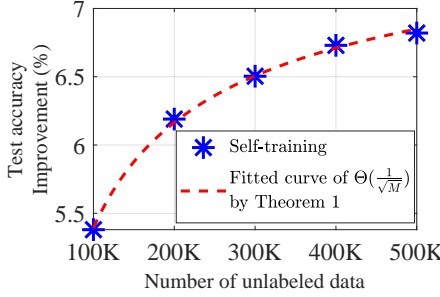

**1. Quantitative justification of generalization improvement by unlabeled data.** Assuming the existence of a ground-truth model with weights $\boldsymbol{W}^*$ that maps the features to the corresponding labels, we prove that the learned model via iterative self-training moves closer to $\boldsymbol{W}^*$ as the number $M$ of unlabeled data increases, indicating a better testing performance. Specifically, we prove that the Frobenius distance to $\boldsymbol{W}^*$, which is approximately linear in the generalization error, decreases in the order of $1/\sqrt{M}$. As an example, Figure 1 shows that the proposed theoretical bound

Figure 1: The trend of test accuracy improvement (%) on CIFAR-10 by self-training on CIFAR-10 (labeled) with different amount of unlabeled data from 80 Million Tiny Images matches our theoretical prediction.

matches the empirical self-training performance versus the number of unlabeled data for image classification; see details in Section 4.2.

**2. Analytical justification of iterative self-training over single shot alternative.** We prove that the student models returned by the iterative self-training method *converges linearly* to a model close to $\boldsymbol{W}^*$, with the rate improvement in the order of $1/\sqrt{M}$.

**3. Sample complexity analysis of labeled and unlabeled data for learning a proper model.** We quantify the impact of labeled and unlabeled data on the generalization of the learned model. In particular, we prove that the sample complexity of labeled data can be reduced compared with supervised learning.

### 1.1 RELATED WORKS

**Semi-supervised learning.** Besides self-training, many recent SemiSL algorithms exploit either consistency regularization or entropy minimization. Consistency regularization is based on the as-

sumption that the learned model will return same or similar output when the input is perturbed (Laine & Aila, 2016; Bachman et al., 2014; Sajjadi et al., 2016; Tarvainen & Valpola, 2017; Reed et al., 2015). (Grandvalet & Bengio, 2005) claims that the unlabeled data are more informative if the pseudo labels of the unlabeled data have lower entropy. Therefore, a line of works (Grandvalet & Bengio, 2005; Miyato et al., 2018) adds a regularization term that minimizes the entropy of the outputs of the unlabeled data. In addition, hybrid algorithms that unify both the above regularizations have been developed like (Berthelot et al., 2019a;b; Sohn et al., 2020).

**Domain adaptation.** Domain adaptation exploits abundant data in the source domain to learn a model for the target domain, where only limited training data are available (Liebelt & Schmid, 2010; Vazquez et al., 2013; Zhang et al., 2013; Long et al., 2015; Tzeng et al., 2014). Source and target domain are related but different. Unsupervised domain adaptation (Ganin & Lempitsky, 2015; Ganin et al., 2016; Gong et al., 2013; Bousmalis et al., 2016), where training data in target domain are unlabeled, is similar to SemiSL, and self-training methods have been used for analysis (Zou et al., 2018; Tang et al., 2012; French et al., 2018). However, self-training and unsupervised domain adaptation are fundamentally different. The former learns a model for the domain where there is limited labeled data, with the help of a large number of *unlabeled* data from a different domain. The latter learns a model for the domain where the training data are unlabeled, with the help of sufficient *labeled* data from a different domain.

**Generalization analysis of supervised learning.** In theory, the testing error is upper bounded by the training error plus the generalization gap between training and testing. These two quantities are often analyzed separately and cannot be proved to be small simultaneously for deep neural networks. For example, neural tangent kernel (NTK) method (Jacot et al., 2018; Du et al., 2018; Lee et al., 2018) shows the training error can be zero, and the Rademacher complexity in (Bartlett & Mendelson, 2002) bounds the generalization gap (Arora et al., 2019a). For one-hidden-layer neural networks (Safran & Shamir, 2018), the testing error can be proved to be zero under mild conditions. One common assumption is that the input data belongs to the Gaussian distribution (Zhong et al., 2017; Ge et al., 2018; Kalai et al., 2008; Bakshi et al., 2019; Zhang et al., 2016; Brutzkus & Globerson, 2017; Li & Yuan, 2017; Soltanolkotabi et al., 2018). Another line of approaches (Brutzkus et al., 2018; Li & Liang, 2018; Wang et al., 2019) consider linearly separable data.

The rest of this paper is organized as follows. Section 2 introduces the problem formulation and self-training algorithm. Major results are summarized in Section 3, and empirical evaluations are presented in Section 4. Section 5 concludes the whole paper. All the proofs are in the Appendix.

## 2 FORMALIZING SELF-TRAINING: NOTATION, FORMULATION, AND ALGORITHM

**Problem formulation.** Given $N$ labeled data sampled from distribution $P_l$, denoted by $\mathcal{D} = \{\boldsymbol{x}_n, y_n\}_{n=1}^N$, and $M$ unlabeled data drawn from distribution $P_u$, denoted by $\widetilde{\mathcal{D}} = \{\widetilde{\boldsymbol{x}}_m\}_{m=1}^M$. The aim is to find a neural network model $g(\boldsymbol{W})$, where $\boldsymbol{W}$ denotes the trainable weights, that minimizes the testing error on data sampled from $P_l$.

---

**Table 1: Iterative Self-Training**

(S1) Initialize iteration $\ell = 0$ and obtain a model $\boldsymbol{W}^{(\ell)}$ as the teacher using labeled data $\mathcal{D}$ only;

(S2) Use the teacher model to obtain pseudo labels $\widetilde{y}_m$ of unlabeled data in $\widetilde{\mathcal{D}}$;

(S3) Train the neural network by minimizing (1) via $T$-step mini-batch gradient descent method using disjoint subsets $\{\mathcal{D}_t\}_{t=0}^{T-1}$ and $\{\widetilde{\mathcal{D}}_t\}_{t=0}^{T-1}$ of $\widetilde{\mathcal{D}}$. Let $\boldsymbol{W}^{(\ell+1)}$ denote the obtained student model;

(S4) Use $\boldsymbol{W}^{(\ell+1)}$ as the current teacher model. Let $\ell \leftarrow \ell + 1$ and go back to step (S2);

---

**Iterative self-training.** In each iteration, given the current teacher predictor $g(\boldsymbol{W}^{(\ell)})$, the pseudo-labels for the unlabeled data in $\widetilde{\mathcal{D}}$ are computed as $\tilde{y}_m = g(\boldsymbol{W}^{(\ell)}; \widetilde{\boldsymbol{x}}_m)$. The method then minimizes the weighted empirical risk $\hat{f}_{\mathcal{D}, \widetilde{\mathcal{D}}}(\boldsymbol{W})$ of both labeled and unlabeled data through stochastic gradient

descent, where

$$\hat{f}_{\mathcal{D},\widetilde{\mathcal{D}}}(\boldsymbol{W}) = \frac{\lambda}{2N} \sum_{n=1}^{N} \big(y_n - g(\boldsymbol{W}; \boldsymbol{x}_n)\big)^2 + \frac{\widetilde{\lambda}}{2M} \sum_{m=1}^{M} \big(\widetilde{y}_m - g(\boldsymbol{W}; \widetilde{\boldsymbol{x}}_m)\big)^2, \tag{1}$$

and $\lambda + \widetilde{\lambda} = 1$. The learned student model $g(\boldsymbol{W}^{(\ell+1)})$ is used as the teacher model in the next iteration. The initial model $g(\boldsymbol{W}^{(0)})$ is learned from labeled data. The formal algorithm is summarized as in Table 1.

**Model and assumptions.** This paper considers regression[1], where $g$ is a one-hidden-layer fully connected neural network equipped with $K$ neurons. Namely, given the input $\boldsymbol{x} \in \mathbb{R}^d$ and weights $\boldsymbol{W} = [\boldsymbol{w}_1, \boldsymbol{w}_2, \cdots, \boldsymbol{w}_K] \in \mathbb{R}^{d \times K}$, we have

$$g(\boldsymbol{W}; \boldsymbol{x}) := \frac{1}{K} \sum_{j=1}^{K} \phi(\boldsymbol{w}_j^T \boldsymbol{x}), \tag{2}$$

where $\phi$ is the ReLU activation function[2], and $\phi(z) = \max\{z, 0\}$ for any input $z \in \mathbb{R}$. Here, we fix the top layer weights as 1 for simplicity, and the equivalence of such a simplification is discussed in Appendix K.

Moreover, we assume an unknown ground-truth model with weights $\boldsymbol{W}^*$ that maps all the features to the corresponding labels drawn from $P_l$, i.e., $y = g(\boldsymbol{W}^*; \boldsymbol{x})$, where $(\boldsymbol{x}, y) \sim P_l$. The generalization function (GF) with respect to $g(\boldsymbol{W})$ is defined as

$$I\big(g(\boldsymbol{W})\big) = \mathbb{E}_{(\boldsymbol{x},y)\sim P_l} \big(y - g(\boldsymbol{W}; \boldsymbol{x})\big)^2 = \mathbb{E}_{(\boldsymbol{x},y)\sim P_l} \big(g(\boldsymbol{W}^*; \boldsymbol{x}) - g(\boldsymbol{W}; \boldsymbol{x})\big)^2. \tag{3}$$

By definition $I\big(g(\boldsymbol{W}^*)\big)$ is zero. Clearly, $\boldsymbol{W}^*$ is not unique because any column permutation of $\boldsymbol{W}^*$, which corresponds to permuting neurons, represents the same function as $\boldsymbol{W}^*$ and minimizes GF in (3) too. To simplify the representation, we follow the convention and abuse the notation that the distance from $\boldsymbol{W}$ to $\boldsymbol{W}^*$, denoted by $\|\boldsymbol{W} - \boldsymbol{W}^*\|_F$, means the smallest distance from $\boldsymbol{W}$ to any permutation of $\boldsymbol{W}^*$. Additionally, some important notations are summarized in Table 2.

We assume the inputs of both the labeled and unlabeled data belong to the zero mean Gaussian distribution, i.e., $\boldsymbol{x} \sim \mathcal{N}(0, \delta^2 \boldsymbol{I}_d)$, and $\widetilde{\boldsymbol{x}} \sim \mathcal{N}(0, \widetilde{\delta}^2 \boldsymbol{I}_d)$. The Gaussian assumption is motivated by the data whitening (LeCun et al., 2012) and batch normalization techniques (Ioffe & Szegedy, 2015) that are commonly used in practice to improve learning performance. Moreover, training one-hidden-layer neural network with multiple neurons is NP-Complete (Blum & Rivest, 1992) without any assumption.

**The focus of this paper.** This paper will analyze three aspects about self-training: (1) the generalization performance of $\boldsymbol{W}^{(L)}$, the returned model by self-training after $L$ iterations, measured by $\|\boldsymbol{W}^{(L)} - \boldsymbol{W}^*\|_F$[3]; (2) the influence of parameter $\lambda$ in (1) on the training performance; and (3) the impact of unlabeled data on the training and generalization performance.

Table 2: Some Important Notations

| | |
|---|---|
| $\mathcal{D} = \{\boldsymbol{x}_n, \boldsymbol{y}_n\}_{n=1}^{N}$ | Labeled dataset with $N$ number of samples; |
| $\widetilde{\mathcal{D}} = \{\widetilde{\boldsymbol{x}}_m\}_{m=1}^{M}$ | Unlabeled dataset with $M$ number of samples; |
| $d$ | Dimension of the input $\boldsymbol{x}$ or $\widetilde{\boldsymbol{x}}$; |
| $K$ | Number of neurons in the hidden layer; |
| $\kappa$ | Conditional number (the ratio of the largest and smallest singular values) of $\boldsymbol{W}^*$; |
| $\boldsymbol{W}^{(\ell)}$ | Model returned by self-training after $\ell$ iterations; $\boldsymbol{W}^{(0)}$ is the initial model; |
| $\boldsymbol{W}^*$ | Weights of the ground truth model; |
| $\boldsymbol{W}^{[\hat{\lambda}]}$ | $\boldsymbol{W}^{[\hat{\lambda}]} = \hat{\lambda} \boldsymbol{W}^* + (1 - \hat{\lambda}) \boldsymbol{W}^{(0)}$; |

---

[1]The results can be extended to binary classification with a cross-entropy loss function. Please see Appendix-I.

[2]Because ReLU is non-linear and non-smooth, (1) is non-convex and non-smooth, which poses analytical challenges. The results can be easily extended to smooth functions with bounded gradients, e.g., Sigmoid.

[3]We use this metric because $I\big(g(\boldsymbol{W})\big)$ is shown to be linear in $\|\boldsymbol{W}^{(L)} - \boldsymbol{W}^*\|_F$ numerically when $\boldsymbol{W}^{(L)}$ is close to $\boldsymbol{W}^*$, see Figure 4.

## 3 THEORETICAL RESULTS

**Beyond supervised learning: Challenge of self-training.** The existing theoretical works such as (Zhong et al., 2017; Zhang et al., 2020a;b;c) verify that for one-hidden-layer neural networks, if only labeled data are available, and $x$ are drawn from the standard Gaussian distribution, then supervised learning by minimizing (1) with $\lambda = 1$ can return a model with ground-truth weights $W^*$ (up to column permutation), as long as the number of labeled data $N$ is at least $N^*$, which depends on $\kappa, K$ and $d$. In contrast, this paper focuses on the **low labeled-data regime** when $N$ is less than $N^*$. Specifically,

$$N^*/4 < N \le N^*. \tag{4}$$

Intuitively, if $N < N^*$, the landscape of the empirical risk of the labeled data becomes highly non-convex, even in a neighborhood of $W^*$, thus, the existing analyses for supervised learning do not hold in this region. With additional unlabeled data, the landscape of the weighted empirical risk becomes smoother near $W^*$. Moreover, as $M$ increases, and starting from a nearby initialization, the returned model $W^{(L)}$ by iterative self-training can converge to a local minimum that is closer to $W^*$ (see illustration in Figure 2).

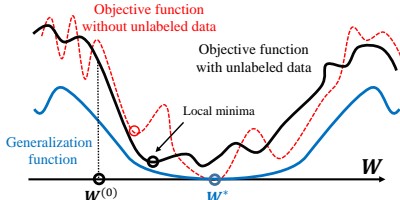

Figure 2: Adding unlabeled data in the empirical risk function drives its local minimum closer to $W^*$, which minimizes the generalization function.

Compared with supervised learning, the formal analyses of self-training need to handle new technical challenges from two aspects. First, the existing analyses of supervised learning exploit the fact that the GF and the empirical risk have the same minimizer, i.e., $W^*$. This property does not hold for self-training as $W^*$ no longer minimizes the weighted empirical risk in (1). Second, the iterative manner of self-training complicates the analyses. Specifically, the empirical risk in each iteration is different and depends on the model trained in the previous iteration through the pseudo labels.

In what follows, we provide theoretical insights and the formal theorems. Some important quantities $\hat{\lambda}$ and $\mu$ are defined below

$$\hat{\lambda} := \frac{\lambda\delta^2}{\lambda\delta^2 + \widetilde{\lambda}\widetilde{\delta}^2}, \quad \text{and} \quad \mu = \mu(\delta, \tilde{\delta}) := \sqrt{\frac{\lambda\delta^2 + \widetilde{\lambda}\widetilde{\delta}^2}{\lambda\rho(\delta) + \widetilde{\lambda}\rho(\tilde{\delta})}}, \tag{5}$$

where $\rho$ is a positive function defined in (73). $\hat{\lambda}$ is an increasing function of $\lambda$. Also, from Lemma 11 (in Appendix), $\rho(\delta)$ is in the order of $\delta^2$ when $\delta \le 1$ for ReLU activation functions. Thus, $\mu$ is a fixed constant, denoted by $\mu^*$, for all $\delta, \tilde{\delta} \le 1$. When $\delta$ and $\tilde{\delta}$ are large, $\mu$ increases as they increase. The formal definition of $N^*$ in (4) is $c(\kappa)\mu^{*2}K^3d \log q$, where $c(\kappa)$ is some polynomial function of $\kappa$ and can be viewed as constant.

### 3.1 INFORMAL KEY THEORETICAL FINDINGS

To the best of our knowledge, Theorems 1 and 2 provide the first theoretical characterization of iterative self-training on nonlinear neural networks. Before formally presenting them, we summarize the highlights as follows.

**1. Linear convergence of the learned models.** The learned models converge linearly to a model close to $W^*$. Thus, the iterative approach returns a model with better generalization than that by the single-shot method. Moreover, the convergence rate is a constant term plus a term in the order of $1/\sqrt{M}$ (see $\Delta_1$ in Figure 3), indicating a faster convergence with more unlabeled data.

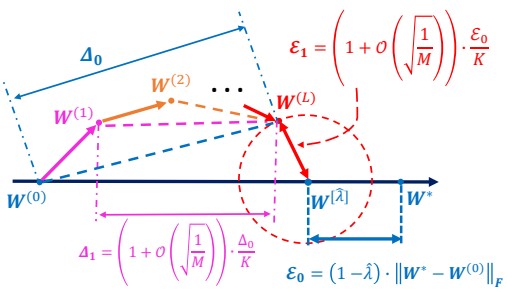

Figure 3: Illustration of the (1) ground truth $W^*$, (2) iterations $\{W^{(\ell)}\}_{\ell=0}^L$, (3) convergent point $W^{(L)}$, and (4) $W^{[\hat{\lambda}]} = \hat{\lambda}W^* + (1 - \hat{\lambda})W^{(0)}$.

**2. Returning a model with guaranteed generalization in the low labeled-data regime.** Even when the number of labeled data is much less than the required sample complexity to obtain $\boldsymbol{W}^*$ in supervised learning, we prove that with the help of unlabeled data, the iterative self-training can return a model in the neighborhood of $\boldsymbol{W}^{[\hat{\lambda}]}$, where $\boldsymbol{W}^{[\hat{\lambda}]}$ is in the line segment of $\boldsymbol{W}^{(0)}$ ($\hat{\lambda} = 0$) and ground truth $\boldsymbol{W}^*$ ($\hat{\lambda} = 1$). Moreover, $\hat{\lambda}$ is upper bounded by $\sqrt{N/N^*}$. Thus $\boldsymbol{W}^{(L)}$ moves closer to $\boldsymbol{W}^*$ as $N$ increases ($\mathcal{E}_0$ in Figure 3), indicating a better generalization performance with more labeled data.

**3. Guaranteed generalization improvement by unlabeled data.** The distance between $\boldsymbol{W}^{(L)}$ and $\boldsymbol{W}^{[\hat{\lambda}]}$ ($\mathcal{E}_1$ in Figure 3) scales in the order of $1/\sqrt{M}$. With a larger number of unlabeled data $M$, $\boldsymbol{W}^{(L)}$ moves closer to $\boldsymbol{W}^{[\hat{\lambda}]}$ and thus $\boldsymbol{W}^*$, indicating an improved generalization performance (Theorem 1). When $N$ is close to $N^*$ but still smaller as defined in (12), both $\boldsymbol{W}^{(L)}$ and $\boldsymbol{W}^{[\hat{\lambda}]}$ converge to $\boldsymbol{W}^*$, and thus the learned model achieves zero generalization error (Theorem 2).

### 3.2 FORMAL THEORY IN LOW LABELED-DATA REGIME

*Takeaways of Theorem 1:* Theorem 1 characterizes the convergence rate of the proposed algorithm and the accuracy of the learned model $\boldsymbol{W}^{(L)}$ in a low labeled-data regime. Specifically, the iterates converge linearly, and the learned model is close to $\boldsymbol{W}^{[\hat{\lambda}]}$ and guaranteed to outperform the initial model $\boldsymbol{W}^{(0)}$.

**Theorem 1.** *Suppose the initialization $\boldsymbol{W}^{(0)}$ and the number of labeled data satisfy*

$$\|\boldsymbol{W}^{(0)} - \boldsymbol{W}^*\|_F \leq p^{-1} \cdot \frac{\|\boldsymbol{W}^*\|_F}{c(\kappa)\mu^2 K^{3/2}} \quad with \quad p \in \left(\frac{1}{2}, 1\right], \tag{6}$$

$$and \quad \max\left\{\frac{1}{K}, p - \frac{2p-1}{\mu\sqrt{K}}\right\}^2 \cdot N^* \leq N \leq N^*. \tag{7}$$

*If the value of $\hat{\lambda}$ in (5) and unlabeled data amount $M$ satisfy*

$$\max\left\{\frac{1}{K}, p - \frac{2p-1}{\mu\sqrt{K}}\right\} \leq \hat{\lambda} \leq \min\left\{\sqrt{\frac{N}{N^*}}, p + \frac{2p-1}{\mu\sqrt{K}}\right\}, \tag{8}$$

$$and \quad M \geq (2p-1)^{-2}c(\kappa)\mu^2\left(1 - \hat{\lambda}\right)^2 K^3 d\log q. \tag{9}$$

*Then, when the number $T$ of SGD iterations is large enough in each loop $\ell$, with probability at least $1 - q^{-d}$, the iterates $\{\boldsymbol{W}^{(\ell)}\}_{\ell=0}^L$ converge to $\boldsymbol{W}^{[\hat{\lambda}]}$ as*

$$\|\boldsymbol{W}^{(L)} - \boldsymbol{W}^{[\hat{\lambda}]}\|_F \leq \left(\left(1 + \Theta\left(\frac{\mu(1-\hat{\lambda})}{\sqrt{M}}\right)\right) \cdot \hat{\lambda}\right)^L \cdot \|\boldsymbol{W}^{(0)} - \boldsymbol{W}^{[\hat{\lambda}]}\|_2 + \left(1 + \Theta\left(\frac{\mu(1-\hat{\lambda})}{\sqrt{M}}\right)\right) \cdot \|\boldsymbol{W}^* - \boldsymbol{W}^{[\hat{\lambda}]}\|_F, \tag{10}$$

*where $\boldsymbol{W}^{[\hat{\lambda}]} = \hat{\lambda}\boldsymbol{W}^* + (1 - \hat{\lambda})\boldsymbol{W}^{(0)}$. Typically, when the iteration number $L$ is sufficient large, we have*

$$\|\boldsymbol{W}^{(L)} - \boldsymbol{W}^*\|_F \leq \left(1 + \Theta\left(\frac{\mu(1-\hat{\lambda})}{\sqrt{M}}\right)\right) \cdot 2(1 - \hat{\lambda}) \cdot \|\boldsymbol{W}^* - \boldsymbol{W}^{(0)}\|_F. \tag{11}$$

The accuracy of the learned model $\boldsymbol{W}^{(L)}$ with respect to $\boldsymbol{W}^*$ is characterized as (10), and the learning model is better than initial model as in (11) if the following conditions hold. First, the weights $\lambda$ in (1) are properly chosen as in (8). Second, the number of unlabeled data is sufficiently large as in (9).

**Selection of $\lambda$ in self-training algorithms.** When $\hat{\lambda}$ increases, the required number of unlabeled data is reduced from (9), and the convergence point $\boldsymbol{W}^{(L)}$ becomes closer to $\boldsymbol{W}^*$ from (11), which indicates a smaller generalization error. Thus, a large $\hat{\lambda}$ within its feasible range (8) is desirable. When the initial model $\boldsymbol{W}^{(0)}$ is closer to $\boldsymbol{W}^*$ (corresponding to a larger $p$), and the number of labeled data $N$ increases, the upper bound in (8) increases, and thus, one can select a larger $\hat{\lambda}$.

**The initial model $\boldsymbol{W}^{(0)}$.** The tensor initialization from (Zhong et al., 2017) can return a $\boldsymbol{W}^{(0)}$ that satisfies (6) when the number of labeled data is $N = p^2 N^*$ (see Lemma 3 in Appendix). Combining with the requirement in (7), Theorem 1 applies to the case that $N$ is at least $N^*/4$.

### 3.3 Formal theory of achieving zero generalization error

*Takeaways of Theorem 2:* Theorem 2 indicates the model returned by the proposed algorithm converges linearly to the ground truth $\boldsymbol{W}^*$. Thus the distance between the learned model and the ground truth can be arbitrarily small with the ability to achieve zero generalization error. The required sample complexity is reduced by a constant factor compared with supervised learning.

**Theorem 2.** *Consider the number of unlabeled data satisfies*

$$\left(1 - 1/(\mu\sqrt{K})\right)^2 \cdot N^* \leq N \leq N^*, \tag{12}$$

*we choose $\hat{\lambda}$ such that*

$$1 - 1/(\mu\sqrt{K}) \leq \hat{\lambda} \leq \sqrt{N/N^*}. \tag{13}$$

*Suppose the initial model $\boldsymbol{W}^{(0)}$ and the number of unlabeled data $M$ satisfy*

$$\|\boldsymbol{W}^{(0)} - \boldsymbol{W}^*\|_F \leq \frac{\|\boldsymbol{W}^*\|_F}{c(\kappa)\mu^2 K^{3/2}} \quad and \quad M \geq c(\kappa)\mu^2(1-\hat{\lambda})^2 K^3 d \log q, \tag{14}$$

*the iterates $\{\boldsymbol{W}^{(\ell)}\}_{\ell=0}^L$ converge to the ground truth $\boldsymbol{W}^*$ as follows,*

$$\|\boldsymbol{W}^{(L)} - \boldsymbol{W}^*\|_F \leq \left[\left(1 + \frac{c(\kappa)\hat{\lambda}}{\sqrt{N}} + \frac{c(\kappa)(1-\hat{\lambda})}{\sqrt{M}}\right) \cdot \mu\sqrt{K}(1-\hat{\lambda})\right]^L \cdot \|\boldsymbol{W}^{(0)} - \boldsymbol{W}^*\|_F. \tag{15}$$

The models $\boldsymbol{W}^{(\ell)}$'s converge linearly to the ground truth $\boldsymbol{W}^*$ as (15) when the number of labeled data satisfies (12). In contrast, supervised learning requires at least $N^*$ labeled samples to estimate $\boldsymbol{W}^*$ accurately without unlabeled data, which suggests self-training at least saves a constant fraction of labeled data.

### 3.4 The main proof idea

Our proof builds upon and extends one recent line of works on supervised learning such as (Zhong et al., 2017; Zhang et al., 2020b;c; 2021). The standard framework of these works is first to show that the generalization function $I(g(\boldsymbol{W}))$ in (3) is locally convex near $\boldsymbol{W}^*$, which is its global minimizer. Then, when $M = 0$ and $N$ is sufficiently large, the empirical risk function using labeled data only can approximate $I(g(\boldsymbol{W}))$ well in the neighborhood of $\boldsymbol{W}^*$. Thus, if initialized in this local convex region, the iterations, returned by applying gradient descent approach on the empirical risk function, converge to $\boldsymbol{W}^*$ linearly.

The technical challenge here is that in self-training, when unlabeled data are paired with pseudo labels, $\boldsymbol{W}^*$ is no longer a global minimizer of the empirical risk $\hat{f}_{\mathcal{D},\widetilde{\mathcal{D}}}$ in (1), and $\hat{f}_{\mathcal{D},\widetilde{\mathcal{D}}}$ does not approach $I(g(\boldsymbol{W}))$ even when $M$ and $N$ increase to infinity. Our new idea is to design a population risk function $f(\boldsymbol{W};\hat{\lambda})$ in (17) (see appendix), which is a lower bound of $\hat{f}_{\mathcal{D},\widetilde{\mathcal{D}}}$ when $M$ and $N$ are infinity. $f(\boldsymbol{W};\hat{\lambda})$ is locally convex around its minimizer $\boldsymbol{W}^{[\hat{\lambda}]}$, and $\boldsymbol{W}^{[\hat{\lambda}]}$ approaches $\boldsymbol{W}^*$ as $\hat{\lambda}$ increases. Then we show the iterates generated by $\hat{f}_{\mathcal{D},\widetilde{\mathcal{D}}}$ stay close to $f(\boldsymbol{W};\hat{\lambda})$, and the returned model $\boldsymbol{W}^{(L)}$ is close to $\boldsymbol{W}^{[\hat{\lambda}]}$. New technical tools are developed to bound the distance between the functions $\hat{f}_{\mathcal{D},\widetilde{\mathcal{D}}}$ and $f(\boldsymbol{W};\hat{\lambda})$.

## 4 Empirical results

### 4.1 Synthetic data experiments

We generate a ground-truth neural network with the width $K = 10$. Each entry of $\boldsymbol{W}^*$ is uniformly selected from $[-2.5, 2.5]$. The input of labeled data $\boldsymbol{x}_n$ are generated from Gaussian distribution $\mathcal{N}(0, \boldsymbol{I}_d)$ independently, and the corresponding label $y_n$ is generated through (2) using $\boldsymbol{W}^*$. The unlabeled data $\widetilde{\boldsymbol{x}}_m$ are generated from $\mathcal{N}(0, \widetilde{\delta}^2 \boldsymbol{I}_d)$ independently with $\widetilde{\delta} = 1$ except in Figure 7. $d$ is set as 50 except in Figure 9. The value of $\lambda$ is selected as $\sqrt{N/(2Kd)}$ except in Figure 8. We consider one-hidden-layer except in Figure 4. The initial teacher model $\boldsymbol{W}^{(0)}$ in self-training is randomly selected from $\{\boldsymbol{W} | \|\boldsymbol{W} - \boldsymbol{W}^*\|_F / \|\boldsymbol{W}^*\|_F \leq 0.5\}$ to reduce the computation. In

each iteration, the maximum number of SGD steps $T$ is 10. Self-training terminates if $\|\boldsymbol{W}^{(\ell+1)} - \boldsymbol{W}^{(\ell)}\|_F / \|\boldsymbol{W}^{(\ell)}\|_F \leq 10^{-4}$ or reaching 1000 iterations. In Figures 5 to 8, all the points on the curves are averaged over 1000 independent trials, and the regions in lower transparency indicate the corresponding one-standard-deviation error bars. Our **empirical observations** are summarized below.

**(a) GF (testing performance) proportional to $\|\boldsymbol{W} - \boldsymbol{W}^*\|_F$.** Figure 4 illustrates the GF in (3) against the distance to the ground truth $\boldsymbol{W}^*$. To visualize results for different networks together, GF is normalized in $[0, 1]$, divided by its largest value for each network architecture. All the results are averaged over 100 independent choice of $\boldsymbol{W}$. One can see that for one-hidden-layer neural networks, in a large region near $\boldsymbol{W}^*$, GF is almost linear in $\|\boldsymbol{W} - \boldsymbol{W}^*\|_F$. When the number of hidden layers increases, this region decreases, but the linear dependence still holds locally. This is an empirical justification of using $\|\boldsymbol{W} - \boldsymbol{W}^*\|_F$ to evaluate the GF and, thus, the testing error in Theorems 1 and 2.

**(b) $\|\boldsymbol{W}^{(L)} - \boldsymbol{W}^*\|_F$ as a linear function of $1/\sqrt{M}$.** Figure 5 shows the relative error $\|\boldsymbol{W}^{(L)} - \boldsymbol{W}^*\|_F / \|\boldsymbol{W}^*\|_F$ when the number of unlabeled data and labeled data changes. One can see that the relative error decreases when either $M$ or $N$ increases. Additionally, the dash-dotted lines represent the best fitting of the linear functions of $1/\sqrt{M}$ using the least square method. Therefore, the relative error is indeed a linear function of $1/\sqrt{M}$, as predicted by our results in (11) and (15).

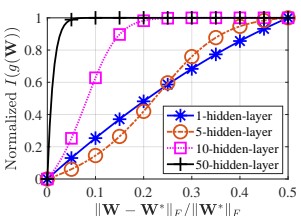

Figure 4: The generalization function against the distance to the ground truth neural network

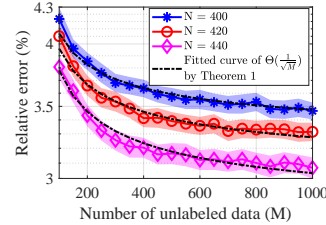

Figure 5: The relative error against the number of unlabeled data.

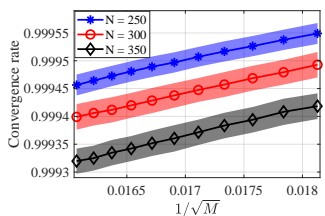

Figure 6: The convergence rate with different $M$ when $N < N^*$.

**(c) Convergence rate as a linear function of $1/\sqrt{M}$.** Figure 6 illustrates the convergence rate when $M$ and $N$ change. We can see that the convergence rate is a linear function of $1/\sqrt{M}$, as predicted by our results (11) and (15). When $M$ increases, the convergence rate is improved, and the method converges faster.

**(d) Increase of $\widetilde{\delta}$ slows down convergence.** Figure 7 shows that the convergence rate becomes worse when the variance of the unlabeled data $\widetilde{\delta}$ increases from 1. When $\widetilde{\delta}$ is less than 1, the convergence rate almost remains the same, which is consistent with our characterization in (10) that the convergence rate is linear in $\mu$. From the discussion after (5), $\mu$ increases as $\widetilde{\delta}$ increases from 1 and stays constant when $\widetilde{\delta}$ is less than 1.

**(e) $\|\boldsymbol{W}^{(L)} - \boldsymbol{W}^*\|_F / \|\boldsymbol{W}^*\|_F$ is improved as a linear function of $\hat{\lambda}$.** Figure 8 shows that the relative errors of $\boldsymbol{W}^{(L)}$ with respect to $\boldsymbol{W}^*$ decrease almost linearly when $\hat{\lambda}$ increases, which is consistent with the theoretical result in (11). Moreover, when $\lambda$ exceeds a certain threshold positively correlated with $N$, the relative error increases rather than decreases. That is consistent with the analysis in (8) that $\hat{\lambda}$ has an upper limit, and such a limit increases as $N$ increases.

**(f) Unlabeled data reduce the sample complexity to learn $\boldsymbol{W}^*$.** Figure 9 depicts the phase transition of returning $\boldsymbol{W}^{(L)}$. For every pair of $d$ and $N$, we construct 100 independent trials, and each trial is said to be successful if $\|\boldsymbol{W}^{(L)} - \boldsymbol{W}^*\|_F / \|\boldsymbol{W}^*\|_F \leq 10^{-2}$. The white blocks correspond to the successful trials, while the block in black indicates all failures. When $d$ increases, the required number of labeled data to learn $\boldsymbol{W}^*$ is linear in $d$. Thus, the sample complexity bound in (12) is order-wise optimal for $d$. Moreover, the phase transition line when $M = 1000$ is below the one when $M = 0$. Therefore, with unlabeled data, the required sample complexity of $N$ is reduced.

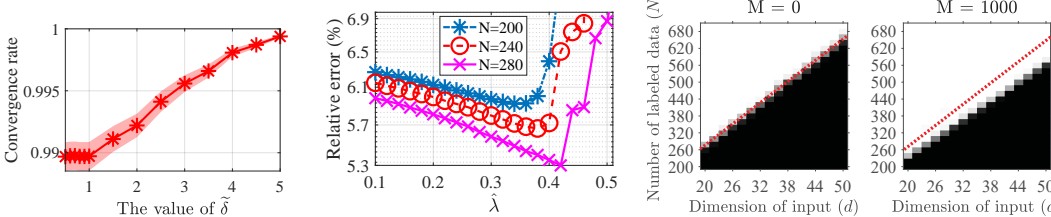

Figure 7: Convergence rate with different $\hat{\delta}$.

Figure 8: $\dfrac{\|\boldsymbol{W}^{(L)}-\boldsymbol{W}^*\|_F}{\|\boldsymbol{W}^*\|_F}$ when $\hat{\lambda}$ and $N$ change.

Figure 9: Empirical phase transition of the curves with (a) $M = 0$ and (b) $M = 1000$.

## 4.2 IMAGE CLASSIFICATION ON AUGMENTED CIFAR-10 DATASET

We evaluate self-training on the augmented CIFAR-10 dataset, which has 50K labeled data. The unlabeled data are mined from 80 Million Tiny Images following the setup in (Carmon et al., 2019)[4], and additional 50K images are selected for each class, which is a total of 500K images, to form the unlabeled data. The self-training method is the same implementation as that in (Carmon et al., 2019). $\lambda$ and $\widetilde{\lambda}$ is selected as $N/(M + N)$ and $M/(N + M)$, respectively, and the algorithm stops after 200 epochs. In Figure 10, the dash lines stand for the best fitting of the linear functions of $1/\sqrt{M}$ via the least square method. One can see that the test accuracy is improved by up to $7\%$ using unlabeled data, and the empirical evaluations match the theoretical predictions. Figure 11 shows the convergence rate calculated based on the first 50 epochs, and the convergence rate is almost a linear function of $1/\sqrt{M}$, as predicted by (10).

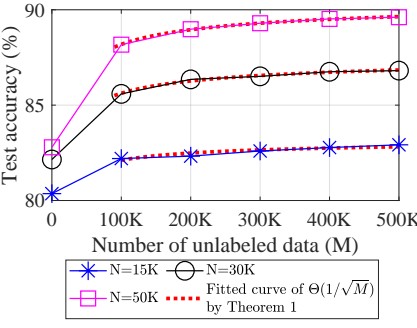

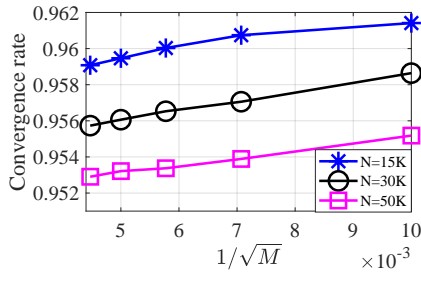

Figure 10: The test accuracy against the number of unlabeled data

Figure 11: The convergence rate against the number of unlabeled data

## 5 CONCLUSION

This paper provides new theoretical insights into understanding the influence of unlabeled data in the iterative self-training algorithm. We show that the improved generalization error and convergence rate is a linear function of $1/\sqrt{M}$, where $M$ is the number of unlabeled data. Moreover, compared with supervised learning, using unlabeled data reduces the required sample complexity of labeled data for achieving zero generalization error. Future directions include generalizing the analysis to multi-layer neural networks and other semi-supervised learning problems such as domain adaptation.

## ACKNOWLEDGEMENT

This work was supported by AFOSR FA9550-20-1-0122, ARO W911NF-21-1-0255, NSF 1932196 and the Rensselaer-IBM AI Research Collaboration (http://airc.rpi.edu), part of the IBM AI Horizons Network (http://ibm.biz/AIHorizons).

---

[4]The codes are downloaded from https://github.com/yaircarmon/semisup-adv

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

# Appendix

## A OVERVIEW OF THE PROOF TECHNIQUES

We first provide an overview of the techniques used in proving Theorems 1 and 2.

**1. Characterization of a proper population risk function.** To characterize the performance of the iterative self-training algorithm via the stochastic gradient descent method, we need first to define a population risk function such that the following two properties hold. First, the landscape of the population risk function should be analyzable near $\{\boldsymbol{W}^{(\ell)}\}_{\ell=0}^{L}$. Second, the distance between the empirical risk function in (1) and the population risk function should be bounded near $\{\boldsymbol{W}^{(\ell)}\}_{\ell=0}^{L}$. The generalization function defined in (3), which is widely used in the supervised learning problem with a sufficient number of samples, failed the second requirement. To this end, we turn to find a new population risk function defined in (17), and the illustrations of the population risk function and objection function are included in Figure 12.

**2. Local convex region of the population risk function.** The purpose is to characterize the iterations via the stochastic gradient descent method in the population risk function. To obtain the local convex region of the population risk function, we first bound the Hessian of the population risk function at its global optimal. Then, we utilize Lemma 12 in Appendix H.1 to obtain the Hessian of the population risk function near the global optimal. The local convex region of the population risk function is summarized in Lemma 1, and the proof of Lemma 1 is included in Appendix H.1.

**3. Bound between the population risk and empirical risk functions.** After the characterization of the iterations via the stochastic gradient descent method in the population risk function, we need to bound the distance between the population risk function and empirical risk function. Therefore, the behaviors of the iterations via the stochastic gradient descent method in the empirical risk function can be described by the ones in the population risk function and the distance between these two. The key lemma is summarized in Lemma 2 (see Appendix H.2), and the proof is included in Appendix H.2.

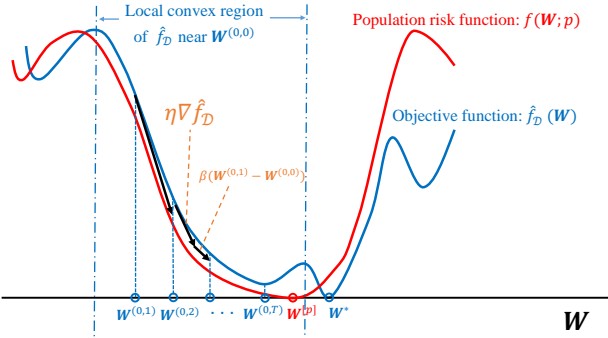

Figure 12: The landscapes of the objection function and population risk function.

In the following contexts, the details of the iterative self-training algorithm are included in Appendix B. We then first provide the proof of Theorem 2 in Appendix E, which can be viewed as a special case of Theorem 1. Then, with the preliminary knowledge from proving Theorem 2, we turn to present the full proof of a more general statement summarized in Theorem 3 (see Appendix F), which is related to Theorem 1. The definition and relative proofs of $\mu$ and $\rho$ are all included in Appendix G. The proofs of preliminary lemmas are included in Appendix H.

## B ITERATIVE SELF-TRAINING ALGORITHM

In this section, we implement the details of the mini-batch stochastic gradient descent used in each stage of the iterative self-training algorithm. After $t$ number of iterations via mini-batch stochastic

gradient descent at $\ell$-th stage of self-training algorithm, the learned model is denoted as $\boldsymbol{W}^{(\ell,t)}$. One can easily check that $\boldsymbol{W}^{(\ell)}$ in the main context is denoted as $\boldsymbol{W}^{(\ell,0)}$ in this section and the following proofs. Last, the pseudo-code of the iterative self-training algorithm is summarized in Algorithm 1.

---

**Algorithm 1** Iterative Self-Training Algorithm

---

**Input:** labeled $\mathcal{D} = \{(\boldsymbol{x}_n, y_n)\}_{n=1}^N$, unlabeled data $\widetilde{\mathcal{D}} = \{\widetilde{\boldsymbol{x}}_m\}_{m=1}^M$, and gradient step size $\eta$;

**Initialization:** preliminary teacher model with weights $\boldsymbol{W}^{(0,0)}$;

**Partition:** randomly and independently pick data from $\mathcal{D}$ and $\widetilde{\mathcal{D}}$ to form $T$ subsets $\{\mathcal{D}_t\}_{t=0}^{T-1}$ and $\{\widetilde{\mathcal{D}}_t\}_{t=0}^{T-1}$, respectively;

**for** $\ell = 0, 1, \cdots, L-1$ **do**

    $y_m = g(\boldsymbol{W}^{(\ell,0)}; \widetilde{\boldsymbol{x}}_m)$ for $m = 1, 2, \cdots, M$

    **for** $t = 0, 1, \cdots, T-1$ **do**

        $\boldsymbol{W}^{(\ell,t+1)} = \boldsymbol{W}^{(\ell,t)} - \eta \cdot \nabla \hat{f}_{\mathcal{D}_t, \widetilde{\mathcal{D}}_t}(\boldsymbol{W}^{(\ell,t)}) + \beta \cdot \left( \boldsymbol{W}^{(\ell,t)} - \boldsymbol{W}^{(\ell,t-1)} \right)$

    **end for**

    $\boldsymbol{W}^{(\ell+1,0)} = \boldsymbol{W}^{(\ell,T)}$

**end for**

---

## C NOTATIONS

In this section, we first introduce some important notations that will be used in the following proofs, and the notations are summarized in Table 1.

As shown in Algorithm 1, $\boldsymbol{W}^{(\ell,t)}$ denotes the learned model after $t$ number of iterations via mini-batch stochastic gradient descent at $\ell$-th stage of the iterative self-training algorithm. Given a student model $\widetilde{\boldsymbol{W}}$, the pseudo label for $\widetilde{\boldsymbol{x}} \in \widetilde{\mathcal{D}}$ is generated as

$$\tilde{y} = g(\widetilde{\boldsymbol{W}}; \widetilde{\boldsymbol{x}}). \tag{16}$$

Further, let $\boldsymbol{W}^{[p]} = p\boldsymbol{W}^* + (1-p)\boldsymbol{W}^{(0,0)}$, we then define the *population risk function* as

$$f(\boldsymbol{W}; p) = \frac{\lambda}{2} \mathbb{E}_{\boldsymbol{x}} \left( y^*(p) - g(\boldsymbol{W}; \boldsymbol{x}) \right)^2 + \frac{\widetilde{\lambda}}{2} \mathbb{E}_{\widetilde{\boldsymbol{x}}} \left( \widetilde{y}^*(p) - g(\boldsymbol{W}; \widetilde{\boldsymbol{x}}) \right)^2, \tag{17}$$

where $y^*(p) = g(\boldsymbol{W}^{[p]}; \boldsymbol{x})$ with $\boldsymbol{x} \sim \mathcal{N}(0, \delta^2 \boldsymbol{I})$ and $\widetilde{y}^*(p) = g(\boldsymbol{W}^{[p]}; \widetilde{\boldsymbol{x}})$ with $\widetilde{\boldsymbol{x}} \sim \mathcal{N}(0, \tilde{\delta}^2 \boldsymbol{I})$. When $p = 1$, we have $\boldsymbol{W}^{[p]} = \boldsymbol{W}^*$ and $y^*(p) = y$ for data in $\mathcal{D}$.

Moreover, we use $\sigma_i$ to denote the $i$-th largest singular value of $\boldsymbol{W}^*$. Then, $\kappa$ is defined as $\sigma_1/\sigma_K$, and $\gamma = \prod_{i=1}^K \sigma_i/\sigma_K$. Additionally, to avoid high dimensional tensors, the first order derivative of the empirical risk function is defined in the form of vectorized $\boldsymbol{W}$ as

$$\nabla \hat{f}(\boldsymbol{W}) = \left[ \frac{\partial f}{\partial \boldsymbol{w}_1}^T, \frac{\partial f}{\partial \boldsymbol{w}_2}^T, \cdots, \frac{\partial f}{\partial \boldsymbol{w}_K}^T \right]^T \in \mathbb{R}^{dK} \tag{18}$$

with $\boldsymbol{W} = [\boldsymbol{w}_1, \boldsymbol{w}_2, \cdots, \boldsymbol{w}_K] \in \mathbb{R}^{d \times K}$. Therefore, the second order derivative of the empirical risk function is in $\mathbb{R}^{dk \times dk}$. Similar to (18), the high order derivatives of the population risk functions are defined based on vectorized $\boldsymbol{W}$ as well. In addition, without special descriptions, $\boldsymbol{\alpha} = [\boldsymbol{\alpha}_1^T, \boldsymbol{\alpha}_2^T, \cdots, \boldsymbol{\alpha}_K^T]^T$ stands for any unit vector that in $\mathbb{R}^{dK}$ with $\boldsymbol{\alpha}_j \in \mathbb{R}^d$. Therefore, we have

$$\|\nabla^2 \hat{f}\|_2 = \max_{\boldsymbol{\alpha}} \|\boldsymbol{\alpha}^T \nabla^2 \hat{f} \boldsymbol{\alpha}\|_2 = \max_{\boldsymbol{\alpha}} \left( \sum_{j=1}^K \boldsymbol{\alpha}_j^T \frac{\partial \hat{f}}{\partial \boldsymbol{w}_j} \right)^2. \tag{19}$$

Finally, since we focus on order-wise analysis, some constant numbers will be ignored in the majority of the steps. In particular, we use $h_1(z) \gtrsim$ (or $\lesssim, \approx)h_2(z)$ to denote there exists some positive constant $C$ such that $h_1(z) \geq$ (or $\leq, =)C \cdot h_2(z)$ when $z \in \mathbb{R}$ is sufficiently large.

Table 3: Some Important Notations

| | |
|---|---|
| $\mathcal{D} = \{\boldsymbol{x}_n, y_n\}_{n=1}^N$ | Labeled dataset with $N$ number of samples; |
| $\widetilde{\mathcal{D}} = \{\widetilde{\boldsymbol{x}}_m\}_{m=1}^M$ | Unlabeled dataset with $M$ number of samples; |
| $\mathcal{D}_t = \{\boldsymbol{x}_n, y_n\}_{n=1}^{N_t}$ | a subset of $\mathcal{D}$ with $N_t$ number of labeled data; |
| $\widetilde{\mathcal{D}}_t = \{\widetilde{\boldsymbol{x}}_m\}_{m=1}^{M_t}$ | a subset of $\widetilde{\mathcal{D}}$ with $M_t$ number of unlabeled data; |
| $d$ | Dimension of the input $\boldsymbol{x}$ or $\widetilde{\boldsymbol{x}}$; |
| $K$ | Number of neurons in the hidden layer; |
| $\boldsymbol{W}^*$ | Weights of the ground truth model; |
| $\boldsymbol{W}^{[p]}$ | $\boldsymbol{W}^{[p]} = p\boldsymbol{W}^* + (1-p)\boldsymbol{W}^{(0,0)}$; |
| $\boldsymbol{W}^{(\ell,t)}$ | Model returned by iterative self-training after $t$ step mini-batch stochastic gradient descent at stage $\ell$; $\boldsymbol{W}^{(0,0)}$ is the initial model; |
| $\hat{f}_{\mathcal{D},\widetilde{\mathcal{D}}}($ or $\hat{f})$ | The empirical risk function defined in (1); |
| $f(\boldsymbol{W};p)$ | The population risk function defined in (17); |
| $\hat{\lambda}$ | The value of $\lambda\delta^2/(\lambda\delta^2 + \widetilde{\lambda}\widetilde{\delta}^2)$; |
| $\mu$ | The value of $\frac{\lambda\delta^2+\widetilde{\lambda}\widetilde{\delta}^2}{\lambda\rho(\delta)+\widetilde{\lambda}\rho(\widetilde{\delta})}$; |
| $\sigma_i$ | The $i$-th largest singular value of $\boldsymbol{W}^*$; |
| $\kappa$ | The value of $\sigma_1/\sigma_K$; |
| $\gamma$ | The value of $\prod_{i=1}^K \sigma_i/\sigma_K$; |
| $q$ | Some large constant in $\mathbb{R}^+$; |

## D  PRELIMINARY LEMMAS

We will first start with some preliminary lemmas. As outlined at the beginning of the supplementary material, Lemma 1 illustrates the local convex region of the population risk function, and Lemma 2 explains the error bound between the population risk and empirical risk functions. Then, Lemma 3 describes the returned initial model $\boldsymbol{W}^{(0,0)}$ via tensor initialization method (Zhong et al., 2017) purely using labeled data. Next, Lemma 4 is the well known Weyl's inequality in the matrix setting. Moreover, Lemma 5 is the concentration theorem for independent random matrices. The definitions of the sub-Gaussian and sub-exponential variables are summarized in Definitions 1 and 2. Lemmas 6 and 7 serve as the technical tools in bounding matrix norms under the framework of the confidence interval.

**Lemma 1.** *Given any $\boldsymbol{W} \in \mathbb{R}^{d \times K}$, let $p$ satisfy*

$$p \lesssim \frac{\sigma_K}{\mu^2 K \cdot \|\boldsymbol{W} - \boldsymbol{W}^*\|_F}. \tag{20}$$

*Then, we have*

$$\frac{\lambda\rho(\delta) + \widetilde{\lambda}\rho(\widetilde{\delta})}{12\kappa^2\gamma K^2} \preceq \nabla^2 f(\boldsymbol{W};p) \preceq \frac{7(\lambda\delta^2 + \widetilde{\lambda}\widetilde{\delta}^2)}{K}. \tag{21}$$

**Lemma 2.** *Let $f$ and $\hat{f}$ be the functions defined in (17) and (1), respectively. Suppose the pseudo label is generated through (16) with weights $\widetilde{W}$. Then, we have*

$$
\begin{aligned}
\|\nabla f(\boldsymbol{W}) - \nabla \hat{f}(\boldsymbol{W})\|_2 \lesssim & \frac{\lambda \delta^2}{K} \sqrt{\frac{d \log q}{N}} \cdot \|\boldsymbol{W} - \boldsymbol{W}^*\| + \frac{\widetilde{\lambda} \tilde{\delta}^2}{K} \sqrt{\frac{d \log q}{M}} \cdot \|\boldsymbol{W} - \widetilde{\boldsymbol{W}}\|_2 \\
& + \frac{\left\| \lambda \delta^2 \cdot \left(\widetilde{\boldsymbol{W}} - \boldsymbol{W}^{[p]}\right) + \widetilde{\lambda} \tilde{\delta}^2 \cdot \left(\boldsymbol{W}^* - \boldsymbol{W}^{[p]}\right) \right\|_2}{2K}
\end{aligned}
\tag{22}
$$

*with probability at least $1 - q^{-d}$.*

**Lemma 3** (Initialization, (Zhong et al., 2017)). *Assuming the number of labeled data satisfies*

$$
N \geq p^2 N^*
\tag{23}
$$

*for some large constant $q$ and $p \in [\frac{1}{K}, 1]$, the tensor initialization method, which is summarized in Appendix I, outputs $\boldsymbol{W}^{(0,0)}$ such that*

$$
\|\boldsymbol{W}^{(0,0)} - \boldsymbol{W}^*\|_F \leq \frac{\sigma_K}{p \cdot c(\kappa) \mu^2 K}
\tag{24}
$$

*with probability at least $1 - q^{-d}$.*

**Lemma 4** (Weyl's inequality, (Bhatia, 2013)). *Let $\boldsymbol{B} = \boldsymbol{A} + \boldsymbol{E}$ be a matrix with dimension $m \times m$. Let $\lambda_i(\boldsymbol{B})$ and $\lambda_i(\boldsymbol{A})$ be the $i$-th largest eigenvalues of $\boldsymbol{B}$ and $\boldsymbol{A}$, respectively. Then, we have*

$$
|\lambda_i(\boldsymbol{B}) - \lambda_i(\boldsymbol{A})| \leq \|\boldsymbol{E}\|_2, \quad \forall \quad i \in [m].
\tag{25}
$$

**Lemma 5** ((Tropp, 2012), Theorem 1.6). *Consider a finite sequence $\{\boldsymbol{Z}_k\}$ of independent, random matrices with dimensions $d_1 \times d_2$. Assume that such random matrix satisfies*

$$
\mathbb{E}(\boldsymbol{Z}_k) = 0 \quad \text{and} \quad \|\boldsymbol{Z}_k\| \leq R \quad \text{almost surely.}
$$

*Define*

$$
\delta^2 := \max \left\{ \left\| \sum_k \mathbb{E}(\boldsymbol{Z}_k \boldsymbol{Z}_k^*) \right\|, \left\| \sum_k \mathbb{E}(\boldsymbol{Z}_k^* \boldsymbol{Z}_k) \right\| \right\}.
$$

*Then for all $t \geq 0$, we have*

$$
\text{Prob}\left\{ \left\| \sum_k \boldsymbol{Z}_k \right\| \geq t \right\} \leq (d_1 + d_2) \exp \left( \frac{-t^2/2}{\delta^2 + Rt/3} \right).
$$

**Definition 1** (Definition 5.7, (Vershynin, 2010)). *A random variable $X$ is called a sub-Gaussian random variable if it satisfies*

$$
(\mathbb{E}|X|^p)^{1/p} \leq c_1 \sqrt{p}
\tag{26}
$$

*for all $p \geq 1$ and some constant $c_1 > 0$. In addition, we have*

$$
\mathbb{E} e^{s(X - \mathbb{E}X)} \leq e^{c_2 \|X\|_{\psi_2}^2 s^2}
\tag{27}
$$

*for all $s \in \mathbb{R}$ and some constant $c_2 > 0$, where $\|X\|_{\phi_2}$ is the sub-Gaussian norm of $X$ defined as $\|X\|_{\psi_2} = \sup_{p \geq 1} p^{-1/2} (\mathbb{E}|X|^p)^{1/p}$.*

*Moreover, a random vector $\boldsymbol{X} \in \mathbb{R}^d$ belongs to the sub-Gaussian distribution if one-dimensional marginal $\boldsymbol{\alpha}^T \boldsymbol{X}$ is sub-Gaussian for any $\boldsymbol{\alpha} \in \mathbb{R}^d$, and the sub-Gaussian norm of $\boldsymbol{X}$ is defined as $\|\boldsymbol{X}\|_{\psi_2} = \sup_{\|\boldsymbol{\alpha}\|_2 = 1} \|\boldsymbol{\alpha}^T \boldsymbol{X}\|_{\psi_2}$.*

**Definition 2** (Definition 5.13, (Vershynin, 2010)). *A random variable $X$ is called a sub-exponential random variable if it satisfies*

$$
(\mathbb{E}|X|^p)^{1/p} \leq c_3 p
\tag{28}
$$

*for all $p \geq 1$ and some constant $c_3 > 0$. In addition, we have*

$$
\mathbb{E} e^{s(X - \mathbb{E}X)} \leq e^{c_4 \|X\|_{\psi_1}^2 s^2}
\tag{29}
$$

*for $s \leq 1/\|X\|_{\psi_1}$ and some constant $c_4 > 0$, where $\|X\|_{\psi_1}$ is the sub-exponential norm of $X$ defined as $\|X\|_{\psi_1} = \sup_{p \geq 1} p^{-1} (\mathbb{E}|X|^p)^{1/p}$.*

**Lemma 6** (Lemma 5.2, (Vershynin, 2010)). *Let $\mathcal{B}(0,1) \in \{\boldsymbol{\alpha} \big| \|\boldsymbol{\alpha}\|_2 = 1, \boldsymbol{\alpha} \in \mathbb{R}^d\}$ denote a unit ball in $\mathbb{R}^d$. Then, a subset $\mathcal{S}_\xi$ is called a $\xi$-net of $\mathcal{B}(0,1)$ if every point $\boldsymbol{z} \in \mathcal{B}(0,1)$ can be approximated to within $\xi$ by some point $\boldsymbol{\alpha} \in \mathcal{B}(0,1)$, i.e., $\|\boldsymbol{z} - \boldsymbol{\alpha}\|_2 \leq \xi$. Then the minimal cardinality of a $\xi$-net $\mathcal{S}_\xi$ satisfies*

$$|\mathcal{S}_\xi| \leq (1 + 2/\xi)^d. \tag{30}$$

**Lemma 7** (Lemma 5.3, (Vershynin, 2010)). *Let $\boldsymbol{A}$ be an $d_1 \times d_2$ matrix, and let $\mathcal{S}_\xi(d)$ be a $\xi$-net of $\mathcal{B}(0,1)$ in $\mathbb{R}^d$ for some $\xi \in (0,1)$. Then*

$$\|\boldsymbol{A}\|_2 \leq (1-\xi)^{-1} \max_{\boldsymbol{\alpha}_1 \in \mathcal{S}_\xi(d_1), \boldsymbol{\alpha}_2 \in \mathcal{S}_\xi(d_2)} |\boldsymbol{\alpha}_1^T \boldsymbol{A} \boldsymbol{\alpha}_2|. \tag{31}$$

**Lemma 8** (Mean Value Theorem). *Let $\boldsymbol{U} \subset \mathbb{R}^{n_1}$ be open and $\boldsymbol{f} : \boldsymbol{U} \longrightarrow \mathbb{R}^{n_2}$ be continuously differentiable, and $\boldsymbol{x} \in \boldsymbol{U}$, $\boldsymbol{h} \in \mathbb{R}^{n_1}$ vectors such that the line segment $\boldsymbol{x} + t\boldsymbol{h}$, $0 \leq t \leq 1$ remains in $\boldsymbol{U}$. Then we have:*

$$\boldsymbol{f}(\boldsymbol{x} + \boldsymbol{h}) - \boldsymbol{f}(\boldsymbol{x}) = \left(\int_0^1 \nabla \boldsymbol{f}(\boldsymbol{x} + t\boldsymbol{h}) dt\right) \cdot \boldsymbol{h},$$

*where $\nabla \boldsymbol{f}$ denotes the Jacobian matrix of $\boldsymbol{f}$.*

## E    PROOF OF THEOREM 2

With $p = 1$ in (17), the *population risk function* is reduced as

$$f(\boldsymbol{W}) = \frac{\lambda}{2}\mathbb{E}_{\boldsymbol{x}}(y - g(\boldsymbol{W};\boldsymbol{x})) + \frac{\widetilde{\lambda}}{2}\mathbb{E}_{\widetilde{\boldsymbol{x}}}(\widetilde{y}^* - g(\boldsymbol{W};\widetilde{\boldsymbol{x}})), \tag{32}$$

where $y = g(\boldsymbol{W}^*;\boldsymbol{x})$ with $\boldsymbol{x} \sim \mathcal{N}(0, \delta^2 \boldsymbol{I})$ and $\widetilde{y}^* = g(\boldsymbol{W}^*;\widetilde{\boldsymbol{x}})$ with $\widetilde{\boldsymbol{x}} \sim \mathcal{N}(0, \widetilde{\delta}^2 \boldsymbol{I})$. In fact, (32) can be viewed as the expectation of the *empirical risk function* in (1) given $\widetilde{y}_m = g(\boldsymbol{W}^*;\widetilde{\boldsymbol{x}}_m)$. Moreover, the ground-truth model $\boldsymbol{W}^*$ is the global optimal to (32) as well. Lemmas 9 and 10 are the special case of Lemmas 1 and 2 with $p = 1$. The proof of Theorem 2 is followed by the presentation of the two lemmas.

The main idea in proving Theorem 2 is to characterize the gradient descent term by the MVT in Lemma 8 as shown in (36) and (37). The IVT is not directly applied in the empirical risk function because of its non-smoothness. However, the population risk functions defined in (17) and (32), which are the expectations over the Gaussian variables, are smooth. Then, as the distance $\|\nabla f(\boldsymbol{W}) - \nabla f(\boldsymbol{W}^*)\|_F$ is upper bounded by a linear function of $\|\boldsymbol{W} - \boldsymbol{W}^*\|_F$ as shown in (47), we can establish the connection between $\|\boldsymbol{W}^{(\ell,t+1)} - \boldsymbol{W}^*\|_F$ and $\|\boldsymbol{W}^{(\ell,t)} - \boldsymbol{W}^*\|_F$ as shown in (50). Finally, by mathematical induction over $\ell$ and $t$, one can characterize $\|\boldsymbol{W}^{(L,0)} - \boldsymbol{W}^*\|_F$ by $\|\boldsymbol{W}^{(0,0)} - \boldsymbol{W}^*\|_F$ as shown in (52), which completes the whole proof.

**Lemma 9** (Lemma 1 with $p = 1$). *Let $f$ and $\hat{f}$ are the functions defined in (32) and (1), respectively. Then, for any $\boldsymbol{W}$ that satisfies,*

$$\|\boldsymbol{W} - \boldsymbol{W}^*\|_F \leq \frac{\sigma_K}{\mu^2 K}, \tag{33}$$

*we have*

$$\frac{\lambda\rho(\delta) + \widetilde{\lambda}\rho(\tilde{\delta})}{12\kappa^2\gamma K^2} \preceq \nabla^2 f(\boldsymbol{W}) \preceq \frac{7(\lambda\delta^2 + \widetilde{\lambda}\tilde{\delta}^2)}{K}. \tag{34}$$

**Lemma 10** (Lemma 2 with $p = 1$). *Let $f$ and $\hat{f}$ be the functions defined in (32) and (1), respectively. Suppose the pseudo label is generated through (16) with weights $\widetilde{\boldsymbol{W}}$. Then, we have*

$$\|\nabla f(\boldsymbol{W}) - \nabla \hat{f}(\boldsymbol{W})\|_2 \lesssim \left(\frac{\lambda\delta^2}{K}\sqrt{\frac{d\log q}{N}} + \frac{(1-\lambda)\tilde{\delta}^2}{K}\sqrt{\frac{d\log q}{M}}\right) \cdot \|\boldsymbol{W} - \boldsymbol{W}^*\|_2$$
$$+ \frac{(1-\lambda)\tilde{\delta}^2}{K}\left(\sqrt{\frac{d\log q}{M}} + \frac{1}{2}\right) \cdot \|\widetilde{\boldsymbol{W}} - \boldsymbol{W}^*\|_2 \tag{35}$$

*with probability at least $1 - q^{-d}$.*

*Proof of Theorem 2.* From Algorithm 1, in the $\ell$-th outer loop, we have

$$
\begin{aligned}
\boldsymbol{W}^{(\ell,t+1)} =& \boldsymbol{W}^{(\ell,t)} - \eta\nabla\hat{f}_{\mathcal{D}_t,\widetilde{\mathcal{D}}_t}(\boldsymbol{W}^{(\ell,t)}) + \beta(\boldsymbol{W}^{(\ell,t)} - \boldsymbol{W}^{(\ell,t-1)}) \\
=& \boldsymbol{W}^{(\ell,t)} - \eta\nabla f(\boldsymbol{W}^{(\ell,t)}) + \beta(\boldsymbol{W}^{(\ell,t)} - \boldsymbol{W}^{(\ell,t-1)}) \\
& + \eta \cdot \big(\nabla f(\boldsymbol{W}^{(\ell,t)}) - \nabla\hat{f}_{\mathcal{D}_t,\widetilde{\mathcal{D}}_t}(\boldsymbol{W}^{(\ell,t)})\big).
\end{aligned}
\tag{36}
$$

Since $\nabla f$ is a smooth function and $\boldsymbol{W}^*$ is a local (global) optimal to $f$, then we have

$$
\begin{aligned}
\nabla f(\boldsymbol{W}^{(\ell,t)}) =& \nabla f(\boldsymbol{W}^{(\ell,t)}) - \nabla f(\boldsymbol{W}^*) \\
=& \int_0^1 \nabla^2 f\big(\boldsymbol{W}^{(\ell,t)} + u \cdot (\boldsymbol{W}^{(\ell,t)} - \boldsymbol{W}^*)\big) du \cdot (\boldsymbol{W}^{(\ell,t)} - \boldsymbol{W}^*),
\end{aligned}
\tag{37}
$$

where the last equality comes from MVT in Lemma 8. For notational convenience, we use $\boldsymbol{H}^{(\ell,t)}$ to denote the integration as

$$
\boldsymbol{H}^{(\ell,t)} := \int_0^1 \nabla^2 f\big(\boldsymbol{W}^{(\ell,t)} + u \cdot (\boldsymbol{W}^{(\ell,t)} - \boldsymbol{W}^*)\big) du.
\tag{38}
$$

Then, we have

$$
\begin{aligned}
\begin{bmatrix} \boldsymbol{W}^{(\ell,t+1)} - \boldsymbol{W}^* \\ \boldsymbol{W}^{(\ell,t)} - \boldsymbol{W}^* \end{bmatrix} =& \begin{bmatrix} \boldsymbol{I} - \eta\boldsymbol{H}^{(\ell,t)} & \beta\boldsymbol{I} \\ \boldsymbol{I} & \boldsymbol{0} \end{bmatrix} \begin{bmatrix} \boldsymbol{W}^{(\ell,t)} - \boldsymbol{W}^* \\ \boldsymbol{W}^{(\ell,t-1)} - \boldsymbol{W}^* \end{bmatrix} \\
& + \eta \begin{bmatrix} \nabla f(\boldsymbol{W}^{(\ell,t)}) - \nabla\hat{f}_{\mathcal{D}_t,\widetilde{\mathcal{D}}_t}(\boldsymbol{W}^{(\ell,t)}) \\ \boldsymbol{0} \end{bmatrix}.
\end{aligned}
\tag{39}
$$

Let $\boldsymbol{H}^{(\ell,t)} = \boldsymbol{S}\boldsymbol{\Lambda}\boldsymbol{S}^T$ be the eigen-decomposition of $\boldsymbol{H}^{(\ell,t)}$. Then, we define

$$
\boldsymbol{A}(\beta) := \begin{bmatrix} \boldsymbol{S}^T & \boldsymbol{0} \\ \boldsymbol{0} & \boldsymbol{S}^T \end{bmatrix} \boldsymbol{A}(\beta) \begin{bmatrix} \boldsymbol{S} & \boldsymbol{0} \\ \boldsymbol{0} & \boldsymbol{S} \end{bmatrix} = \begin{bmatrix} \boldsymbol{I} - \eta\boldsymbol{\Lambda} + \beta\boldsymbol{I} & \beta\boldsymbol{I} \\ \boldsymbol{I} & \boldsymbol{0} \end{bmatrix}.
\tag{40}
$$

Since $\begin{bmatrix} \boldsymbol{S} & \boldsymbol{0} \\ \boldsymbol{0} & \boldsymbol{S} \end{bmatrix} \begin{bmatrix} \boldsymbol{S}^T & \boldsymbol{0} \\ \boldsymbol{0} & \boldsymbol{S}^T \end{bmatrix} = \begin{bmatrix} \boldsymbol{I} & \boldsymbol{0} \\ \boldsymbol{0} & \boldsymbol{I} \end{bmatrix}$, we know $\boldsymbol{A}(\beta)$ and $\begin{bmatrix} \boldsymbol{I} - \eta\boldsymbol{\Lambda} + \beta\boldsymbol{I} & \beta\boldsymbol{I} \\ \boldsymbol{I} & \boldsymbol{0} \end{bmatrix}$ share the same eigenvalues. Let $\gamma_i^{(\boldsymbol{\Lambda})}$ be the $i$-th eigenvalue of $\nabla^2 f(\widehat{\boldsymbol{w}}^{(t)})$, then the corresponding $i$-th eigenvalue of (40), denoted by $\gamma_i^{(\boldsymbol{A})}$, satisfies

$$
(\gamma_i^{(\boldsymbol{A})}(\beta))^2 - (1 - \eta\gamma_i^{(\boldsymbol{\Lambda})} + \beta)\gamma_i^{(\boldsymbol{A})}(\beta) + \beta = 0.
\tag{41}
$$

By simple calculation, we have

$$
|\gamma_i^{(\boldsymbol{A})}(\beta)| = \begin{cases} \sqrt{\beta}, & \text{if} \quad \beta \geq \big(1 - \sqrt{\eta\gamma_i^{(\boldsymbol{\Lambda})}}\big)^2, \\ \frac{1}{2}\Big|(1 - \eta\gamma_i^{(\boldsymbol{\Lambda})} + \beta) + \sqrt{(1 - \eta\gamma_i^{(\boldsymbol{\Lambda})} + \beta)^2 - 4\beta}\Big|, & \text{otherwise.} \end{cases}
\tag{42}
$$

Specifically, we have

$$
\gamma_i^{(\boldsymbol{A})}(0) > \gamma_i^{(\boldsymbol{A})}(\beta), \quad \text{for} \quad \forall\beta \in \big(0, (1 - \eta\gamma_i^{(\boldsymbol{\Lambda})})^2\big),
\tag{43}
$$

and $\gamma_i^{(\boldsymbol{A})}$ achieves the minimum $\gamma_i^{(\boldsymbol{A})*} = \Big|1 - \sqrt{\eta\gamma_i^{(\boldsymbol{\Lambda})}}\Big|$ when $\beta = \Big(1 - \sqrt{\eta\gamma_i^{(\boldsymbol{\Lambda})}}\Big)^2$. From Lemma 9, for any $\boldsymbol{a} \in \mathbb{R}^d$ with $\|\boldsymbol{a}\|_2 = 1$, we have

$$
\boldsymbol{a}^T\nabla f(\boldsymbol{W}^{(\ell,t)})\boldsymbol{a} = \int_0^1 \boldsymbol{a}^T\nabla^2 f\big(\boldsymbol{W}^{(\ell,t)} + u \cdot (\boldsymbol{W}^{(\ell,t)} - \boldsymbol{W}^*)\big)\boldsymbol{a}\,du \leq \int_0^1 \gamma_{\max}\|\boldsymbol{a}\|_2^2 du = \gamma_{\max},
$$

$$
\boldsymbol{a}^T\nabla f(\boldsymbol{W}^{(\ell,t)})\boldsymbol{a} = \int_0^1 \boldsymbol{a}^T\nabla^2 f\big(\boldsymbol{W}^{(\ell,t)} + u \cdot (\boldsymbol{W}^{(\ell,t)} - \boldsymbol{W}^*)\big)\boldsymbol{a}\,du \geq \int_0^1 \gamma_{\min}\|\boldsymbol{a}\|_2^2 du = \gamma_{\min},
$$

$$
\tag{44}
$$

where $\gamma_{\max} = \frac{7(\lambda\delta^2 + \widetilde{\lambda}\tilde{\delta}^2)}{K}$, and $\gamma_{\min} = \frac{\lambda\rho(\delta) + \widetilde{\lambda}\rho(\tilde{\delta})}{12\kappa^2\gamma K^2}$. Therefore, we have

$$\gamma_{\min}^{(\boldsymbol{\Lambda})} = \frac{\lambda\rho(\delta) + \widetilde{\lambda}\rho(\tilde{\delta})}{12\kappa^2\gamma K^2}, \quad \text{and} \quad \gamma_{\max}^{(\boldsymbol{\Lambda})} = \frac{7(\lambda\delta^2 + \widetilde{\lambda}\tilde{\delta}^2)}{K}. \tag{45}$$

Thus, we can select $\eta = \left(\frac{1}{\sqrt{\gamma_{\max}^{(\boldsymbol{\Lambda})}} + \sqrt{\gamma_{\min}^{(\boldsymbol{\Lambda})}}}\right)^2$, and $\|\boldsymbol{A}(\beta)\|_2$ can be bounded by

$$\begin{aligned}
\min_\beta \|\boldsymbol{A}(\beta)\|_2 &\leq 1 - \sqrt{\left(\frac{\lambda\rho(\delta) + \widetilde{\lambda}\rho(\tilde{\delta})}{12\kappa^2\gamma K^2}\right) / \left(2 \cdot \frac{7(\lambda\delta^2 + \widetilde{\lambda}\tilde{\delta}^2)}{K}\right)} \\
&= 1 - \frac{\mu(\delta, \tilde{\delta})}{\sqrt{168\kappa^2\gamma K}},
\end{aligned} \tag{46}$$

where $\mu(\delta, \tilde{\delta}) = \left(\frac{\lambda\rho(\delta) + \widetilde{\lambda}\rho(\tilde{\delta})}{\lambda\delta^2 + \widetilde{\lambda}\tilde{\delta}^2}\right)^{1/2}$.

From Lemma 10, we have

$$\begin{aligned}
\|\nabla f(\boldsymbol{W}^{(\ell,t)}) - \nabla\hat{f}(\boldsymbol{W}^{(\ell,t)})\|_2 &= \left(\frac{\lambda\delta^2}{K}\sqrt{\frac{d\log q}{N_t}} + \frac{\widetilde{\lambda}\tilde{\delta}^2}{K}\sqrt{\frac{d\log q}{M_t}}\right) \cdot \|\boldsymbol{W}^{(\ell,t)} - \boldsymbol{W}^*\|_2 \\
&\quad + \frac{\widetilde{\lambda}\tilde{\delta}^2}{K}\left(\sqrt{\frac{d\log q}{M_t}} + \frac{1}{2}\right) \cdot \|\boldsymbol{W}^{(\ell,0)} - \boldsymbol{W}^*\|_2.
\end{aligned} \tag{47}$$

Given $\varepsilon > 0$ and $\tilde{\varepsilon} > 0$ with $\varepsilon + \tilde{\varepsilon} < 1$, let

$$\begin{aligned}
\eta \cdot \frac{\lambda\delta^2}{K}\sqrt{\frac{d\log q}{N_t}} &\leq \frac{\varepsilon\mu(\delta, \tilde{\delta})}{\sqrt{168\kappa^2\gamma K}}, \\
\text{and} \quad \eta \cdot \frac{\widetilde{\lambda}\tilde{\delta}^2}{K}\sqrt{\frac{d\log q}{M_t}} &\leq \frac{\tilde{\varepsilon}\mu(\delta, \tilde{\delta})}{\sqrt{168\kappa^2\gamma K}},
\end{aligned} \tag{48}$$

where we need

$$\begin{aligned}
N_t &\geq \varepsilon^{-2}\mu^{-2}\left(\frac{\lambda\delta^2}{\lambda\delta^2 + \widetilde{\lambda}\tilde{\delta}^2}\right)^2 \kappa^2\gamma K^3 d\log q, \\
\text{and} \quad M_t &\geq \tilde{\varepsilon}^{-2}\mu^{-2}\left(\frac{\widetilde{\lambda}\tilde{\delta}^2}{\lambda\delta^2 + \widetilde{\lambda}\tilde{\delta}^2}\right)^2 \kappa^2\gamma K^3 d\log q.
\end{aligned} \tag{49}$$

Therefore, from (46), (47) and (48), we have

$$\begin{aligned}
&\|\boldsymbol{W}^{(\ell,t+1)} - \boldsymbol{W}^*\|_2 \\
&\leq \left(1 - \frac{(1 - \varepsilon - \tilde{\varepsilon})\mu(\delta, \tilde{\delta})}{\sqrt{168\kappa^2\gamma K}}\right)\|\boldsymbol{W}^{(\ell,t)} - \boldsymbol{W}^*\|_2 + \eta \cdot \frac{\widetilde{\lambda}\tilde{\delta}^2}{K}\left(\sqrt{\frac{d\log q}{M_t}} + \frac{1}{2}\right) \cdot \|\boldsymbol{W}^{(\ell,0)} - \boldsymbol{W}^*\|_2 \\
&\leq \left(1 - \frac{(1 - \varepsilon - \tilde{\varepsilon})\mu(\delta, \tilde{\delta})}{\sqrt{168\kappa^2\gamma K}}\right)\|\boldsymbol{W}^{(\ell,t)} - \boldsymbol{W}^*\|_2 + \eta \cdot \frac{\widetilde{\lambda}\tilde{\delta}^2}{K}\|\boldsymbol{W}^{(\ell,0)} - \boldsymbol{W}^*\|_2
\end{aligned} \tag{50}$$

when $M \geq 4d\log q$. By mathematical induction on (50) over $t$, we have

$$\begin{aligned}
&\|\boldsymbol{W}^{(\ell,t)} - \boldsymbol{W}^*\|_2 \\
&\leq \left(1 - \frac{(1 - \varepsilon - \tilde{\varepsilon})\mu}{\sqrt{168\kappa^2\gamma K}}\right)^t \cdot \|\boldsymbol{W}^{(\ell,0)} - \boldsymbol{W}^*\|_2 \\
&\quad + \frac{\sqrt{168\kappa^2\gamma K}}{(1 - \varepsilon - \tilde{\varepsilon})\mu} \cdot \frac{\sqrt{K}}{14(\lambda\delta^2 + \widetilde{\lambda}\tilde{\delta}^2)} \cdot \frac{\widetilde{\lambda}\tilde{\delta}^2}{K}\|\boldsymbol{W}^{(\ell,0)} - \boldsymbol{W}^*\|_2 \\
&\leq \left[\left(1 - \frac{(1 - \varepsilon - \tilde{\varepsilon})\mu}{\sqrt{168\kappa^2\gamma K}}\right)^t + \frac{\sqrt{\kappa^2\gamma}\widetilde{\lambda}\tilde{\delta}^2}{(1 - \varepsilon - \tilde{\varepsilon})\mu(\lambda\delta^2 + \widetilde{\lambda}\tilde{\delta}^2)}\right] \cdot \|\boldsymbol{W}^{(\ell,0)} - \boldsymbol{W}^*\|_2
\end{aligned} \tag{51}$$

By mathematical induction on (51) over $\ell$, we have

$$
\begin{aligned}
&\|\boldsymbol{W}^{(\ell,T)} - \boldsymbol{W}^*\|_2 \\
&\leq \left[\left(1 - \frac{(1 - \varepsilon - \tilde{\varepsilon})\mu}{\sqrt{168\kappa^2\gamma K}}\right)^T + \frac{\sqrt{\kappa^2\gamma}\widetilde{\lambda}\tilde{\delta}^2}{(1 - \varepsilon - \tilde{\varepsilon})\mu(\lambda\delta^2 + \widetilde{\lambda}\tilde{\delta}^2)}\right]^\ell \cdot \|\boldsymbol{W}^{(0,0)} - \boldsymbol{W}^*\|_2
\end{aligned}
\tag{52}
$$

$\square$

## F   PROOF OF THEOREM 1

Instead of proving Theorem 1, we turn to prove a stronger version, as shown in Theorem 3. One can verify that Theorem 1 is a special case of Theorem 3 by selecting $\hat{\lambda}$ in the order of $p$ and $\widetilde{\varepsilon}$ is in the order of $(2p - 1)$.

The major idea in proving Theorem 3 is similar to that of Theorem 2. The first step is to characterize the gradient descent term on the population risk function by the MVT in Lemma 8 as shown in (58) and (59). Then, the connection between $\|\boldsymbol{W}^{(\ell+1,0)} - \boldsymbol{W}^{[p]}\|_F$ and $\|\boldsymbol{W}^{(\ell,0)} - \boldsymbol{W}^{[p]}\|_F$ are characterized in (64). Compared with proving Theorem 2, where the induction over $\ell$ holds naturally with large size of labeled data, the induction over $\ell$ requires a proper value of $p$ as shown in (69). By induction over $\ell$ on (64), the relative error $\|\boldsymbol{W}^{(L,0)} - \boldsymbol{W}^{[p]}\|_F$ can be characterized by $\|\boldsymbol{W}^{(0,0)} - \boldsymbol{W}^{[p]}\|_F$ as shown in (71).

**Theorem 3.** *Suppose the initialization $\boldsymbol{W}^{(0,0)}$ satisfies with*

$$
|p - \hat{\lambda}| \leq \frac{2(1 - \widetilde{\varepsilon})p - 1}{\mu\sqrt{K}}
\tag{53}
$$

*for some constant $\widetilde{\varepsilon} \in (0, 1/2)$, where*

$$
\hat{\lambda} := \frac{\lambda\delta^2}{\lambda\delta^2 + \widetilde{\lambda}\tilde{\delta}^2} = \left(\frac{N}{\kappa^2\gamma K^3\mu^2 d \log q}\right)^{\frac{1}{2}}
\tag{54}
$$

*and*

$$
\mu = \mu(\delta, \tilde{\delta}) = \frac{\lambda\delta^2 + \widetilde{\lambda}\tilde{\delta}^2}{\lambda\rho(\delta) + \widetilde{\lambda}\rho(\tilde{\delta})}.
\tag{55}
$$

*Then, if the number of samples in $\widetilde{\mathcal{D}}$ further satisfies*

$$
M \gtrsim \tilde{\varepsilon}^{-2}\kappa^2\gamma\mu^2\left(1 - \hat{\lambda}\right)^2 K^3 d \log q,
\tag{56}
$$

*the iterates $\{\boldsymbol{W}^{(\ell,t)}\}_{\ell,t=0}^{L,T}$ converge to $\boldsymbol{W}^{[p]}$ with $p$ satisfies (53) as*

$$
\begin{aligned}
&\lim_{T\to\infty} \|\boldsymbol{W}^{(\ell,T)} - \boldsymbol{W}^{[p]}\|_2 \\
&\leq \frac{1}{1 - \widetilde{\varepsilon}} \cdot \left(1 - p^* + \mu\sqrt{K}\left|(\hat{\lambda} - p^*)\right|\right) \cdot \|\boldsymbol{W}^{(0,0)} - \boldsymbol{W}^*\|_2 + \frac{\tilde{\varepsilon}}{(1 - \widetilde{\varepsilon})} \cdot \|\boldsymbol{W}^{(\ell,0)} - \boldsymbol{W}^{[p]}\|_2,
\end{aligned}
\tag{57}
$$

*with probability at least $1 - q^{-d}$.*

*Proof of Theorem 3.* From Algorithm 1, in the $\ell$-th outer loop, we have

$$
\begin{aligned}
\boldsymbol{W}^{(\ell,t+1)} &= \boldsymbol{W}^{(\ell,t)} - \eta\nabla\hat{f}_{\mathcal{D}_t,\widetilde{\mathcal{D}}_t}(\boldsymbol{W}^{(\ell,t)}) + \beta(\boldsymbol{W}^{(\ell,t)} - \boldsymbol{W}^{(\ell,t-1)}) \\
&= \boldsymbol{W}^{(\ell,t)} - \eta\nabla f(\boldsymbol{W}^{(\ell,t)}) + \beta(\boldsymbol{W}^{(\ell,t)} - \boldsymbol{W}^{(\ell,t-1)}) \\
&\quad + \eta \cdot \left(\nabla f(\boldsymbol{W}^{(\ell,t)}) - \nabla\hat{f}_{\mathcal{D}_t,\widetilde{\mathcal{D}}_t}(\boldsymbol{W}^{(\ell,t)})\right)
\end{aligned}
\tag{58}
$$

Since $\nabla f$ is a smooth function and $\boldsymbol{W}^{[p]}$ is a local (global) optimal to $f$, then we have

$$
\begin{aligned}
\nabla f(\boldsymbol{W}^{(\ell,t)}) &= \nabla f(\boldsymbol{W}^{(\ell,t)}) - \nabla f(\boldsymbol{W}^{[p]}) \\
&= \int_0^1 \nabla^2 f\left(\boldsymbol{W}^{(\ell,t)} + u \cdot (\boldsymbol{W}^{(\ell,t)} - \boldsymbol{W}^{[p]})\right) du \cdot (\boldsymbol{W}^{(\ell,t)} - \boldsymbol{W}^{[p]}),
\end{aligned}
\tag{59}
$$

where the last equality comes from Lemma 8.

Similar to the proof of Theorem 2, we have

$$\|\boldsymbol{W}^{(\ell,t+1)} - \boldsymbol{W}^{[p]}\|_2 \leq \|\boldsymbol{A}(\beta)\|_2 \cdot \|\boldsymbol{W}^{(\ell,t)} - \boldsymbol{W}^{[p]}\|_2 + \eta \cdot \|\nabla f(\boldsymbol{W}^{(\ell,t)}) - \nabla \hat{f}_{\mathcal{D}_t, \widetilde{\mathcal{D}}_t}(\boldsymbol{W}^{(\ell,t)})\|_2. \quad (60)$$

From Lemma 2, we have

$$
\begin{aligned}
&\|\nabla f(\boldsymbol{W}^{(\ell,t)}) - \nabla \hat{f}(\boldsymbol{W}^{(\ell,t)})\|_2 \\
&\lesssim \frac{\lambda \delta^2}{K} \sqrt{\frac{d \log q}{N_t}} \cdot \|\boldsymbol{W}^{(\ell,t)} - \boldsymbol{W}^*\| + \frac{\widetilde{\lambda} \tilde{\delta}^2}{K} \sqrt{\frac{d \log q}{M_t}} \cdot \|\boldsymbol{W}^{(\ell,t)} - \boldsymbol{W}^{(\ell,0)}\|_2 \\
&\quad + \frac{\left|\lambda \delta^2 \cdot (\boldsymbol{W}^{(0,0)} - \boldsymbol{W}^{[p]}) - \widetilde{\lambda} \tilde{\delta}^2 \cdot (\boldsymbol{W}^* - \boldsymbol{W}^{[p]})\right|}{K}
\end{aligned}
\quad (61)
$$

When $\ell = 0$, following the similar steps from (41) to (46), we have

$$
\begin{aligned}
&\|\nabla f(\boldsymbol{W}^{(\ell,t)}) - \nabla \hat{f}(\boldsymbol{W}^{(\ell,t)})\|_2 \\
&\lesssim \frac{\lambda \delta^2}{K} \sqrt{\frac{d \log q}{N_t}} \cdot \|\boldsymbol{W}^{(\ell,t)} - \boldsymbol{W}^{[p]}\| + \frac{\widetilde{\lambda} \tilde{\delta}^2}{K} \sqrt{\frac{d \log q}{M_t}} \cdot \|\boldsymbol{W}^{(\ell,t)} - \boldsymbol{W}^{[p]}\|_2 \\
&\quad + \frac{\lambda \delta^2}{K} \sqrt{\frac{d \log q}{N_t}} \cdot \|\boldsymbol{W}^* - \boldsymbol{W}^{[p]}\| + \frac{\widetilde{\lambda} \tilde{\delta}^2}{K} \sqrt{\frac{d \log q}{M_t}} \cdot \|\boldsymbol{W}^{(0,0)} - \boldsymbol{W}^{[p]}\|_2 \\
&\quad + \frac{\left|\lambda \delta^2 \cdot (1-p) - \widetilde{\lambda} \tilde{\delta}^2 \cdot p\right|}{K} \cdot \|\boldsymbol{W}^{(0,0)} - \boldsymbol{W}^*\|_2
\end{aligned}
\quad (62)
$$

and

$$
\begin{aligned}
&\|\boldsymbol{W}^{(\ell,t+1)} - \boldsymbol{W}^{[p]}\|_2 \\
&\leq \left(1 - \frac{1 - \tilde{\varepsilon}}{\mu(\delta, \tilde{\delta}) \sqrt{154 \kappa^2 \gamma K}}\right) \cdot \|\boldsymbol{W}^{(\ell,t)} - \boldsymbol{W}^{[p]}\|_2 \\
&\quad + \eta \cdot \left(\frac{\lambda \delta^2 (1-p)}{K} \sqrt{\frac{d \log q}{N_t}} + \frac{\left|\lambda \delta^2 \cdot (1-p) - \widetilde{\lambda} \tilde{\delta}^2 \cdot p\right|}{K}\right) \cdot \|\boldsymbol{W}^{(0,0)} - \boldsymbol{W}^*\|_2 \\
&\quad + \eta \cdot \frac{\tilde{\varepsilon} \widetilde{\lambda} \tilde{\delta}^2 \cdot p}{K} \cdot \sqrt{\frac{d \log q}{M_t}} \|\boldsymbol{W}^{(0,0)} - \boldsymbol{W}^*\|_2.
\end{aligned}
\quad (63)
$$

Therefore, we have

$$
\begin{aligned}
&\lim_{T \to \infty} \|\boldsymbol{W}^{(\ell,T)} - \boldsymbol{W}^{[p]}\|_2 \\
&\leq \frac{\mu \sqrt{154 \kappa^2 \gamma K}}{1 - \tilde{\varepsilon}} \cdot \eta \cdot \left[\left(\frac{\lambda \delta^2 (1-p)}{K} \sqrt{\frac{d \log q}{N_t}} + \frac{\left|\lambda \delta^2 \cdot (1-p) - \widetilde{\lambda} \tilde{\delta}^2 \cdot p\right|}{K}\right) \cdot \|\boldsymbol{W}^{(0,0)} - \boldsymbol{W}^*\|_2 \right. \\
&\qquad \left. + \frac{\tilde{\varepsilon} \widetilde{\lambda} \tilde{\delta}^2 \cdot p}{K} \cdot \sqrt{\frac{d \log q}{M_t}} \cdot \|\boldsymbol{W}^{(0,0)} - \boldsymbol{W}^*\|_2\right] \\
&\leq \frac{\mu \sqrt{154 \kappa^2 \gamma K}}{1 - \tilde{\varepsilon}} \cdot \frac{K}{14(\lambda \delta^2 + \widetilde{\lambda} \tilde{\delta}^2)} \cdot \left[\left(\frac{\lambda \delta^2 (1-p)}{K} \sqrt{\frac{d \log q}{N_t}} + \frac{\left|\lambda \delta^2 \cdot (1-p) - \widetilde{\lambda} \tilde{\delta}^2 \cdot p\right|}{K}\right) \right. \\
&\qquad \left. \cdot \|\boldsymbol{W}^{(0,0)} - \boldsymbol{W}^*\|_2 + \frac{\tilde{\varepsilon} \widetilde{\lambda} \tilde{\delta}^2 \cdot p}{K} \cdot \sqrt{\frac{d \log q}{M_t}} \cdot \|\boldsymbol{W}^{(0,0)} - \boldsymbol{W}^*\|_2\right] \\
&\simeq \frac{1}{1 - \tilde{\varepsilon}} \cdot \left(1 - p + \sqrt{K} \cdot \left|(1-p)\mu \hat{\lambda} - p\mu(1 - \hat{\lambda})\right|\right) \cdot \|\boldsymbol{W}^{(0,0)} - \boldsymbol{W}^*\|_2 \\
&\quad + \frac{\tilde{\varepsilon} p}{(1 - \tilde{\varepsilon})} \cdot \|\boldsymbol{W}^{(0,0)} - \boldsymbol{W}^*\|_2 \\
&= \frac{1}{1 - \tilde{\varepsilon}} \cdot \left(1 - p + \mu \sqrt{K}\left|\widehat{\lambda} - p\right|\right) \cdot \|\boldsymbol{W}^{(0,0)} - \boldsymbol{W}^*\|_2 + \frac{\tilde{\varepsilon} p}{(1 - \tilde{\varepsilon})} \cdot \|\boldsymbol{W}^{(0,0)} - \boldsymbol{W}^*\|_2,
\end{aligned}
\quad (64)
$$

where $\hat{\lambda} = \frac{\lambda \delta^2}{\lambda \delta^2 + \widetilde{\lambda} \tilde{\delta}^2}$.

To guarantee the convergence in the outer loop, we require

$$\lim_{T\to\infty} \|\boldsymbol{W}^{(\ell,T)} - \boldsymbol{W}^{[p]}\|_2 \le \|\boldsymbol{W}^{(0,0)} - \boldsymbol{W}^{[p]}\|_2 = p\|\boldsymbol{W}^{(0,0)} - \boldsymbol{W}^*\|_2,$$

$$\text{and} \quad \lim_{T\to\infty} \|\boldsymbol{W}^{(\ell,T)} - \boldsymbol{W}^*\|_2 \le \|\boldsymbol{W}^{(0,0)} - \boldsymbol{W}^*\|_2. \tag{65}$$

Since we have

$$\|\boldsymbol{W}^{(\ell,T)} - \boldsymbol{W}^{[p]}\|_2 \le \|\boldsymbol{W}^{(\ell,T)} - \boldsymbol{W}^*\|_2 + \|\boldsymbol{W}^* - \boldsymbol{W}^{[p]}\|_2$$

$$= \|\boldsymbol{W}^{(\ell,T)} - \boldsymbol{W}^*\|_2 + (1-p) \cdot \|\boldsymbol{W}^* - \boldsymbol{W}^{(0,0)}\|_2, \tag{66}$$

it is clear that (65) holds if and only if

$$\frac{1}{1-\widetilde{\varepsilon}} \cdot \left(1 - p + \widetilde{\varepsilon}p + \mu\sqrt{K}|\hat{\lambda} - p|\right) + 1 - p \le 1. \tag{67}$$

To guarantee the iterates strictly converges to the desired point, we let

$$\frac{1}{1-\widetilde{\varepsilon}} \cdot \left(1 - p + \widetilde{\varepsilon}p + \mu\sqrt{K}|\hat{\lambda} - p|\right) + 1 - p \le 1 - \frac{1}{C} \tag{68}$$

for some larger constant $C$, which is equivalent to

$$|p - \hat{\lambda}| \le \frac{2(1-\widetilde{\varepsilon})p - 1}{\mu\sqrt{K}}. \tag{69}$$

To make the bound in (69) meaningful, we need

$$p \ge \frac{1}{2(1-\widetilde{\varepsilon})}. \tag{70}$$

When $\ell > 1$, following similar steps in (64), we have

$$\lim_{T\to\infty} \|\boldsymbol{W}^{(\ell,T)} - \boldsymbol{W}^{[p]}\|_2$$

$$\le \frac{1}{1-\widetilde{\varepsilon}} \cdot \left(1 - p + \mu\sqrt{K}|(\hat{\lambda} - p)|\right) \cdot \|\boldsymbol{W}^{(0,0)} - \boldsymbol{W}^*\|_2 + \frac{\widetilde{\varepsilon}p}{1-\widetilde{\varepsilon}} \cdot \|\boldsymbol{W}^{(\ell,0)} - \boldsymbol{W}^{[p]}\|_2, \tag{71}$$

Given (69) holds, from (71), we have

$$\lim_{L\to\infty, T\to\infty} \|\boldsymbol{W}^{(L,T)} - \boldsymbol{W}^{[p]}\|_2$$

$$\le \frac{1}{1-\widetilde{\varepsilon}} \cdot \left(1 - p + \mu\sqrt{K}|\hat{\lambda} - p|\right) \cdot \|\boldsymbol{W}^{(0,0)} - \boldsymbol{W}^*\|_2 \tag{72}$$

$$\le \frac{1}{1-\widetilde{\varepsilon}} \cdot \left(1 - p + \mu\sqrt{K}|\hat{\lambda} - p|\right) \cdot \|\boldsymbol{W}^{(0,0)} - \boldsymbol{W}^*\|_2.$$

$$\square$$

## G  DEFINITION AND RELATIVE PROOFS OF $\rho$

In this section, the formal definition of $\rho$ is included in Definition 3, and a corresponding claim about $\rho$ is summarized in Lemma 11. One can quickly check that the ReLU activation function satisfies the conditions in Lemma 11.

The major idea in proving Lemma 11 is to show $H_r(\delta)$ and $J_r(\delta)$ in Definition 3 are in the order of $\delta^r$ when $\delta$ is small.

**Definition 3.** *Let* $H_r(\delta) = \mathbb{E}_{z\sim\mathcal{N}(0,\delta^2)}\big(\phi'(\sigma_K z)z^r\big)$ *and* $J_r(\delta) = \mathbb{E}_{z\sim\mathcal{N}(0,\delta^2)}\big(\phi'^2(\sigma_K z)z^r\big)$. *Then,* $\rho = \rho(\delta)$ *is defined as*

$$\rho(\delta) = \min\left\{J_0(\delta) - H_0^2(\delta) - H_1^2(\delta), J_2(\delta) - H_1^2(\delta) - H_2^2(\delta), H_0(\delta) \cdot H_2(\delta) - H_1^2(\delta)\right\}, \tag{73}$$

*where* $\sigma_K$ *is the minimal singular value of* $\boldsymbol{W}^*$.

**Lemma 11** (Order analysis of $\rho$). *If $\rho(\delta) > 0$ for $\delta \in (0, \xi)$ for some positive constant $\xi$ and the sub-gradient of $\rho(\delta)$ at $0$ can be non-zero, then $\rho(\delta) = \Theta(\delta^2)$ when $\delta \to 0^+$. Typically, for ReLU activation function, $\mu$ in (5) is a fixed constant for all $\delta, \tilde{\delta} \leq 1$.*

*Proof of Lemma 11 .* From Definition 3, we know that $H_r(\delta) = \mathbb{E}_{\boldsymbol{z} \sim \mathcal{N}(0, \delta^2)} \phi'(\sigma_K z) z^r$. Suppose we have $H_r(\delta) = \Theta(\delta^r)$ and $J_r(\delta) = \Theta(\delta^r)$, then from (73) we have

$$
\begin{aligned}
J_0(\delta) - H_0^2(\delta) - H_1^2(\delta) &\in \Theta(1) - \Theta(\delta^2), \\
J_2(\delta) - H_1^2(\delta) - H_2^2(\delta) &\in \Theta(\delta^2), \\
H_0(\delta) \cdot H_2(\delta) - H_1^2(\delta) &\in \Theta(\delta^2) - \Theta(\delta^4).
\end{aligned}
\tag{74}
$$

Because $\rho$ is a continuous function with $\rho(z) > 0$ for some $z > 0$. Therefore, $\rho \neq J_0(\delta) - H_0^2(\delta) - H_1^2(\delta)$ when $\delta \to 0^+$, otherwise $\rho(z) < 0$ for any $z > 0$. When $\delta \to 0^+$, both $J_2(\delta) - H_1^2(\delta) - H_2^2(\delta)$ and $H_0(\delta) \cdot H_2(\delta) - H_1^2(\delta)$ are in the order of $\delta^2$, which indicates that $\mu$ is a fixed constant when both $\delta$ and $\tilde{\delta}$ are close to 0. In addition, $J_2(\delta) - H_1^2(\delta) - H_2^2(\delta)$ goes to $+\infty$ while both $J_0(\delta) - H_0^2(\delta) - H_1^2(\delta)$ and $H_0(\delta) \cdot H_2(\delta) - H_1^2(\delta)$ go to $-\infty$ when $\delta \to +\infty$. Therefore, with a large enough $\delta$, we have

$$
\rho(\delta) \in \Theta(\delta^2) - \Theta(\delta^4) \quad \text{or} \quad \Theta(1) - \Theta(\delta^2),
\tag{75}
$$

which indicates that $\mu$ is a strictly decreasing function when $\delta$ and $\tilde{\delta}$ are large enough.

Next, we provide the conditions that guarantee $H_r(\delta) = \Theta(\delta^r)$ hold, and the relative proof for $J_r(\delta)$ can be derived accordingly following the similar steps as well. From Definition 3, we have

$$
\begin{aligned}
\lim_{\delta \to 0^+} \frac{H_r(\delta)}{\delta^r} &= \lim_{\delta \to 0^+} \int_{-\infty}^{+\infty} \phi'(\sigma_K z) \left(\frac{z}{\delta}\right)^r \frac{1}{\sqrt{2\pi}\delta} e^{-\frac{z^2}{\delta^2}} dz \\
&\overset{(a)}{=} \lim_{\delta \to 0^+} \int_{-\infty}^{+\infty} \phi'(\sigma_K \delta t) \frac{t^r}{\sqrt{2\pi}} e^{-t^2} dt \\
&= \lim_{\delta \to 0^+} \int_{-\infty}^{0^-} \phi'(\sigma_K \delta t) \frac{t^r}{\sqrt{2\pi}} e^{-t^2} dt + \lim_{\delta \to 0^+} \int_{0^+}^{+\infty} \phi'(\sigma_K \delta t) \frac{t^r}{\sqrt{2\pi}} e^{-t^2} dt \\
&= \phi'(0^-) \int_{-\infty}^{0^-} \frac{t^r}{\sqrt{2\pi}} e^{-t^2} dt + \phi'(0^+) \int_{0^+}^{+\infty} \frac{t^r}{\sqrt{2\pi}} e^{-t^2} dt,
\end{aligned}
\tag{76}
$$

where equality $(a)$ holds by letting $t = \frac{z}{\delta}$. It is easy to verify that

$$
\int_{0^+}^{+\infty} \frac{t^r}{\sqrt{2\pi}} e^{-t^2} dt = (-1)^r \int_{-\infty}^{0^-} \frac{t^r}{\sqrt{2\pi}} e^{-t^2} dt,
$$

and both are bounded for a fixed $r$. Thus, as long as either $\phi'(0^-)$ or $\phi'(0^+)$ is non-zero, we have $H_r(\delta) = \Theta(\delta^r)$ when $\delta \to 0^+$.

If $\phi$ has bounded gradient as $|\phi'| \leq C_\phi$ for some positive constant $C_\phi$. Then, we have

$$
\begin{aligned}
\left| \frac{H_r(\delta)}{\delta^r} \right| &= \left| \int_{-\infty}^{+\infty} \phi'(\sigma_K z) \left(\frac{z}{\delta}\right)^r \frac{1}{\sqrt{2\pi}\delta} e^{-\frac{z^2}{\delta^2}} dz \right| \\
&= \left| \int_{-\infty}^{+\infty} \phi'(\sigma_K \delta t) \frac{t^r}{\sqrt{2\pi}} e^{-t^2} dt \right| \\
&\leq C_\phi \cdot \left| \int_{-\infty}^{+\infty} \frac{t^r}{\sqrt{2\pi}} e^{-t^2} dt \right|
\end{aligned}
\tag{77}
$$

Therefore, we have $H_r(\delta) = \mathcal{O}(\delta^r)$ for all $\delta > 0$ when $\phi$ has bounded gradient.

Typiclly, for ReLU function, one can directly calculate that $H_r(\delta) = \delta^r$ for $\delta \in \mathbb{R}$, and $\rho(\delta) = C\delta^2$ when $\delta \leq 1$ for some constant $C = 0.091$. Then, it is easy to check that $\mu$ is a constant when $\delta, \tilde{\delta} \leq 1$. $\qquad \square$

# H    PROOF OF PRELIMINARY LEMMAS

## H.1    PROOF OF LEMMA 1

The eigenvalues of $\nabla^2 f(\cdot; p)$ at any fixed point $\boldsymbol{W}$ can be bounded in the form of (80) by Weyl's inequality (Lemma 4). Therefore, the primary technical challenge lies in bounding $\|\nabla^2 f(\boldsymbol{W}; p) - \nabla^2 f(\boldsymbol{W}^{[p]}; p)\|_2$, which is summarized in Lemma 12. Lemma 13 provides the exact calulation of the lower bound of $\mathbb{E}_{\boldsymbol{x}}\left(\sum_{j=1}^{K} \boldsymbol{\alpha}_j^T \boldsymbol{x} \phi'(\boldsymbol{w}_j^{[p]T} \boldsymbol{x})\right)^2$ when $\boldsymbol{x}$ belongs to Gaussian distribution with zero mean, which is used in proving the lower bound of the Hessian matrix in (81).

**Lemma 12.** *Let $f(\boldsymbol{W}; p)$ be the population risk function defined in* (17) *with $p$ and $\boldsymbol{W}$ satisfying* (20). *Then, we have*

$$\|\nabla^2 f(\boldsymbol{W}^{[p]}; p) - \nabla^2 f(\boldsymbol{W}; p)\|_2 \lesssim \frac{\lambda \delta^2 + (1-\lambda)\tilde{\delta}^2}{K} \cdot \frac{\|\boldsymbol{W}^{[p]} - \boldsymbol{W}\|_2}{\sigma_K}. \tag{78}$$

**Lemma 13** (Lemma D.6, (Zhong et al., 2017)). *For any $\{\boldsymbol{w}_j\}_{j=1}^{K} \in \mathbb{R}^d$, let $\boldsymbol{\alpha} \in \mathbb{R}^{dK}$ be the unit vector defined in* (19). *When the $\phi$ is ReLU function, we have*

$$\min_{\|\boldsymbol{\alpha}\|_2=1} \mathbb{E}_{\boldsymbol{x}\sim\mathcal{N}(0,\sigma^2)}\left(\sum_{j=1}^{K} \boldsymbol{\alpha}_j^T \boldsymbol{x} \phi'(\boldsymbol{w}_j^T \boldsymbol{x})\right)^2 \gtrsim \rho(\sigma), \tag{79}$$

*where $\rho(\sigma)$ is defined in Definition 3.*

*Proof of Lemma 1.* Let $\lambda_{\max}(\boldsymbol{W})$ and $\lambda_{\min}(\boldsymbol{W})$ denote the largest and smallest eigenvalues of $\nabla^2 f(\boldsymbol{W}; p)$ at point $\boldsymbol{W}$, respectively. Then, from Lemma 4, we have

$$\begin{aligned}
\lambda_{\max}(\boldsymbol{W}) &\le \lambda_{\max}(\boldsymbol{W}^{[p]}) + \|\nabla^2 f(\boldsymbol{W}; p) - \nabla^2 f(\boldsymbol{W}^{[p]}; p)\|_2, \\
\lambda_{\min}(\boldsymbol{W}) &\ge \lambda_{\min}(\boldsymbol{W}^{[p]}) - \|\nabla^2 f(\boldsymbol{W}; p) - \nabla^2 f(\boldsymbol{W}^{[p]}; p)\|_2.
\end{aligned} \tag{80}$$

Then, we provide the lower bound of the Hessian matrix of the population function at $\boldsymbol{W}^{[p]}$. For any $\boldsymbol{\alpha} \in \mathbb{R}^{dK}$ defined in (19) with $\|\boldsymbol{\alpha}\|_2 = 1$, we have

$$\begin{aligned}
&\min_{\|\boldsymbol{\alpha}\|_2=1} \boldsymbol{\alpha}^T \nabla^2 f(\boldsymbol{W}^{[p]}; p)\boldsymbol{\alpha} \\
=&\frac{1}{K^2} \min_{\|\boldsymbol{\alpha}\|_2=1} \left[\lambda \mathbb{E}_{\boldsymbol{x}}\left(\sum_{j=1}^{K} \boldsymbol{\alpha}_j^T \boldsymbol{x} \phi'(\boldsymbol{w}_j^{[p]T} \boldsymbol{x})\right)^2 + \widetilde{\lambda} \mathbb{E}_{\widetilde{\boldsymbol{x}}}\left(\sum_{j=1}^{K} \boldsymbol{\alpha}_j^T \widetilde{\boldsymbol{x}} \phi'(\boldsymbol{w}_j^{[p]T} \widetilde{\boldsymbol{x}})\right)^2\right] \\
\ge&\frac{1}{K^2} \min_{\|\boldsymbol{\alpha}\|_2=1} \lambda \mathbb{E}_{\boldsymbol{x}}\left(\sum_{j=1}^{K} \boldsymbol{\alpha}_j^T \boldsymbol{x} \phi'(\boldsymbol{w}_j^{[p]T} \boldsymbol{x})\right)^2 + \min_{\|\boldsymbol{\alpha}\|_2=1} \widetilde{\lambda} \mathbb{E}_{\widetilde{\boldsymbol{x}}}\left(\sum_{j=1}^{K} \boldsymbol{\alpha}_j^T \widetilde{\boldsymbol{x}} \phi'(\boldsymbol{w}_j^{[p]T} \widetilde{\boldsymbol{x}})\right)^2 \\
\ge&\frac{\lambda \rho(\delta) + \widetilde{\lambda} \rho(\tilde{\delta})}{11\kappa^2\gamma K^2},
\end{aligned} \tag{81}$$

where the last inequality comes from Lemma 13.

Next, the upper bound can be bounded as

$$\begin{aligned}
&\max_{\|\boldsymbol{\alpha}\|_2=1} \boldsymbol{\alpha}^T \nabla^2 f(\boldsymbol{W}^{[p]}; p)\boldsymbol{\alpha} \\
=&\frac{1}{K^2} \max_{\|\boldsymbol{\alpha}\|_2=1} \left[\lambda \mathbb{E}_{\boldsymbol{x}}\left(\sum_{j=1}^{K} \boldsymbol{\alpha}_j^T \boldsymbol{x} \phi'(\boldsymbol{w}_j^{[p]T} \boldsymbol{x})\right)^2 + \widetilde{\lambda} \mathbb{E}_{\widetilde{\boldsymbol{x}}}\left(\sum_{j=1}^{K} \boldsymbol{\alpha}_j^T \widetilde{\boldsymbol{x}} \phi'(\boldsymbol{w}_j^{[p]T} \widetilde{\boldsymbol{x}})\right)^2\right] \\
\le&\frac{1}{K^2} \max_{\|\boldsymbol{\alpha}\|_2=1} \lambda \mathbb{E}_{\boldsymbol{x}}\left(\sum_{j=1}^{K} \boldsymbol{\alpha}_j^T \boldsymbol{x} \phi'(\boldsymbol{w}_j^{[p]T} \boldsymbol{x})\right)^2 + \max_{\|\boldsymbol{\alpha}\|_2=1} \widetilde{\lambda} \mathbb{E}_{\widetilde{\boldsymbol{x}}}\left(\sum_{j=1}^{K} \boldsymbol{\alpha}_j^T \widetilde{\boldsymbol{x}} \phi'(\boldsymbol{w}_j^{[p]T} \widetilde{\boldsymbol{x}})\right)^2.
\end{aligned} \tag{82}$$

For $\mathbb{E}_{\boldsymbol{x}}\Big(\sum_{j=1}^{K}\boldsymbol{\alpha}_j^T\boldsymbol{x}\phi'(\boldsymbol{w}_j^{[p]T}\boldsymbol{x})\Big)^2$, we have

$$
\begin{aligned}
&\mathbb{E}_{\boldsymbol{x}}\Big(\sum_{j=1}^{K}\boldsymbol{\alpha}_j^T\boldsymbol{x}\phi'(\boldsymbol{w}_j^{[p]T}\boldsymbol{x})\Big)^2 \\
=&\mathbb{E}_{\boldsymbol{x}}\sum_{j_1=1}^{K}\sum_{j_2=1}^{K}\boldsymbol{\alpha}_{j_1}^T\boldsymbol{x}\phi'(\boldsymbol{w}_{j_1}^{[p]T}\boldsymbol{x})\boldsymbol{\alpha}_{j_2}^T\boldsymbol{x}\phi'(\boldsymbol{w}_{j_2}^{[p]T}\boldsymbol{x}) \\
=&\sum_{j_1=1}^{K}\sum_{j_2=1}^{K}\mathbb{E}_{\boldsymbol{x}}\boldsymbol{\alpha}_{j_1}^T\boldsymbol{x}\phi'(\boldsymbol{w}_{j_1}^{[p]T}\boldsymbol{x})\boldsymbol{\alpha}_{j_2}^T\boldsymbol{x}\phi'(\boldsymbol{w}_{j_2}^{[p]T}\boldsymbol{x}) \\
\leq&\sum_{j_1=1}^{K}\sum_{j_2=1}^{K}\Big[\mathbb{E}_{\boldsymbol{x}}(\boldsymbol{\alpha}_{j_1}^T\boldsymbol{x})^4\mathbb{E}_{\boldsymbol{x}}(\phi'(\boldsymbol{w}_{j_1}^{[p]T}\boldsymbol{x}))^4\mathbb{E}_{\boldsymbol{x}}(\boldsymbol{\alpha}_{j_2}^T\boldsymbol{x})^4\mathbb{E}_{\boldsymbol{x}}(\phi'(\boldsymbol{w}_{j_2}^{[p]T}\boldsymbol{x}))^4\Big]^{1/4} \\
\leq&\sum_{j_1=1}^{K}\sum_{j_2=1}^{K}3\delta^2\|\boldsymbol{\alpha}_{j_1}\|_2\|\boldsymbol{\alpha}_{j_2}\|_2 \\
\leq&6\delta^2\sum_{j_1=1}^{K}\sum_{j_2=1}^{K}\frac{1}{2}(\|\boldsymbol{\alpha}_{j_1}\|_2^2+\|\boldsymbol{\alpha}_{j_2}\|_2^2) \\
=&6K\delta^2
\end{aligned}
\tag{83}
$$

Therefore, we have

$$
\begin{aligned}
&\max_{\|\boldsymbol{\alpha}\|_2=1}\boldsymbol{\alpha}^T\nabla^2 f(\boldsymbol{W}^{[p]};p)\boldsymbol{\alpha} \\
\leq&\frac{1}{K^2}\max_{\|\boldsymbol{\alpha}\|_2=1}\lambda\mathbb{E}_{\boldsymbol{x}}\Big(\sum_{j=1}^{K}\boldsymbol{\alpha}_j^T\boldsymbol{x}\phi'(\boldsymbol{w}_j^{[p]T}\boldsymbol{x})\Big)^2+\max_{\|\boldsymbol{\alpha}\|_2=1}\widetilde{\lambda}\mathbb{E}_{\widetilde{\boldsymbol{x}}}\Big(\sum_{j=1}^{K}\boldsymbol{\alpha}_j^T\widetilde{\boldsymbol{x}}\phi'(\boldsymbol{w}_j^{[p]T}\widetilde{\boldsymbol{x}})\Big)^2 \\
\leq&\frac{6(\lambda\delta^2+\widetilde{\lambda}\tilde{\delta}^2)}{K}.
\end{aligned}
\tag{84}
$$

Then, given (20), we have

$$
\|\boldsymbol{W}^{(0,0)}-\boldsymbol{W}^{[p]}\|_F=p\|\boldsymbol{W}^{(0,0)}-\boldsymbol{W}^*\|_F\lesssim\frac{\sigma_K}{\mu^2 K}.
\tag{85}
$$

Combining (85) and Lemma 12, we have

$$
\|\nabla^2 f(\boldsymbol{W};p)-\nabla^2 f(\boldsymbol{W}^{[p]};p)\|_2\lesssim\frac{\lambda\rho(\delta)+\widetilde{\lambda}\rho(\tilde{\delta})}{132\kappa^2\gamma K^2}.
\tag{86}
$$

Therefore, (86) and (80) completes the whole proof. $\qquad\square$

## H.2 PROOF OF LEMMA 2

The task of bounding of the quantity between $\|\nabla\hat{f}-\nabla f\|_2$ is dividing into bounding $I_1$, $I_2$, $I_3$ and $I_4$ as shown in (89). $I_1$ and $I_3$ represent the deviation of the mean of several random variables to their expectation, which can be bounded through concentration inequality, i.e, Chernoff bound. $I_2$ and $I_4$ come from the inconsistency of the output label $y$ and pseudo label $\widetilde{y}$ in the empirical risk function in (1) and population risk function in (17). The major challenge lies in characterizing the upper bound of $\boldsymbol{I}_2$ and $\boldsymbol{I}_4$ as the linear function of $\widetilde{\boldsymbol{W}}-\boldsymbol{W}^{[p]}$ and $\boldsymbol{W}^{[p]}-\boldsymbol{W}^*$, which is summarized in (96).

*Proof of Lemma 2.* From (1), we know that

$$\frac{\partial \hat{f}}{\partial \boldsymbol{w}_k}(\boldsymbol{W}) = \frac{\lambda}{N} \sum_{n=1}^{N} \Big( \frac{1}{K} \sum_{j=1}^{K} \phi(\boldsymbol{w}_j^T \boldsymbol{x}_n) - y_n \Big) \boldsymbol{x}_n + \frac{1-\lambda}{M} \sum_{m=1}^{M} \Big( \frac{1}{K} \sum_{j=1}^{K} \phi(\boldsymbol{w}_j^T \widetilde{\boldsymbol{x}}_m) - \widetilde{y}_n \Big) \widetilde{\boldsymbol{x}}_m$$

$$= \frac{\lambda}{K^2 N} \sum_{n=1}^{N} \sum_{j=1}^{K} \Big( \phi(\boldsymbol{w}_j^T \boldsymbol{x}_n) - \phi(\boldsymbol{w}_j^{*T} \boldsymbol{x}_n) \Big) \boldsymbol{x}_n$$

$$+ \frac{1-\lambda}{K^2 M} \sum_{m=1}^{M} \sum_{j=1}^{K} \Big( \phi(\boldsymbol{w}_j^T \widetilde{\boldsymbol{x}}_m) - \phi(\widetilde{\boldsymbol{w}}_j^T \widetilde{\boldsymbol{x}}_m) \Big) \widetilde{\boldsymbol{x}}_m. \tag{87}$$

From (32), we know that

$$\frac{\partial \hat{f}}{\partial \boldsymbol{w}_k}(\boldsymbol{W}) = \frac{\lambda}{K^2} \mathbb{E}_{\boldsymbol{x}} \sum_{j=1}^{K} \Big( \phi(\boldsymbol{w}_j^T \boldsymbol{x}) - \phi(\boldsymbol{w}_j^{*T} \boldsymbol{x}) \Big) \boldsymbol{x} + \frac{1-\lambda}{K^2} \mathbb{E}_{\widetilde{\boldsymbol{x}}} \sum_{j=1}^{K} \Big( \phi(\boldsymbol{w}_j^T \widetilde{\boldsymbol{x}}) - \phi(\widetilde{\boldsymbol{w}}_j^T \widetilde{\boldsymbol{x}}) \Big) \widetilde{\boldsymbol{x}}. \tag{88}$$

Then, from (17), we have

$$\frac{\partial \hat{f}}{\partial \boldsymbol{w}_k}(\boldsymbol{W}) - \frac{\partial f}{\partial \boldsymbol{w}_k}(\boldsymbol{W}; p)$$

$$= \frac{\lambda}{K^2 N} \sum_{j=1}^{K} \Big[ \sum_{n=1}^{N} (\phi(\boldsymbol{w}_j^T \boldsymbol{x}_n) - \phi(\boldsymbol{w}_j^{*T} \boldsymbol{x}_n)) \boldsymbol{x}_n - \mathbb{E}_{\boldsymbol{x}} (\phi(\boldsymbol{w}_j^T \boldsymbol{x}) - \phi(\boldsymbol{w}_j^{[p]T} \boldsymbol{x})) \boldsymbol{x} \Big]$$

$$+ \frac{1-\lambda}{K^2 M} \sum_{j=1}^{K} \Big[ \sum_{m=1}^{M} (\phi(\boldsymbol{w}_j^T \widetilde{\boldsymbol{x}}_m) - \phi(\widetilde{\boldsymbol{w}}_j^T \widetilde{\boldsymbol{x}}_m)) \widetilde{\boldsymbol{x}}_m - \mathbb{E}_{\widetilde{\boldsymbol{x}}} (\phi(\boldsymbol{w}_j^T \widetilde{\boldsymbol{x}}) - \phi(\boldsymbol{w}_j^{[p]T} \widetilde{\boldsymbol{x}})) \widetilde{\boldsymbol{x}} \Big]$$

$$= \frac{\lambda}{K^2} \sum_{j=1}^{K} \Big[ \frac{1}{N} \sum_{n=1}^{N} (\phi(\boldsymbol{w}_j^T \boldsymbol{x}_n) - \phi(\boldsymbol{w}_j^{*T} \boldsymbol{x}_n)) \boldsymbol{x}_n - \mathbb{E}_{\boldsymbol{x}} (\phi(\boldsymbol{w}_j^T \boldsymbol{x}) - \phi(\boldsymbol{w}_j^{*T} \boldsymbol{x})) \boldsymbol{x} \Big] \tag{89}$$

$$+ \frac{\lambda}{K^2} \sum_{j=1}^{K} \mathbb{E}_{\boldsymbol{x}} \Big[ (\phi(\boldsymbol{w}_j^{*T} \boldsymbol{x}) - \phi(\boldsymbol{w}_j^{[p]T} \boldsymbol{x})) \boldsymbol{x} \Big]$$

$$+ \frac{1-\lambda}{K^2} \sum_{j=1}^{K} \Big[ \frac{1}{M} \sum_{m=1}^{M} (\phi(\boldsymbol{w}_j^T \widetilde{\boldsymbol{x}}_m) - \phi(\widetilde{\boldsymbol{w}}_j^T \widetilde{\boldsymbol{x}}_m)) \widetilde{\boldsymbol{x}}_m - \mathbb{E}_{\widetilde{\boldsymbol{x}}} (\phi(\boldsymbol{w}_j^T \widetilde{\boldsymbol{x}}) - \phi(\widetilde{\boldsymbol{w}}_j^T \widetilde{\boldsymbol{x}})) \widetilde{\boldsymbol{x}} \Big]$$

$$+ \frac{1-\lambda}{K^2} \sum_{j=1}^{K} \mathbb{E}_{\widetilde{\boldsymbol{x}}} \Big[ (\phi(\widetilde{\boldsymbol{w}}_j^T \widetilde{\boldsymbol{x}}) - \phi(\boldsymbol{w}_j^{[p]T}(p)\widetilde{\boldsymbol{x}})) \widetilde{\boldsymbol{x}} \Big]$$

$$:= \boldsymbol{I}_1 + \boldsymbol{I}_2 + \boldsymbol{I}_3 + \boldsymbol{I}_4.$$

For any $\boldsymbol{\alpha}_j \in \mathbb{R}^d$ with $\|\boldsymbol{\alpha}_j\|_2 \leq 1$, we define a random variable $Z(j) = (\phi(\boldsymbol{w}_j^T \boldsymbol{x}) - \phi(\boldsymbol{w}_j^{*T} \boldsymbol{x})) \boldsymbol{\alpha}_j^T \boldsymbol{x}$ and $Z_n(j) = (\phi(\boldsymbol{w}_j^T \boldsymbol{x}_n) - \phi(\boldsymbol{w}_j^{*T} \boldsymbol{x}_n)) \boldsymbol{\alpha}_j^T \boldsymbol{x}_n$ as the realization of $Z(j)$ for $n = 1, 2 \cdots, N$. Then, for any $p \in \mathbb{N}^+$, we have

$$(\mathbb{E}|Z|^p)^{1/p} = \Big( \mathbb{E}|\phi(\boldsymbol{w}_j^T \boldsymbol{x}) - \phi(\boldsymbol{w}_j^{*T} \boldsymbol{x})|^p \cdot |\boldsymbol{\alpha}_j^T \boldsymbol{x}|^p \Big)^{1/p}$$

$$\leq \Big( \mathbb{E}|(\boldsymbol{w}_j - \boldsymbol{w}_j^*)^T \boldsymbol{x}|^p \cdot |\boldsymbol{\alpha}_j^T \boldsymbol{x}|^p \Big)^{1/p} \tag{90}$$

$$\leq C \cdot \delta^2 \|\boldsymbol{w}_j - \boldsymbol{w}_j^*\|_2 \cdot p,$$

where $C$ is a positive constant and the last inequality holds since $\boldsymbol{x} \sim \mathcal{N}(0, \delta^2)$. From Definition 2, we know that $Z$ belongs to sub-exponential distribution with $\|Z\|_{\psi_1} \lesssim \delta^2 \|\boldsymbol{w}_j - \boldsymbol{w}_j^*\|_2$. Therefore, by Chernoff inequality, we have

$$\mathbb{P} \Big\{ \Big| \frac{1}{N} \sum_{n=1}^{N} Z_n(j) - \mathbb{E}Z(j) \Big| < t \Big\} \leq 1 - \frac{e^{-C(\delta^2 \|\boldsymbol{w}_j - \boldsymbol{w}_j^*\|_2)^2 \cdot Ns^2}}{e^{Nst}} \tag{91}$$

for some positive constant $C$ and any $s \in \mathbb{R}$.

Let $t = \delta^2 \|\boldsymbol{w}_j - \boldsymbol{w}_j^*\|_2 \sqrt{\frac{d \log q}{N}}$ and $s = \frac{2}{C\delta^2 \|\boldsymbol{w}_j - \boldsymbol{w}_j^*\|_2} \cdot t$ for some large constant $q > 0$, we have

$$\left| \frac{1}{N} \sum_{n=1}^{N} Z_n(j) - \mathbb{E}Z(j) \right| \lesssim \delta^2 \|\boldsymbol{w}_j - \boldsymbol{w}_j^*\|_2 \cdot \sqrt{\frac{d \log q}{N}} \tag{92}$$

with probability at least $1 - q^{-d}$. From Lemma 7, we have

$$\left\| \frac{1}{N} \sum_{n=1}^{N} \left( \phi(\boldsymbol{w}_j^T \boldsymbol{x}_n) - \phi(\boldsymbol{w}_j^{*T} \boldsymbol{x}_n) \right) \boldsymbol{x}_n - \mathbb{E}_{\boldsymbol{x}} \left( \phi(\boldsymbol{w}_j^T \boldsymbol{x}) - \phi(\boldsymbol{w}_j^{*T} \boldsymbol{x}) \right) \boldsymbol{x} \right\|_2$$
$$\leq 2\delta^2 \|\boldsymbol{w}_j - \boldsymbol{w}_j^*\|_2 \cdot \sqrt{\frac{d \log q}{N}} \tag{93}$$

with probability at least $1 - (q/5)^{-d}$. Since $q$ is a large constant, we release the probability as $1 - q^{-d}$ for simplification. Similar to $Z$, we have

$$\left\| \frac{1}{M} \sum_{m=1}^{M} \left( \phi(\boldsymbol{w}_j^T \widetilde{\boldsymbol{x}}_m) - \phi(\widetilde{\boldsymbol{w}}_j^T \widetilde{\boldsymbol{x}}_m) \right) \widetilde{\boldsymbol{x}}_m - \mathbb{E}_{\widetilde{\boldsymbol{x}}} \left( \phi(\boldsymbol{w}_j^T \widetilde{\boldsymbol{x}}) - \phi(\widetilde{\boldsymbol{w}}_j^T \widetilde{\boldsymbol{x}}) \right) \widetilde{\boldsymbol{x}} \right\|_2$$
$$\lesssim \widetilde{\delta}^2 \|\boldsymbol{w}_j - \widetilde{\boldsymbol{w}}_j\|_2 \cdot \sqrt{\frac{d \log q}{M}} \tag{94}$$

with probability at least $1 - q^{-d}$.

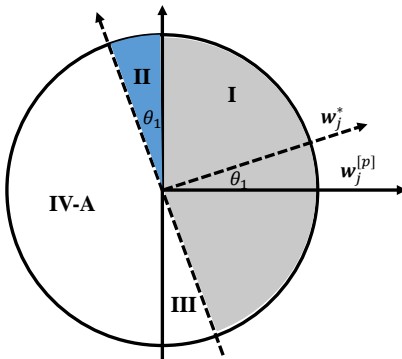

Figure 13: The subspace spanned by $\boldsymbol{w}_j^*$ and $\boldsymbol{w}_j^{[p]}$

For term $\mathbb{E}_{\boldsymbol{x}} \left[ \left( \phi(\boldsymbol{w}_j^{*T} \boldsymbol{x}) - \phi(\boldsymbol{w}_j^{[p]T} \boldsymbol{x}) \right) \boldsymbol{x} \right]$, let us define the angle between $\boldsymbol{w}_j^*$ and $\boldsymbol{w}_j^{[p]}$ as $\theta_1$. Figure 13 shows the subspace spanned by the vector $\boldsymbol{w}_j^*$ and $\widetilde{\boldsymbol{w}}_j$. We divide the subspace by 4 pieces, where the gray region denotes area I, and the blue area denotes area II. Areas III and IV are the symmetries of II and I from the origin, respectively. Hence, we have

$$\mathbb{E}_{\boldsymbol{x}} \left[ \left( \phi(\boldsymbol{w}_j^{*T} \boldsymbol{x}) - \phi(\boldsymbol{w}_j^{[p]T} \boldsymbol{x}) \right) \boldsymbol{x} \right]$$
$$= \mathbb{E}_{\boldsymbol{x} \in \text{ area I}} \left[ \left( \phi(\boldsymbol{w}_j^{*T} \boldsymbol{x}) - \phi(\boldsymbol{w}_j^{[p]T} \boldsymbol{x}) \right) \boldsymbol{x} \right] + \mathbb{E}_{\boldsymbol{x} \in \text{ area II}} \left[ \left( \phi(\boldsymbol{w}_j^{*T} \boldsymbol{x}) - \phi(\boldsymbol{w}_j^{[p]T} \boldsymbol{x}) \right) \boldsymbol{x} \right]$$
$$+ \mathbb{E}_{\boldsymbol{x} \in \text{ area III}} \left[ \left( \phi(\boldsymbol{w}_j^{*T} \boldsymbol{x}) - \phi(\boldsymbol{w}_j^{[p]T} \boldsymbol{x}) \right) \boldsymbol{x} \right] + \mathbb{E}_{\boldsymbol{x} \in \text{ area IV}} \left[ \left( \phi(\boldsymbol{w}_j^{*T} \boldsymbol{x}) - \phi(\boldsymbol{w}_j^{[p]T} \boldsymbol{x}) \right) \boldsymbol{x} \right]$$
$$= \mathbb{E}_{\boldsymbol{x} \in \text{ area I}} \left[ \left( \phi(\boldsymbol{w}_j^{*T} \boldsymbol{x}) - \phi(\boldsymbol{w}_j^{[p]T} \boldsymbol{x}) \right) \boldsymbol{x} \right] + \mathbb{E}_{\boldsymbol{x} \in \text{ area II}} \left[ \boldsymbol{w}_j^{*T} \widetilde{\boldsymbol{x}} \widetilde{\boldsymbol{x}} \right] - \mathbb{E}_{\boldsymbol{x} \in \text{ area III}} \left[ \boldsymbol{w}_j^{[p]T} \boldsymbol{x} \boldsymbol{x} \right] \tag{95}$$
$$= \mathbb{E}_{\boldsymbol{x} \in \text{ area I}} \left[ (\boldsymbol{w}_j^* - \boldsymbol{w}_j^{[p]})^T \boldsymbol{x} \boldsymbol{x} \right] + \mathbb{E}_{\boldsymbol{x} \in \text{ area II}} \left[ (\boldsymbol{w}_j^* - \boldsymbol{w}_j^{[p]})^T \boldsymbol{x} \boldsymbol{x} \right]$$
$$= \frac{1}{2} \mathbb{E}_{\boldsymbol{x}} \left[ (\boldsymbol{w}_j^* - \boldsymbol{w}_j^{[p]})^T \boldsymbol{x} \boldsymbol{x} \right]$$

Therefore, we have

$$
\begin{aligned}
&\left\| \frac{\lambda}{K^2}\mathbb{E}_{\boldsymbol{x}}\Big[ \big(\phi(\boldsymbol{w}_j^{*T}\boldsymbol{x}) - \phi(\boldsymbol{w}_j^{[p]T}\boldsymbol{x})\big)\boldsymbol{x} \Big] + \frac{1-\lambda}{K^2}\mathbb{E}_{\widetilde{\boldsymbol{x}}}\Big[ \big(\phi(\widetilde{\boldsymbol{w}}_j^T\widetilde{\boldsymbol{x}}) - \phi(\boldsymbol{w}_j^{[p]T}\widetilde{\boldsymbol{x}})\big)\widetilde{\boldsymbol{x}} \Big] \right\|_2 \\
&= \left\| \frac{\lambda}{2K^2}\mathbb{E}_{\boldsymbol{x}}\big[ (\boldsymbol{w}_j^* - \boldsymbol{w}_j^{[p]})^T\boldsymbol{x}\boldsymbol{x} \big] + \frac{1-\lambda}{2K^2}\mathbb{E}_{\widetilde{\boldsymbol{x}}}\big[ (\widetilde{\boldsymbol{w}}_j - \boldsymbol{w}_j^{[p]})^T\boldsymbol{x}\boldsymbol{x} \big] \right\|_2 \qquad (96)\\
&= \frac{\big\| \lambda\delta^2\cdot\big(\widetilde{\boldsymbol{w}}_j - \boldsymbol{w}_j^{[p]}\big) + (1-\lambda)\tilde{\delta}^2\cdot\big(\boldsymbol{w}_j^* - \boldsymbol{w}_j^{[p]}\big) \big\|_2}{2K^2}.
\end{aligned}
$$

From (93), (94) and (96), we have

$$
\begin{aligned}
&\left\| \frac{\partial\hat{f}}{\partial\boldsymbol{w}_k}(\boldsymbol{W};p) - \frac{\partial f}{\partial\boldsymbol{w}_k}(\boldsymbol{W}) \right\|_2 \\
&\leq \frac{\lambda}{K^2}\sum_{j=1}^K \left\| \frac{1}{N}\sum_{n=1}^N \big(\phi(\boldsymbol{w}_j^T\boldsymbol{x}_n) - \phi(\boldsymbol{w}_j^{*T}\boldsymbol{x}_n)\big)\boldsymbol{x}_n - \mathbb{E}_{\boldsymbol{x}}\big(\phi(\boldsymbol{w}_j^T\boldsymbol{x}) - \phi(\boldsymbol{w}_j^{*T}\boldsymbol{x})\big)\boldsymbol{x} \right\|_2 \\
&\quad + \frac{1-\lambda}{K^2}\sum_{j=1}^K \left\| \frac{1}{M}\sum_{m=1}^M \big(\phi(\boldsymbol{w}_j^T\widetilde{\boldsymbol{x}}_m) - \phi(\widetilde{\boldsymbol{w}}_j^T\widetilde{\boldsymbol{x}}_m)\big)\widetilde{\boldsymbol{x}}_m - \mathbb{E}_{\widetilde{\boldsymbol{x}}}\big(\phi(\boldsymbol{w}_j^T\widetilde{\boldsymbol{x}}) - \phi(\widetilde{\boldsymbol{w}}_j^T\widetilde{\boldsymbol{x}})\big)\widetilde{\boldsymbol{x}} \right\|_2 \\
&\quad + \sum_{j=1}^K \left\| \frac{\lambda}{K^2}\mathbb{E}_{\boldsymbol{x}}\Big[ \big(\phi(\boldsymbol{w}_j^{*T}\boldsymbol{x}) - \phi(\boldsymbol{w}_j^{[p]T}\boldsymbol{x})\big)\boldsymbol{x} \Big] + \frac{1-\lambda}{K^2}\mathbb{E}_{\widetilde{\boldsymbol{x}}}\Big[ \big(\phi(\widetilde{\boldsymbol{w}}_j^T\widetilde{\boldsymbol{x}}) - \phi(\boldsymbol{w}_j^{[p]T}\widetilde{\boldsymbol{x}})\big)\widetilde{\boldsymbol{x}} \Big] \right\|_2 \quad (97)\\
&\leq \frac{\lambda}{K^2}\delta^2\sqrt{\frac{d\log q}{N}}\cdot\sum_{j=1}^K \|\boldsymbol{w}_j - \boldsymbol{w}_j^*\|_2 + \frac{1-\lambda}{K^2}\cdot\tilde{\delta}^2\sqrt{\frac{d\log q}{M}}\cdot\sum_{j=1}^K \|\boldsymbol{w}_j - \widetilde{\boldsymbol{w}}_j\|_2 \\
&\quad + \frac{1}{2K^2}\cdot\sum_{j=1}^K \big\| \lambda\delta^2\cdot\big(\widetilde{\boldsymbol{w}}_j - \boldsymbol{w}_j^{[p]}\big) + \widetilde{\lambda}\tilde{\delta}^2\cdot\big(\boldsymbol{w}_j^* - \boldsymbol{w}_j^{[p]}\big) \big\|_2 \\
&\leq \frac{\lambda}{K^{3/2}}\delta^2\sqrt{\frac{d\log q}{N}}\cdot\|\boldsymbol{W} - \boldsymbol{W}^*\|_2 + \frac{1-\lambda}{K^{3/2}}\cdot\tilde{\delta}^2\sqrt{\frac{d\log q}{M}}\cdot\|\boldsymbol{W} - \widetilde{\boldsymbol{W}}\|_2 \\
&\quad + \frac{1}{2K^{3/2}}\big\| \lambda\delta^2\cdot\big(\widetilde{\boldsymbol{W}} - \boldsymbol{W}^{[p]}\big) + (1-\lambda)\tilde{\delta}^2\cdot\big(\boldsymbol{W}^* - \boldsymbol{W}^{[p]}\big) \big\|_2
\end{aligned}
$$

with probability at least $1 - q^{-d}$.

In conclusion, let $\boldsymbol{\alpha}\in\mathbb{R}^{Kd}$ and $\boldsymbol{\alpha}_j\in\mathbb{R}^d$ with $\boldsymbol{\alpha} = [\boldsymbol{\alpha}_1^T, \boldsymbol{\alpha}_2^T, \cdots, \boldsymbol{\alpha}_K^T]^T$, we have

$$
\begin{aligned}
\|\nabla f(\boldsymbol{W}) - \nabla\hat{f}(\boldsymbol{W})\|_2 &= \left| \boldsymbol{\alpha}^T\big(\nabla f(\boldsymbol{W}) - \nabla\hat{f}(\boldsymbol{W})\big) \right| \\
&\leq \sum_{k=1}^K \left| \boldsymbol{\alpha}_k^T\big(\frac{\partial\hat{f}}{\partial\boldsymbol{w}_k}(\boldsymbol{W}) - \frac{\partial f}{\partial\boldsymbol{w}_k}(\boldsymbol{W})\big) \right| \\
&\lesssim \sum_{k=1}^K \left\| \frac{\partial\hat{f}}{\partial\boldsymbol{w}_k}(\boldsymbol{W}) - \frac{\partial f}{\partial\boldsymbol{w}_k}(\boldsymbol{W}) \right\|_2 \cdot \|\boldsymbol{\alpha}_k\|_2 \\
&\lesssim \frac{\lambda}{K}\delta^2\sqrt{\frac{d\log q}{N}}\cdot\|\boldsymbol{W} - \boldsymbol{W}^*\|_2 + \frac{1-\lambda}{K}\cdot\tilde{\delta}^2\sqrt{\frac{d\log q}{M}}\cdot\|\boldsymbol{W} - \widetilde{\boldsymbol{W}}\|_2 \\
&\quad + \frac{1}{2K}\big\| \lambda\delta^2\cdot\big(\widetilde{\boldsymbol{W}} - \boldsymbol{W}^{[p]}\big) + (1-\lambda)\tilde{\delta}^2\cdot\big(\boldsymbol{W}^* - \boldsymbol{W}^{[p]}\big) \big\|_2
\end{aligned}
$$
$$(98)$$

with probability at least $1 - q^{-d}$. $\qquad\square$

### H.3 PROOF OF LEMMA 12

The distance of the second order derivatives of the population risk function $f(\cdot;p)$ at point $\boldsymbol{W}$ and $\boldsymbol{W}^{[p]}$ can be converted into bounding $\boldsymbol{P}_1, \boldsymbol{P}_2, \boldsymbol{P}_3$ and $\boldsymbol{P}_4$, which are defined in (101). The major

idea in proving $\boldsymbol{P}_1$ is to connect the error bound to the angle between $\boldsymbol{W}$ and $\boldsymbol{W}^{[p]}$. Similar ideas apply in bounding the other three items as well.

*Proof of Lemma 12.* From (17), we have

$$\frac{\partial^2 f}{\partial \boldsymbol{w}_{j_1} \partial \boldsymbol{w}_{j_2}}(\boldsymbol{W}^{[p]}; p) = \frac{\lambda}{K^2} \mathbb{E}_{\boldsymbol{x}} \phi'(\boldsymbol{w}_{j_1}^{[p]T} \boldsymbol{x}) \phi'(\boldsymbol{w}_{j_2}^{[p]T} \boldsymbol{x}) \boldsymbol{x} \boldsymbol{x}^T + \frac{1-\lambda}{K^2} \mathbb{E}_{\widetilde{\boldsymbol{x}}} \phi'(\boldsymbol{w}_{j_1}^{[p]T} \widetilde{\boldsymbol{x}}) \phi'(\boldsymbol{w}_{j_2}^{[p]T} \widetilde{\boldsymbol{x}}) \widetilde{\boldsymbol{x}} \widetilde{\boldsymbol{x}}^T,$$
(99)

and $\quad \dfrac{\partial^2 f}{\partial \boldsymbol{w}_{j_1} \partial \boldsymbol{w}_{j_2}}(\boldsymbol{W}; p) = \dfrac{\lambda}{K^2} \mathbb{E}_{\boldsymbol{x}} \phi'(\boldsymbol{w}_{j_1}^T \boldsymbol{x}) \phi'(\boldsymbol{w}_{j_2}^T \boldsymbol{x}) \boldsymbol{x} \boldsymbol{x}^T + \dfrac{1-\lambda}{K^2} \mathbb{E}_{\widetilde{\boldsymbol{x}}} \phi'(\boldsymbol{w}_{j_1}^T \widetilde{\boldsymbol{x}}) \phi'(\boldsymbol{w}_{j_2}^T \widetilde{\boldsymbol{x}}) \widetilde{\boldsymbol{x}} \widetilde{\boldsymbol{x}}^T,$
(100)

where $\boldsymbol{w}_j^{[p]}$ is the $j$-th column of $\boldsymbol{W}^{[p]}$. Then, we have

$$\frac{\partial^2 f}{\partial \boldsymbol{w}_{j_1} \partial \boldsymbol{w}_{j_2}}(\boldsymbol{W}^*) - \frac{\partial^2 f}{\partial \boldsymbol{w}_{j_1} \partial \boldsymbol{w}_{j_2}}(\boldsymbol{W})$$

$$= \frac{\lambda}{K^2} \mathbb{E}_{\boldsymbol{x}} \Big[ \phi'(\boldsymbol{w}_{j_1}^{[p]T} \boldsymbol{x}) \phi'(\boldsymbol{w}_{j_2}^{[p]T} \boldsymbol{x}) - \phi'(\boldsymbol{w}_{j_1}^T \boldsymbol{x}) \phi'(\boldsymbol{w}_{j_2}^T \boldsymbol{x}) \Big] \boldsymbol{x} \boldsymbol{x}^T$$

$$\quad + \frac{1-\lambda}{K^2} \mathbb{E}_{\widetilde{\boldsymbol{x}}} \Big[ \phi'(\boldsymbol{w}_{j_1}^{[p]T} \widetilde{\boldsymbol{x}}) \phi'(\boldsymbol{w}_{j_2}^{[p]T} \widetilde{\boldsymbol{x}}) - \phi'(\boldsymbol{w}_{j_1}^T \widetilde{\boldsymbol{x}}) \phi'(\boldsymbol{w}_{j_2}^T \widetilde{\boldsymbol{x}}) \Big] \widetilde{\boldsymbol{x}} \widetilde{\boldsymbol{x}}^T$$

$$= \frac{\lambda}{K^2} \mathbb{E}_{\boldsymbol{x}} \Big[ \phi'(\boldsymbol{w}_{j_1}^{[p]T} \boldsymbol{x}) \big( \phi'(\boldsymbol{w}_{j_2}^{[p]T} \boldsymbol{x}) - \phi'(\boldsymbol{w}_{j_2}^T \boldsymbol{x}) \big) + \phi'(\boldsymbol{w}_{j_2}^T \boldsymbol{x}) \big( \phi'(\boldsymbol{w}_{j_1}^{[p]T} \boldsymbol{x}) - \phi'(\boldsymbol{w}_{j_1}^T \boldsymbol{x}) \big) \Big] \boldsymbol{x} \boldsymbol{x}^T$$

$$\quad + \frac{1-\lambda}{K^2} \mathbb{E}_{\widetilde{\boldsymbol{x}}} \Big[ \phi'(\boldsymbol{w}_{j_1}^{[p]T} \widetilde{\boldsymbol{x}}) \big( \phi'(\boldsymbol{w}_{j_2}^{[p]T} \widetilde{\boldsymbol{x}}) - \phi'(\boldsymbol{w}_{j_2}^T \widetilde{\boldsymbol{x}}) \big) + \phi'(\boldsymbol{w}_{j_2}^T \widetilde{\boldsymbol{x}}) \big( \phi'(\boldsymbol{w}_{j_1}^{[p]T} \widetilde{\boldsymbol{x}}) - \phi'(\boldsymbol{w}_{j_1}^T \widetilde{\boldsymbol{x}}) \big) \Big] \widetilde{\boldsymbol{x}} \widetilde{\boldsymbol{x}}^T$$

$$= \frac{\lambda}{K^2} \Big[ \mathbb{E}_{\boldsymbol{x}} \phi'(\boldsymbol{w}_{j_1}^{[p]T} \boldsymbol{x}) \big( \phi'(\boldsymbol{w}_{j_2}^{[p]T} \boldsymbol{x}) - \phi'(\boldsymbol{w}_{j_2}^T \boldsymbol{x}) \big) \boldsymbol{x} \boldsymbol{x}^T + \mathbb{E}_{\boldsymbol{x}} \phi'(\boldsymbol{w}_{j_2}^T \boldsymbol{x}) \big( \phi'(\boldsymbol{w}_{j_1}^{[p]T} \boldsymbol{x}) - \phi'(\boldsymbol{w}_{j_1}^T \boldsymbol{x}) \big) \boldsymbol{x} \boldsymbol{x}^T \Big]$$

$$\quad + \frac{1-\lambda}{K^2} \Big[ \mathbb{E}_{\widetilde{\boldsymbol{x}}} \phi'(\boldsymbol{w}_{j_1}^{[p]T} \widetilde{\boldsymbol{x}}) \big( \phi'(\boldsymbol{w}_{j_2}^{[p]T} \widetilde{\boldsymbol{x}}) - \phi'(\boldsymbol{w}_{j_2}^T \widetilde{\boldsymbol{x}}) \big) \widetilde{\boldsymbol{x}} \widetilde{\boldsymbol{x}}^T + \mathbb{E}_{\widetilde{\boldsymbol{x}}} \phi'(\boldsymbol{w}_{j_2}^T \widetilde{\boldsymbol{x}}) \big( \phi'(\boldsymbol{w}_{j_1}^{[p]T} \widetilde{\boldsymbol{x}}) - \phi'(\boldsymbol{w}_{j_1}^T \widetilde{\boldsymbol{x}}) \big) \widetilde{\boldsymbol{x}} \widetilde{\boldsymbol{x}}^T \Big]$$

$$:= \frac{\lambda}{K^2} (\boldsymbol{P}_1 + \boldsymbol{P}_2) + \frac{1-\lambda}{K^2} (\boldsymbol{P}_3 + \boldsymbol{P}_4).$$
(101)

For any $\boldsymbol{a} \in \mathbb{R}^d$ with $\|\boldsymbol{a}\|_2 = 1$, we have

$$\boldsymbol{a}^T \boldsymbol{P}_1 \boldsymbol{a} = \mathbb{E}_{\boldsymbol{x}} \phi'(\boldsymbol{w}_{j_1}^{[p]T} \boldsymbol{x}) \big( \phi'(\boldsymbol{w}_{j_2}^{[p]T} \boldsymbol{x}) - \phi'(\boldsymbol{w}_{j_2}^T \boldsymbol{x}) \big) (\boldsymbol{a}^T \boldsymbol{x})^2$$
(102)

where $\boldsymbol{a} \in \mathbb{R}^d$. Let $I = \phi'(\boldsymbol{w}_{j_1}^{[p]T} \boldsymbol{x}) \big( \phi'(\boldsymbol{w}_{j_2}^{[p]T} \boldsymbol{x}) - \phi'(\boldsymbol{w}_{j_2}^T \boldsymbol{x}) \big) \cdot (\boldsymbol{a}^T \boldsymbol{x})^2$. It is easy to verify there exists a group of orthonormal vectors such that $\mathcal{B} = \{\boldsymbol{a}, \boldsymbol{b}, \boldsymbol{c}, \boldsymbol{a}_4^\perp, \cdots, \boldsymbol{a}_d^\perp\}$ with $\{\boldsymbol{a}, \boldsymbol{b}, \boldsymbol{c}\}$ spans a subspace that contains $\boldsymbol{a}, \boldsymbol{w}_{j_2}$ and $\boldsymbol{w}_{j_2}^*$. Then, for any $\boldsymbol{x}$, we have a unique $\boldsymbol{z} = \begin{bmatrix} z_1, & z_2, & \cdots, & z_d \end{bmatrix}^T$ such that

$$\boldsymbol{x} = z_1 \boldsymbol{a} + z_2 \boldsymbol{b} + z_3 \boldsymbol{c} + \cdots + z_d \boldsymbol{a}_d^\perp.$$

Also, since $\boldsymbol{x} \sim \mathcal{N}(\boldsymbol{0}, \delta^2 \boldsymbol{I}_d)$, we have $\boldsymbol{z} \sim \mathcal{N}(\boldsymbol{0}, \delta^2 \boldsymbol{I}_d)$. Then, we have

$$I = \mathbb{E}_{z_1, z_2, z_3} |\phi'(\boldsymbol{w}_{j_2}^T \boldsymbol{x}) - \phi'(\boldsymbol{w}_{j_2}^{[p]T} \boldsymbol{x})| \cdot |\boldsymbol{a}^T \boldsymbol{x}|^2$$

$$= \int |\phi'(\boldsymbol{w}_{j_2}^T \boldsymbol{x}) - \phi'(\boldsymbol{w}_{j_2}^{[p]T} \boldsymbol{x})| \cdot |\boldsymbol{a}^T \boldsymbol{x}|^2 \cdot f_Z(z_1, z_2, z_3) dz_1 dz_2 dz_3,$$

where $\boldsymbol{x} = z_1 \boldsymbol{a} + z_2 \boldsymbol{b} + z_3 \boldsymbol{c}$ and $f_Z(z_1, z_2, z_3)$ is probability density function of $(z_1, z_2, z_3)$. Next, we consider spherical coordinates with $z_1 = R \cos\phi_1, z_2 = R \sin\phi_1 \sin\phi_2, z_3 = R \sin\phi_1 \cos\phi_2$. Hence,

$$I = \int |\phi'(\boldsymbol{w}_{j_2}^T \boldsymbol{x}) - \phi'(\boldsymbol{w}_{j_2}^{[p]T} \boldsymbol{x})| \cdot |R \cos\phi_1|^2 \cdot f_Z(R, \phi_1, \phi_2) R^2 \sin\phi_1 dR d\phi_1 d\phi_2.$$
(103)

It is easy to verify that $\phi'(\boldsymbol{w}_{j_2}^T \boldsymbol{x})$ only depends on the direction of $\boldsymbol{x}$ and

$$f_Z(R, \phi_1, \phi_2) = \frac{1}{(2\pi\delta^2)^{\frac{3}{2}}} e^{-\frac{z_1^2 + z_2^2 + z_3^2}{2\delta^2}} = \frac{1}{(2\pi\delta^2)^{\frac{3}{2}}} e^{-\frac{R^2}{2\delta^2}}$$

only depends on $R$. Then, we have

$$
\begin{aligned}
&I(i_2, j_2) \\
&= \int |\phi'(\boldsymbol{w}_{j_2}^T(\boldsymbol{x}/R)) - \phi'(\boldsymbol{w}_{j_2}^{[p]T}(\boldsymbol{x}/R))| \cdot |R\cos\phi_1|^2 \cdot f_Z(R)R^2\sin\phi_1 dRd\phi_1 d\phi_2 \\
&= \int_0^\infty R^4 f_z(R)dR \int_0^\pi \int_0^{2\pi} |\cos\phi_1|^2 \cdot \sin\phi_1 \cdot |\phi'(\boldsymbol{w}_{j_2}^T(\boldsymbol{x}/R)) - \phi'(\boldsymbol{w}_{j_2}^{[p]T}(\boldsymbol{x}/R))|d\phi_1 d\phi_2 \\
&\overset{(a)}{\leq} 3\delta^2 \cdot \int_0^\infty R^2 f_z(R)dR \int_0^\pi \int_0^{2\pi} \sin\phi_1 \cdot |\phi'(\boldsymbol{w}_{j_2}^T(\boldsymbol{x}/R)) - \phi'(\boldsymbol{w}_{j_2}^{[p]T}(\boldsymbol{x}/R))|d\phi_1 d\phi_2 \\
&= 3\delta^2 \cdot \mathbb{E}_{z_1,z_2,z_3} |\phi'(\boldsymbol{w}_{j_2}^T\boldsymbol{x}) - \phi'(\boldsymbol{w}_{j_2}^{[p]T}\boldsymbol{x})| \\
&\leq 3\delta^2 \cdot \mathbb{E}_{\boldsymbol{x}} |\phi'(\boldsymbol{w}_{j_2}^T\boldsymbol{x}) - \phi'(\boldsymbol{w}_{j_2}^{[p]T}\boldsymbol{x})|,
\end{aligned}
\tag{104}
$$

where the inequality (a) is derived from the fact that $|cos\phi_1| \leq 1$ and

$$
\begin{aligned}
\int_0^\infty R^4 \frac{1}{(2\pi\delta^2)^{\frac{3}{2}}} e^{-\frac{R^2}{2\delta^2}} dR &= \int_0^\infty -\frac{R^3\delta^2}{(2\pi\delta^2)^{\frac{3}{2}}} d(e^{-\frac{R^2}{2\delta^2}}) \\
&= \int_0^\infty e^{-\frac{R^2}{2\delta^2}} d\frac{R^3\delta^2}{(2\pi\delta^2)^{\frac{3}{2}}} \\
&= 3\delta^2 \int_0^\infty R^2 \frac{1}{(2\pi\delta^2)^{\frac{3}{2}}} e^{-\frac{R^2}{2\delta^2}} dR.
\end{aligned}
\tag{105}
$$

Define a set $\mathcal{A}_1 = \{\boldsymbol{x}|(\boldsymbol{w}_{j_2}^{[p]T}\boldsymbol{x})(\boldsymbol{w}_{j_2}^T\boldsymbol{x}) < 0\}$. If $\boldsymbol{x} \in \mathcal{A}_1$, then $\boldsymbol{w}_{j_2}^{[p]T}\boldsymbol{x}$ and $\boldsymbol{w}_{j_2}^T\boldsymbol{x}$ have different signs, which means the value of $\phi'(\boldsymbol{w}_{j_2}^T\boldsymbol{x})$ and $\phi'(\boldsymbol{w}_{j_2}^{[p]T}\boldsymbol{x})$ are different. This is equivalent to say that

$$
|\phi'(\boldsymbol{w}_{j_2}^T\boldsymbol{x}) - \phi'(\boldsymbol{w}_{j_2}^{[p]T}\boldsymbol{x})| = \left\{ \begin{array}{ll} 1, & \text{if } \boldsymbol{x} \in \mathcal{A}_1 \\ 0, & \text{if } \boldsymbol{x} \in \mathcal{A}_1^c \end{array} \right. .
\tag{106}
$$

Moreover, if $\boldsymbol{x} \in \mathcal{A}_1$, then we have

$$
|\boldsymbol{w}_{j_2}^{[p]T}\boldsymbol{x}| \leq |\boldsymbol{w}_{j_2}^{[p]T}\boldsymbol{x} - \boldsymbol{w}_{j_2}^T\boldsymbol{x}| \leq \|\boldsymbol{w}_{j_2}^{[p]} - \boldsymbol{w}_{j_2}\|_2 \cdot \|\boldsymbol{x}\|_2.
\tag{107}
$$

Let us define a set $\mathcal{A}_2$ such that

$$
\mathcal{A}_2 = \left\{ \boldsymbol{x} \Big| \frac{|\boldsymbol{w}_{j_2}^{[p]T}\boldsymbol{x}|}{\|\boldsymbol{w}_{j_2}^*\|_2 \|\boldsymbol{x}\|_2} \leq \frac{\|\boldsymbol{w}_{j_2}^* - \boldsymbol{w}_{j_2}\|_2}{\|\boldsymbol{w}_{j_2}^*\|_2} \right\} = \left\{ \theta_{\boldsymbol{x}, \boldsymbol{w}_{j_2}^*} \Big| |\cos\theta_{\boldsymbol{x}, \boldsymbol{w}_{j_2}^{[p]}}| \leq \frac{\|\boldsymbol{w}_{j_2}^{[p]} - \boldsymbol{w}_{j_2}\|_2}{\|\boldsymbol{w}_{j_2}^{[p]}\|_2} \right\}.
\tag{108}
$$

Hence, we have that

$$
\begin{aligned}
\mathbb{E}_{\boldsymbol{x}} |\phi'(\boldsymbol{w}_{j_2}^T\boldsymbol{x}) - \phi'(\boldsymbol{w}_{j_2}^{[p]T}\boldsymbol{x})|^2 &= \mathbb{E}_{\boldsymbol{x}} |\phi'(\boldsymbol{w}_{j_2}^T\boldsymbol{x}) - \phi'(\boldsymbol{w}_{j_2}^{[p]T}\boldsymbol{x})| \\
&= \text{Prob}(\boldsymbol{x} \in \mathcal{A}_1) \\
&\leq \text{Prob}(\boldsymbol{x} \in \mathcal{A}_2).
\end{aligned}
\tag{109}
$$

Since $\boldsymbol{x} \sim \mathcal{N}(\boldsymbol{0}, \delta^2\|\boldsymbol{a}\|_2^2\boldsymbol{I})$, $\theta_{\boldsymbol{x}, \boldsymbol{w}_{j_2}^{[p]}}$ belongs to the uniform distribution on $[-\pi, \pi]$, we have

$$
\begin{aligned}
\text{Prob}(\boldsymbol{x} \in \mathcal{A}_2) = \frac{\pi - \arccos\frac{\|\boldsymbol{w}_{j_2}^{[p]} - \boldsymbol{w}_{j_2}\|_2}{\|\boldsymbol{w}_{j_2}^{[p]}\|_2}}{\pi} &\leq \frac{1}{\pi}\tan(\pi - \arccos\frac{\|\boldsymbol{w}_{j_2}^{[p]} - \boldsymbol{w}_{j_2}\|_2}{\|\boldsymbol{w}_{j_2}^{[p]}\|_2}) \\
&= \frac{1}{\pi}\cot(\arccos\frac{\|\boldsymbol{w}_{j_2}^{[p]} - \boldsymbol{w}_{j_2}\|_2}{\|\boldsymbol{w}_{j_2}^{[p]}\|_2}) \\
&\leq \frac{2}{\pi}\frac{\|\boldsymbol{w}_{j_2}^{[p]} - \boldsymbol{w}_{j_2}\|_2}{\|\boldsymbol{w}_{j_2}^{[p]}\|_2}.
\end{aligned}
\tag{110}
$$

Hence, (104) and (110) suggest that

$$I \leq \frac{6\delta^2}{\pi} \frac{\|\boldsymbol{w}_{j_2} - \boldsymbol{w}_{j_2}^{[p]}\|_2}{\sigma_K} \cdot \|\boldsymbol{a}\|_2^2. \tag{111}$$

The same bound that shown in (111) holds for $\boldsymbol{P}_2$ as well.

$\boldsymbol{P}_3$ and $\boldsymbol{P}_4$ satisfy (111) except for changing $\delta^2$ to $\tilde{\delta}^2$.

Therefore, we have

$$
\begin{aligned}
&\|\nabla^2 f(\boldsymbol{W}^{[p]}; p) - \nabla^2 f(\boldsymbol{W}; p)\|_2 \\
&= \max_{\|\boldsymbol{\alpha}\|_2 \leq 1} \left| \boldsymbol{\alpha}^T (\nabla^2 f(\boldsymbol{W}^{[p]}; p) - \nabla^2 f(\boldsymbol{W}; p)) \boldsymbol{\alpha} \right| \\
&\leq \sum_{j_1=1}^{K} \sum_{j_2=1}^{K} \left| \boldsymbol{\alpha}_{j_1}^T \left( \frac{\partial^2 f}{\partial \boldsymbol{w}_{j_1} \partial \boldsymbol{w}_{j_2}} (\boldsymbol{W}^{[p]}; p) - \frac{\partial^2 f}{\partial \boldsymbol{w}_{j_1} \partial \boldsymbol{w}_{j_2}} (\boldsymbol{W}; p) \right) \boldsymbol{\alpha}_{j_2} \right| \\
&\leq \frac{1}{K^2} \sum_{j_1=1}^{K} \sum_{j_2=1}^{K} \left( \lambda \|\boldsymbol{P}_1 + \boldsymbol{P}_2\|_2 + (1-\lambda) \|\boldsymbol{P}_3 + \boldsymbol{P}_4\|_2 \right) \|\boldsymbol{\alpha}_{j_1}\|_2 \|\boldsymbol{\alpha}_{j_2}\|_2 \\
&\leq \frac{1}{K^2} \sum_{j_1=1}^{K} \sum_{j_2=1}^{K} 4(\lambda \delta^2 + (1-\lambda)\tilde{\delta}^2) \frac{\|\boldsymbol{w}_{j_2}^{[p]} - \boldsymbol{w}_{j_2}\|_2}{\sigma_K} \|\boldsymbol{\alpha}_{j_1}\|_2 \|\boldsymbol{\alpha}_{j_2}\|_2 \\
&\leq \frac{4}{K} (\lambda \delta^2 + (1-\lambda)\tilde{\delta}^2) \cdot \frac{\|\boldsymbol{W}^{[p]} - \boldsymbol{W}\|_2}{\sigma_K},
\end{aligned}
\tag{112}
$$

where $\boldsymbol{\alpha} \in \mathbb{R}^{Kd}$ and $\boldsymbol{\alpha}_j \in \mathbb{R}^d$ with $\boldsymbol{\alpha} = [\boldsymbol{\alpha}_1^T, \boldsymbol{\alpha}_2^T, \cdots, \boldsymbol{\alpha}_K^T]^T$. $\qquad \square$

# I   INITIALIZATION VIA TENSOR METHOD

In this section, we briefly summarize the tensor initialization in (Zhong et al., 2017) by studying the target function class as

$$y = \frac{1}{K} \sum_{j=1}^{K} v_j^* \phi(\boldsymbol{w}_j^{*T} \boldsymbol{x}), \tag{113}$$

where $v_j^* \in R$. Note that for ReLU function, we have $v_j^* \phi(\boldsymbol{w}_j^{*T} \boldsymbol{x}) = \text{sign}(v_j^*) \phi(|v_j^*| \boldsymbol{w}_j^{*T} \boldsymbol{x})$. Without loss of generalization, we can assume $v_j^* \in \{+1, -1\}$. Additionally, it is clear that the function studied in (2) is the special case of (113) when $v_j^* = 1$ for all $j$. In addition, Theorem 5.6 in (Zhong et al., 2017) show that the sign of $v_j^*$ can be directly recovered using tensor initialization, which indicates the the equivalence of (2) and (113) when using tensor initialization.

We first define some high order momenta in the following way:

$$\boldsymbol{M}_1 = \mathbb{E}_{\boldsymbol{x}}\{y\boldsymbol{x}\} \in \mathbb{R}^d, \tag{114}$$

$$\boldsymbol{M}_2 = \mathbb{E}_{\boldsymbol{x}}\left[ y(\boldsymbol{x} \otimes \boldsymbol{x} - \delta^2 \boldsymbol{I}) \right] \in \mathbb{R}^{d \times d}, \tag{115}$$

$$\boldsymbol{M}_3 = \mathbb{E}_{\boldsymbol{x}}\left[ y(\boldsymbol{x}^{\otimes 3} - \boldsymbol{x}\widetilde{\otimes}\delta^2 \boldsymbol{I}) \right] \in \mathbb{R}^{d \times d \times d}, \tag{116}$$

where $\mathbb{E}_{\boldsymbol{x}}$ is the expectation over $\boldsymbol{x}$ and $\boldsymbol{z}^{\otimes 3} := \boldsymbol{z} \otimes \boldsymbol{z} \otimes \boldsymbol{z}$. The operator $\widetilde{\otimes}$ is defined as

$$\boldsymbol{v}\widetilde{\otimes}\boldsymbol{Z} = \sum_{i=1}^{d_2} (\boldsymbol{v} \otimes \boldsymbol{z}_i \otimes \boldsymbol{z}_i + \boldsymbol{z}_i \otimes \boldsymbol{v} \otimes \boldsymbol{z}_i + \boldsymbol{z}_i \otimes \boldsymbol{z}_i \otimes \boldsymbol{v}), \tag{117}$$

for any vector $\boldsymbol{v} \in \mathbb{R}^{d_1}$ and $\boldsymbol{Z} \in \mathbb{R}^{d_1 \times d_2}$.

Following the same calculation formulas in the Claim 5.2 (Zhong et al., 2017), there exist some known constants $\psi_i, i = 1, 2, 3$, such that

$$M_1 = \sum_{j=1}^{K} \psi_1 \cdot \|w_j^*\|_2 \cdot \overline{w}_j^*, \tag{118}$$

$$M_2 = \sum_{j=1}^{K} \psi_2 \cdot \|w_j^*\|_2 \cdot \overline{w}_j^* \overline{w}_j^{*T}, \tag{119}$$

$$M_3 = \sum_{j=1}^{K} \psi_3 \cdot \|w_j^*\|_2 \cdot \overline{w}_j^{*\otimes 3}, \tag{120}$$

where $\overline{w}_j^* = w_j^*/\|w_j^*\|_2$ in (114)-(116) is the normalization of $w_j^*$. Therefore, we can see that the information of $\{w_j^*\}_{j=1}^{K}$ are separated as the direction of $w_j$ and the magnitude of $w_j$ in $M_1$, $M_2$ and $M_3$.

$M_1$, $M_2$ and $M_3$ can be estimated through the samples $\{(x_n, y_n)\}_{n=1}^{N}$, and let $\widehat{M}_1$, $\widehat{M}_2$, $\widehat{M}_3$ denote the corresponding estimates. First, we will decompose the rank-$K$ tensor $M_3$ and obtain the $\{\overline{w}_j^*\}_{j=1}^{K}$. By applying the tensor decomposition method (Kuleshov et al., 2015) to $\widehat{M}_3$, the outputs, denoted by $\widehat{\overline{w}}_j^*$, are the estimations of $\{s_j \overline{w}_j^*\}_{j=1}^{K}$, where $s_j$ is an unknown sign. Second, we will estimate $s_j$, $v_j^*$ and $\|w_j^*\|_2$ through $M_1$ and $M_2$. Note that $M_2$ does not contain the information of $s_j$ because $s_j^2$ is always 1. Then, through solving the following two optimization problem:

$$\begin{aligned}
\widehat{\alpha}_1 &= \arg\min_{\alpha_1 \in \mathbb{R}^K} : \quad \left| \widehat{M}_1 - \sum_{j=1}^{K} \psi_1 \alpha_{1,j} \widehat{\overline{w}}_j^* \right|, \\
\widehat{\alpha}_2 &= \arg\min_{\alpha_2 \in \mathbb{R}^K} : \quad \left| \widehat{M}_2 - \sum_{j=1}^{K} \psi_2 \alpha_{2,j} \widehat{\overline{w}}_j^* \widehat{\overline{w}}_j^{*T} \right|,
\end{aligned} \tag{121}$$

The estimation of $s_j$ can be given as

$$\hat{s}_j = \text{sign}(\widehat{\alpha}_{1,j}/\widehat{\alpha}_{2,j}).$$

Also, we know that $|\widehat{\alpha}_{1,j}|$ is the estimation of $\|w_j^*\|$ and

$$\hat{v}_j = \text{sign}(\widehat{\alpha}_{1,j}/s_j) = \text{sign}(\widehat{\alpha}_{2,j}).$$

Thus, $W^{(0)}$ is given as

$$\left[ \text{sign}(\widehat{\alpha}_{2,1})\widehat{\alpha}_{1,1}\widehat{\overline{w}}_1^*, \quad \cdots, \quad \text{sign}(\widehat{\alpha}_{2,K})\widehat{\alpha}_{1,K}\widehat{\overline{w}}_K^* \right].$$

---

**Subroutine 1** Tensor Initialization Method

---

1: **Input:** labeled data $\mathcal{D} = \{(x_n, y_n)\}_{n=1}^{N}$;
2: Partition $\mathcal{D}$ into three disjoint subsets $\mathcal{D}_1, \mathcal{D}_2, \mathcal{D}_3$;
3: Calculate $\widehat{M}_1, \widehat{M}_2$ following (114), (115) using $\mathcal{D}_1, \mathcal{D}_2$, respectively;
4: Obtain the estimate subspace $\widehat{V}$ of $\widehat{M}_2$;
5: Calculate $\widehat{M}_3(\widehat{V}, \widehat{V}, \widehat{V})$ through $\mathcal{D}_3$;
6: Obtain $\{\widehat{s}_j\}_{j=1}^{K}$ via tensor decomposition method (Kuleshov et al., 2015) on $\widehat{M}_3(\widehat{V}, \widehat{V}, \widehat{V})$;
7: Obtain $\widehat{\alpha}_1, \widehat{\alpha}_2$ by solving optimization problem (121);
8: **Return:** $w_j^{(0)} = \text{sign}(\widehat{\alpha}_{2,j})\widehat{\alpha}_{1,j}\widehat{V}\widehat{u}_j$ and $v_j^{(0)} = \text{sign}(\widehat{\alpha}_{2,j}), j = 1, ..., K$.

---

To reduce the computational complexity of tensor decomposition, one can project $\widehat{M}_3$ to a lower-dimensional tensor (Zhong et al., 2017). The idea is to first estimate the subspace spanned by $\{w_j^*\}_{j=1}^{K}$, and let $\widehat{V}$ denote the estimated subspace. Moreover, we have

$$M_3(\widehat{V}, \widehat{V}, \widehat{V}) = \mathbb{E}_x \left[ y \left( (\widehat{V}^T x)^{\otimes 3} - (\widehat{V}^T x) \widetilde{\otimes} \mathbb{E}_x (\widehat{V}^T x)(\widehat{V}^T x)^T \right) \right] \in \mathbb{R}^{K \times K \times K}, \tag{122}$$

Then, one can decompose the estimate $\widehat{\boldsymbol{M}}_3(\widehat{\boldsymbol{V}}, \widehat{\boldsymbol{V}}, \widehat{\boldsymbol{V}})$ to obtain unit vectors $\{\hat{\boldsymbol{s}}_j\}_{j=1}^K \in \mathbb{R}^K$. Since $\overline{\boldsymbol{w}}^*$ lies in the subspace $\boldsymbol{V}$, we have $\boldsymbol{V}\boldsymbol{V}^T\overline{\boldsymbol{w}}_j^* = \overline{\boldsymbol{w}}_j^*$. Then, $\widehat{\boldsymbol{V}}\hat{\boldsymbol{s}}_j$ is an estimate of $\overline{\boldsymbol{w}}_j^*$. The initialization process is summarized in Subroutine 1.

## J CLASSIFICATION PROBLEMS

The framework in this paper is extendable to binary classification problem. For binary classification problem, the output $y$ given input $\boldsymbol{x}$ is defined as

$$\text{Prob}\{y = 1\} = g(\boldsymbol{W}^*; \boldsymbol{x}) \tag{123}$$

with some ground truth parameter $\boldsymbol{W}^*$. To guarantee the output is within $[0, 1]$, the activation function is often used as sigmoid. For classification, the loss function is cross-entropy, and the objective function over labeled data $\mathcal{D}$ is defined as

$$f_{\mathcal{D}}(\boldsymbol{W}) = \frac{1}{N} \sum_{(\boldsymbol{x}_n, y_n) \in \mathcal{D}} -y_n \log g(\boldsymbol{W}; \boldsymbol{x}_n) - (1 - y_n) \log(1 - g(\boldsymbol{W}; \boldsymbol{x}_n)). \tag{124}$$

The expectation of objective function can be written as

$$\begin{aligned}
\mathbb{E}_{\mathcal{D}} f_{\mathcal{D}}(\boldsymbol{W}) &= \mathbb{E}_{(\boldsymbol{x}, y)} - y \log(g(\boldsymbol{W}; \boldsymbol{x}_n)) - (1 - y) \log(1 - g(\boldsymbol{W}; \boldsymbol{x})) \\
&= \mathbb{E}_{\boldsymbol{x}} \mathbb{E}_{(y|\boldsymbol{x})} - y \log(g(\boldsymbol{W}; \boldsymbol{x}_n)) - (1 - y) \log(1 - g(\boldsymbol{W}; \boldsymbol{x})) \\
&= \mathbb{E}_{\boldsymbol{x}} \Big[ -g(\boldsymbol{W}^*; \boldsymbol{x}) \log(g(\boldsymbol{W}; \boldsymbol{x}_n)) - (1 - g(\boldsymbol{W}^*; \boldsymbol{x})) \log(1 - g(\boldsymbol{W}; \boldsymbol{x}_n)) \Big]
\end{aligned} \tag{125}$$

Please note that (125) is exactly the same as (32) with $\lambda = 1$ when the loss function is squared loss.

For cross entropy loss function, the second order derivative of (125) is calculated as

$$\frac{\partial f_{\mathcal{D}}(\boldsymbol{W})}{\partial \boldsymbol{w}_j \partial \boldsymbol{w}_k} = \frac{1}{N} \Big[ \frac{y_n}{g^2(\boldsymbol{W}; \boldsymbol{x})} + \frac{1 - y_n}{(1 - g(\boldsymbol{W}; \boldsymbol{x}))^2} \Big] \cdot \phi'(\boldsymbol{w}_j^T \boldsymbol{x}) \phi'(\boldsymbol{w}_k^T \boldsymbol{x}) \boldsymbol{x} \boldsymbol{x}^T. \tag{126}$$

when $j \neq k$. Refer to (88) in (Fu et al., 2020) or (132) in (Zhang et al., 2020b), we have

$$\Big\| \frac{y_n(\phi'(\boldsymbol{w}_j^T \boldsymbol{x})\phi'(\boldsymbol{w}_k^T \boldsymbol{x}))}{g^2(\boldsymbol{W}; \boldsymbol{x})} \Big\|_2 \leq \Big\| \frac{\phi'(\boldsymbol{w}_j^T \boldsymbol{x})\phi'(\boldsymbol{w}_k^T \boldsymbol{x})}{g^2(\boldsymbol{W}; \boldsymbol{x})} \Big\|_2 \leq K^2. \tag{127}$$

Following similar steps in (90), from Defintion 2, we know that $\alpha_j^T \frac{\partial f_{\mathcal{D}}(\boldsymbol{W})}{\partial \boldsymbol{w}_j \partial \boldsymbol{w}_k} \alpha_k$ belongs to the sub-exponential distribution. Therefore, similiar results for objective function with cross-entropy loss can be established as well. One can check (Fu et al., 2020) or (Zhang et al., 2020b) for details.

## K ONE-HIDDEN LAYER NEURAL NETWORK WITH TOP LAYER WEIGHTS

For a general one-hidden layer neural network, the output of the neural network is defined as

$$g(\boldsymbol{W}, \boldsymbol{v}; \boldsymbol{x}) = \frac{1}{K} \sum_{j=1}^K v_j \phi(\boldsymbol{w}_j^T \boldsymbol{x}), \tag{128}$$

where $\boldsymbol{v} = [v_1, v_2, \cdots, v_K] \in R^K$. Then, the target function can be defined as

$$y = g(\boldsymbol{W}^*, \boldsymbol{v}^*; \boldsymbol{x}) = \frac{1}{K} \sum_{j=1}^K v_j^* \phi(\boldsymbol{w}_j^{*T} \boldsymbol{x}) \tag{129}$$

for some unknown weights $\boldsymbol{W}^*$ and $\boldsymbol{v}^*$.

In the following paragraphs, we will provide a short description for the equivalence of (129) and (2) in theoretical analysis. Note that for ReLU functions, we have $v_j \phi(\boldsymbol{w}_j^T \boldsymbol{x}) = \text{sign}(v_j) \phi(|v_j| \boldsymbol{w}_j^T \boldsymbol{x})$.

Without loss of generalization, we can assume $v_j, v_j^* \in \{+1, -1\}$ for all $j \in [K]$[5]. From Appendix I, we know that the sign of $v_j^*$ can exactly estimated through tensor initialization. There, we can focus on analysis the neural network in the form as

$$g(\boldsymbol{W}; \boldsymbol{x}) = \frac{1}{K} \sum_{j=1}^{K} v_j^* \phi(\boldsymbol{w}_j^T \boldsymbol{x}). \tag{130}$$

Considering the objective function in (1) and population risk function in (17), we have

$$
\begin{aligned}
&\left\| \frac{\partial \hat{f}}{\partial \boldsymbol{w}_k}(\boldsymbol{W}) - \frac{\partial f}{\partial \boldsymbol{w}_k}(\boldsymbol{W}; p) \right\|_2 \\
=&\left\| \frac{\lambda}{K^2 N} \sum_{j=1}^{K} v_j^* \Big[ \sum_{n=1}^{N} \big( \phi(\boldsymbol{w}_j^T \boldsymbol{x}_n) - \phi(\boldsymbol{w}_j^{*T} \boldsymbol{x}_n) \big) \boldsymbol{x}_n - \mathbb{E}_{\boldsymbol{x}} \big( \phi(\boldsymbol{w}_j^T \boldsymbol{x}) - \phi(\boldsymbol{w}_j^{[p]T} \boldsymbol{x}) \big) \boldsymbol{x} \Big] \right. \\
&\left. + \frac{1-\lambda}{K^2 M} v_j^* \sum_{j=1}^{K} \Big[ \sum_{m=1}^{M} \big( \phi(\boldsymbol{w}_j^T \widetilde{\boldsymbol{x}}_m) - \phi(\widetilde{\boldsymbol{w}}_j^T \widetilde{\boldsymbol{x}}_m) \big) \widetilde{\boldsymbol{x}}_m - \mathbb{E}_{\widetilde{\boldsymbol{x}}} \big( \phi(\boldsymbol{w}_j^T \widetilde{\boldsymbol{x}}) - \phi(\boldsymbol{w}_j^{[p]T} \widetilde{\boldsymbol{x}}) \big) \widetilde{\boldsymbol{x}} \Big] \right\|_2 \\
\leq& \sum_{j=1}^{K} \cdot |v_j^*| \cdot \left\| \frac{\lambda}{K^2 N} \Big[ \sum_{n=1}^{N} \big( \phi(\boldsymbol{w}_j^T \boldsymbol{x}_n) - \phi(\boldsymbol{w}_j^{*T} \boldsymbol{x}_n) \big) \boldsymbol{x}_n - \mathbb{E}_{\boldsymbol{x}} \big( \phi(\boldsymbol{w}_j^T \boldsymbol{x}) - \phi(\boldsymbol{w}_j^{[p]T} \boldsymbol{x}) \big) \boldsymbol{x} \Big] \right. \\
&\left. + \frac{1-\lambda}{K^2 M} \Big[ \sum_{m=1}^{M} \big( \phi(\boldsymbol{w}_j^T \widetilde{\boldsymbol{x}}_m) - \phi(\widetilde{\boldsymbol{w}}_j^T \widetilde{\boldsymbol{x}}_m) \big) \widetilde{\boldsymbol{x}}_m - \mathbb{E}_{\widetilde{\boldsymbol{x}}} \big( \phi(\boldsymbol{w}_j^T \widetilde{\boldsymbol{x}}) - \phi(\boldsymbol{w}_j^{[p]T} \widetilde{\boldsymbol{x}}) \big) \widetilde{\boldsymbol{x}} \Big] \right\|_2 \\
=& \sum_{j=1}^{K} \left\| \frac{\lambda}{K^2 N} \Big[ \sum_{n=1}^{N} \big( \phi(\boldsymbol{w}_j^T \boldsymbol{x}_n) - \phi(\boldsymbol{w}_j^{*T} \boldsymbol{x}_n) \big) \boldsymbol{x}_n - \mathbb{E}_{\boldsymbol{x}} \big( \phi(\boldsymbol{w}_j^T \boldsymbol{x}) - \phi(\boldsymbol{w}_j^{[p]T} \boldsymbol{x}) \big) \boldsymbol{x} \Big] \right. \\
&\left. + \frac{1-\lambda}{K^2 M} \Big[ \sum_{m=1}^{M} \big( \phi(\boldsymbol{w}_j^T \widetilde{\boldsymbol{x}}_m) - \phi(\widetilde{\boldsymbol{w}}_j^T \widetilde{\boldsymbol{x}}_m) \big) \widetilde{\boldsymbol{x}}_m - \mathbb{E}_{\widetilde{\boldsymbol{x}}} \big( \phi(\boldsymbol{w}_j^T \widetilde{\boldsymbol{x}}) - \phi(\boldsymbol{w}_j^{[p]T} \widetilde{\boldsymbol{x}}) \big) \widetilde{\boldsymbol{x}} \Big] \right\|_2,
\end{aligned}
\tag{131}
$$

which is exact the same as (89). Similar results can be derived for Lemma 12. Therefore, the conclusions and proofs of Lemma 1 and Lemma 2 does not change at all.

Additionally, fixing the second-layer weights and only training the hidden layer is the state-of-the-art practice in analyzing two-layer neural networks (Arora et al., 2019b;a; Allen-Zhu et al., 2019; Safran & Shamir, 2018; Li & Liang, 2018; Brutzkus & Globerson, 2017; Oymak & Soltanolkotabi, 2018; Zhang et al., 2019). Additionally, as indicated in (Safran & Shamir, 2018), training a one-hidden-layer neural network with all $v_j$ fixed as 1 has intractable many spurious local minima, which indicates that training problem is not trivial.

---

[5]To see this, one can view $|v_j^*| \boldsymbol{w}_j^*$ as the new ground truth weights, and the goal for this paper is to recover the new ground truth weights.

