# OpenReview forum: "How unlabeled data improve generalization in self-training? A one-hidden-layer theoretical analysis"
_ICLR.cc/2022/Conference — ICLR 2022 Poster_

### Official Review · Reviewer_SoSg · 2021-10-31

**Correctness:** 4
**Technical Novelty And Significance:** 4
**Empirical Novelty And Significance:** 3
**Recommendation:** 8
**Confidence:** 3

**Main Review:**

This is an impressive, technically solid and well-written paper with some intriguing results. The authors take particular care in explaining the general strategy of their proof, not just in the main paper, but also in various parts of the supplementary. There may be some things to improve (that I would like to see in the camera ready version), but overall this paper is of significantly above average quality amongst accepted papers. There are a couple of typos but far fewer than average for papers published at top tier conferences.

/
============================================
Comments/questions on the general mathematical content.
============================================


1. I think the results presented are a mathematical curiousity highly worthy of further study: it is worth noting that the number of labelled samples is assumed to be greater than a multiple of the number of parameters, and the paper studies a regime which is not data-sparse *statistically*, but which is data-sparse from the *optimization* standpoint. A priori, it is reasonable to assume that the global optimum of the supervised learning problem that ignores the unlabelled data would still approach the ground truth and perhaps outperform the current model. Furthermore, note that both the labelled and the unlabelled datapoints are assumed to come from gaussian distributions, with different variances, thus there is almost no statistical information coming from the unlabelled data (the only possible information it could contain is about the mean (0) or about the fact that the data is isotropic. Thus the self-training strategy appears to function only as an optimization trick that relies on some form of self regularisation/momentum.

Questions for the authors related to comment [1]:

1.1. Do you agree that the effect is only on optimization, or do you believe that the mean of the data or the isotropicity being communicated by the unlabelled data play a role?

1.2. Do you think the result would still hold if the unlabelled data was chosen from a different non istropic gaussian distribution?

1.3.  Honestly, do you think it may actually be possible to show the same convergence rates for the *supervised* learning problem (without relying on the unlabelled data at all) by borrowing from your techniques and doing away with the unlabelled data altogether?

1.4. Did you try applying such a model on real data (e.g. image data) where instead of also picking real data for the unlabelled dataset, you draw samples from a suitable Gaussian?


2.   I guess it would be nice to go into more details about the relationship between the results here and those in reference [2]: it is briefly mentioned at the top of page 2, but more details about the proof techniques in the appendix would be nice. It seems that in that reference, the authors study a completely different effect: they study classification assuming the classes are well-separated. In particular, there unambiguously is statistical information contained in the unlabelled data in this case.  It would be nice to have a one page summary of the results and proof techniques in [2] in the supplementary.


3. You mention in Lemma 3 that although your main theorem 2 requires an assumption about the initial guess not being too far away from the ground truth, any initial point chosen with the "tensor method" satisfies this assumption with high probability. Trying to understand this method requires going through the very long paper [1] and parts of other papers (including [3]). It would be nice to have an extra section in the appendix which explains the "tensor method" from first principles, with suitable pointers to the relevant results.

//
 =================================
Small issues with the maths/presentation
==================================


1. I think there is a slight problem with the proof of lemma 1: I guess the lower bound only holds for any constant strictly less than 1/11, but doesn't necessarily hold for the constant 1/11 as is claimed here. Indeed, first of ll on page 26, equation (76), I think there should be a minus sign instead of a plus at the second line. Then on a related note, I agree that combining equation (79) with Lemma 11 yields equation (80) (with the \simless sign indicating there is an unstated constatant which can be arbitrarily small since one chooses it in the interpretation of the condition on p in equation (19), but combining this with equation 76 only shows the a lower bound of the type "$a-o(a)$" where "$a$" is the lower bound claimed. Hence the constant  should be anything slightly smaller than 1/11.

2. On page 27, you use Lemma D6 from [1]. It would probably be better to write down the lemma in full and announce its role in advance at the beginning of the proof of Lemma 1.


3. Question: I am not sure I get how you go from the third line to the fourth line in equation (78). Could you elaborate? (I agree only that with an extra factor of 2 it would be trivial).


4. $\kappa$ used extensively in the main paper (starting on page 4) but is only defined in the appendix. Even the fact that $c(\kappa)$ denotes "a constant that is allowed to depend on $\kappa$" (I presume) is worth mentioning explicitly (currently not even mentioned in the supplementary).


5. I think the definition of the population risk function (equation (17)) has squares missing (the rest of the calculations are indeed consistent with the formula with the squares present).


6. The proof of Lemma 10 includes statements about the properties of $\mu$ which are not stated in the lemma statement (though they are mentioned in the main paper with a pointer to the supplementary). The lemma statement should be enriched.

7. A little bit after equation (94) on page 31,  there is a $\bar{\omega}_j$ which is not defined.

8 At the second line of equation (95) on page 31, there is a $\phi'(x)$ which should be $\phi'$ $([w_{j_1}^{[p]}]^\top x)$.

9. On page 32, on the line that starts with "Also, since $x\sim....$ it is claimed that $z$ is an isotropic normal distribution with variance $\delta^2\|a\|^2$. I guess this is only true if you force $b,c,...a_d^\perp$ to be normed, and also replace the "a" in the basis by a normalised version (currently it is not even stated that the vectors are orthonormal). I mean it is clear what you are doing here but there are minor corrections to make in the notation.


10. I think there are minus signs missing inside the exponents in the equation  between equations (97) and (98). Furtheremore I guess $x_1,x_2,x_3$ should be$ z_1,z_2,z_3$.


11. Question: How do you go from the third to the fourth line inside equation (98)? I know it is just computation but still, I would like to see details.

12.  Question: In the main paper at the bottom of page 6, "the lower bound on p means that $W^0$ cannot be too faraway from $W^*$"
Do you mean the *upper* bound on p? It seems lower bounds on the distance between $W^0$ and $W^*$ correspond to upper bounds on p and vice versa.



13.  I guess at the top of page 27 one shouldn't write "for any $\\|\alpha\\|=1$" since $\alpha$ runs in the min in the equations below. It would also be nice to remind people that $\\|\alpha\\|^2=1$ refers inside the norm to the flattened version of $\alpha$.


14 . Page 25, equation (71): should the last two (out of three) right hand sides be swapped ?It seems that $H_2^2(\delta)$ is $O(\delta^4)$ so it is the second expression on the LHS that should be in $\Theta(\delta^2)-\Theta(\delta^4)$.



//
===================================
Minor typos/minor grammatical errors
===================================

1. Title: =====>  "How *does* unlabeled data improve..."

2. Beginning of section 2: "the objective function is to find a neural network..."  ====> "the aim is to find a neural network..."


3. Bottom of page 8: "the block in white depicts all the trials are successful..." =====>" the white blocks correspond to succesful trials, while..."


4. The typography of the title for the appendix could be improved.

5. In point 2 at the top of page 15, "Then, utilising lemma 11...to obtain..." (there is no verb) =====> "Then, we utilize lemma 11 to obtain..."

6. Statement of lemma 4 on page 18: "suppose ...be a matrix" ===> "let....be a matrix"


7. Bottom of page 19 (beginning of the last paragraph): "by intermediate value theorem" ====>"by the  intermediate value theorem"

8. Bottom of page 19: "Final, by..." ====> "Finally, by..."


9.Beginning of section E on page 22: "by intermediate value theorem" ====>"by the  intermediate value theorem"

10. Beginning of section F on page 25: ..."check that ReLU activation function satisfies the conditions" =====>"check that the ReLU activation function satisfies the conditions"

11. page 25, last line: "\mu is strictly decreasing function"===> "\mu is a strictly decreasing function"

12. First sentence of section G2. "The error bound between $....$ is dividing into bounding $....$, $...$, $...$, and $...$."
=====>"The task of bounding of the quantity  $....$ is divided into bounding $....$, $...$, $...$, and $...$."





//===============After Rebuttal============

I am happy that the authors tried hard to address all of my concerns, I will keep my original score.



Here are my takeaways by theme:

Tensor Method:   In particular, I am very glad to see that they have taken the trouble to write a general explanation of the tensor initialisation method. The simplification by introducing the Psi_i etc. is particularly useful to understand the main point (as opposed to the precise definition of the constants given in the reference).
That is very helpful. However, it could be made even better by introducing the method from Kuleshov et al 2015, which is currently missing.


MVT:   I am very happy with the response from the authors regarding the MVT.




Second layer weights: I am reasonably happy with the answer in appendix K. However, more details are needed. The analysis given here only successfully explains how to fix the proof of Lemma 2, not lemma 1 (which contains second order terms. I think the authors should also redact the relevant part of the paper and appendix better. Currently, it is not unambiguously stated at the top of appendix K that the extension proposed does not actually suggest *training* the second layer weights. I know it is explained (defensively) at the end of the appendix, but things should be explained more straightforwardly from the beginning.







// =====After rebuttal: minor typos introduced in the revision=============

P 27: Typiclly, for Relu function ====> TypicAlly, for THE Relu function

P 28: when phi is THE Relu function

P 37 =====> The expectation of THE objective function
.... Is THE squarE loss


P 39 ...can BE estimated exactly through.... (Missing verb)



P 40 Conclusions does not change at al ======> conclusions DO not change at all.










//
===============================
References:
===============================


[1] Kai Zhong, Zhao Song, Prateek Jain, Peter L. Bartlett, Inderjit S. Dhillon. "Recovery Guarantees for One-hidden-layer Neural Networks", JMLR 2017

[2] Colin Wei, Kendrick Shen, Yining Chen, Tengyu Ma. "Theoretical Analysis of Self-Training with Deep Networks on Unlabeled Data", ICLR 2021

[3] Volodymyr Kuleshov, Arun Chaganty, Percy Liang. Tensor Factorization via Matrix Factorization. AiStats 2015.





**Summary Of The Paper:**

In this paper, the authors make first forrays into the study of the theoretical properties of the following iterative self-training algorithm in a semi-supervised regression setting where we have N labeled samples and M unlabeled samples (and the ground truth is realisable):

(1)  train the model on the labeled data, obtain pseudo labels for the unlabeled data by feeding them through the trained model
(2)  train an auxiliary loss (see equation (1)) on the augmented dataset the includes the unlabeled data with the corresponding pseudolabels (the loss is a convex combination of the empirical loss on the labelled and unlabeled datasets), and
(3) iteratively apply step (2) until convergence.

It is assumed that the ground truth is realisable (though without overparametrization assumptions), the labels are observed exactly (without noise), the model is a simple two-layer neural network with the second layer weights all fixed to one, and the labeled and unlabeled datapoints come from two isotropic Gaussian distributions of variances $\delta$ and $\tilde{\delta}$.

In theorem 2, the following surprising result is shown: under some reasonable assumptions on the initial point and the number of unlabeled datapoints, for a suitable choice of the convex combination parameter $\lambda$, the model converges towards the ground truth with an assumption on the number of samples which is weaker (by a constant factor at best) than what is required in the case where no unlabeled data is provided (which is a particular case of the case studied in [1])

In the more general Theorem 1, it is shown that under much weaker condition (without a requirement on the number of labelled samples), the model still converges towards a given convex combination of the ground truth and the initial point, and is guaranteed to outperform the initial model.


The main idea of the proofs is to define an auxiliary loss referred to as the population risk function (which estimates the risk taking as "labels" the images of the points by a convex combination of the ground truth and the initial point, cf. equation (17)) and to show (1) that the optimization lanscape of this functions is mild  (mainly proved in Lemmas 1 and 11), and (2) that this auxiliary function is close to the empirical risk which is minimized by the model (mainly proved in Lemma 2). The proofs of the techincal content of lemmas 1 and 2 rely both on computational geometry arguments and on existing results from [1].


Experiments are provided which show:
(1) an excellent match between the dependence of the bounds and that of the observed generalisation gaps on various parameters and
(2) that the unlabeled data indeed improves performance in data-sparse regimes on some real data.



=======================
Reference:
=======================


[1] Kai Zhong, Zhao Song, Prateek Jain, Peter L. Bartlett, Inderjit S. Dhillon. "Recovery Guarantees for One-hidden-layer Neural Networks", JMLR 2017


**Summary Of The Review:**

This is an impressive, technically solid and well-written paper with some intriguing results. There are only very minor issues.
The paper could perhaps be made even better by adding further reader-friendly discussions of some of the related works to make it appeal to a wider audience, as well as perhaps by discussing the main results and their implications in more details.

---

### Official Review · Reviewer_uV34 · 2021-11-04

**Correctness:** 2
**Technical Novelty And Significance:** 3
**Empirical Novelty And Significance:** 3
**Recommendation:** 6
**Confidence:** 3

**Main Review:**

**Pros**

- This paper tackles an interesting and important theoretical question behind the success of self-training where the model is repeatedly retrained on pseudo-labels generated on unlabeled data.
- The experimental results seem to support the theoretical claims extremely well, and the preliminary experiments on realistic models and data also agree with some of the theoretical predictions.

**Cons**

- There may be significant technical flaws in the proof. Both the proof for theorem 1 and the proof for theorem 2 hinge on what the paper referred to as the “intermediate value theorem” (eq 36 and later eq 56). However, the theorem being invoked here seems to be a vector version of the *mean value theorem* applied to $\nabla f$. Had it just been a naming mistake, it would be ok, but I believe that this particular MVT might not exist. I was able to find a version for functions of scalar input and vector output in Rudin [1] which can also be found on the wikipedia page (https://en.wikipedia.org/wiki/Mean_value_theorem#Mean_value_theorem_for_vector-valued_functions), but this is actually an *inequality* instead of an equality. Incidentally, the same wikipedia section states "There is no exact analog of the mean value theorem for vector-valued functions." I also found another version of the theorem on stackexchange (​​https://math.stackexchange.com/questions/1397248/mean-value-theorem-for-vector-valued-multivariable-function) which is an equality but requires an additional vector $a$ and still does not apply to the vector version that is shown in the proof. If there is a reference for this particular version of MVT, it would be nice to have it in the paper since it is, in my opinion, highly non-trivial if it does exist. This error could be plausibly fixed by using Taylor expansion instead of MVT but the it would require some changes to the proofs.
- Aside from the problem above, the paper also makes a somewhat strong assumption that makes the result much weaker than it appears. Specifically, the paper assumes that the nework is of the form $g(x) = \frac{1}{k} \sum_{i=1}^k \phi(w_i \cdot x)$. Note that this function is **convex** in $W$! This is not what we conventionally call a *one-hiden-layer neural network* but rather just a convex function which is much less expressive.  In contrast, [2] uses $g(x) = \sum_{i=1}^k v_i \phi(w_i \cdot x)$ which is actually a neural network. Although the optimization land scape with squared loss is still non-convex, the function class is much less expressive than a neural network. I am not sure if this function class is representative of non-linear models' behaviors on much more complex data. Although the experiments seem to suggest that it somewhat predicts behaviors of actual models, I think this discrepancy is misleading to say the least, if not overclaiming.
- some notations are not defined. For example, I cannnot find the definition of $c(\kappa)$. If I missed it, please let me know.

**Questions**
- In the appendix, shouldn't the population risk function $f$ use squared difference?
- The results suggest the unlabeled data can be drawn from different distribution with some constraints (i.e., $\tilde{\delta}$) but what do those constraints translate to for real world data?

**Reference**

[1] Principles of Mathematical Analysis. Rudin, Walter.

[2] Recovery Guarantees for One-hidden-layer Neural Networks. Zhong et al.


-------
**Update**

I thank the authors for the detailed response.

My concern regarding MVT has been fixed and that part of the paper is now technically sound.

Further, I believe that the authors' extension for two-layer models is technically sound insofar as you believe the tensor initialization method is indicative of what happens in the training dynamics of real models. On the other hand, personally, I am not convinced by the tensor initialization argument and I do not feel the analysis is adequate for explaining what happens in the self-training procedure of actual models.

In the light of these changes, I believe the paper is now sufficient for publication and I am increasing my score to 6.


**Summary Of The Paper:**

This paper presents the first theoretical analysis of self-training on one hidden-layer neural network with gaussian input. The paper shows that under certain conditions (e.g., the initialization is neither too far or too close to the optimal), iterative self-training on the aformentioned setting can provably recover the ground truth labeling matrix and offers improvement in sample complexity over supervised learning with only labeled data. Experiments validate the theoretical results.

**Summary Of The Review:**

The paper tackles a challenging and important problem. While the experimental results seem to match the theoretical predictions, there are potential significant flaws in the proofs of the main theoretical results. The paper would have been a pretty strong paper, but due to these potential errors, I unfortunatley cannot recommend acceptance.

---

### Official Review · Reviewer_7chM · 2021-11-04

**Correctness:** 3
**Technical Novelty And Significance:** 3
**Empirical Novelty And Significance:** 3
**Recommendation:** 6
**Confidence:** 3

**Main Review:**

Strengths:
1. The theoretical analysis is rigorous and contains novel technical contributions.
2. The empirical findings evaluate many interesting predictions of the theory.
3. The exposition of the proof sketch is clear and informative.
4. Comparisons with previous works are clear.

Weaknesses:
1. Self-training can hurt accuracy in some settings (confirmation bias). I wish the theory discusses when this happens.
2. I wish the theory offers guidance on how to improve self-training algorithms in practice.
3. Section 3.3 talks about '0 generalization error'. I'm confused by this term since there is always estimation error.

**Summary Of The Paper:**

This paper theoretically analyzes the iterative self-training algorithm with 1-hidden layer neural network. When the labeled and unlabeled data is generated from zero-mean Gaussian distributions with variances of different scales, it is shown that both the convergence rate and the generalization error decreases at a rate of $1/\sqrt{M}$ where $M$ is the number of unlabeled data. Experiments on synthetic data and augmented CIFAR-10 corroborate theoretical findings.

**Summary Of The Review:**

This paper is theoretical rigorous and has novel technical contributions.

---

### Official Review · Reviewer_LAdw · 2021-11-08

**Correctness:** 3
**Technical Novelty And Significance:** 2
**Empirical Novelty And Significance:** 1
**Recommendation:** 6
**Confidence:** 3

**Main Review:**

EDIT: After the discussion, I have raised my score to 6. Thanks to the authors for patiently addressing the various concerns and adding required appendices.


The paper studies an important problem of theoretically understanding how and why self-training works. The paper makes a genuine attempt to connect theory and practical insights and tries to distill some main takeaways from their analysis in  order to simplify understanding. However, I find that the paper in its current form does not push the boundaries on our understanding of self-training.

(1) My main criticism stems from the fast that several works have studied self-training in the linear setting (as rightly mentioned in the paper). This paper claims to study two-layer networks which are more challenging due to non-convexities. However, the assumptions of the theorem (which the proof heavily relies on) essentially study the problem in a "locally linear" region around the optimal solution. The paper claims novelty in introducing the quantity \lambda (weighting between the loss on the labels and pseudolabels while self-training) and an associated bound on the weighted objective to enable analyses. However, I really struggled with understanding the quantity \lambda and what it implies (more details below).
(2) The main self-training description (Table 1) does not mention \lambda. It is only described in-text in equation 1. How does one set \lambda in practice? I saw that the CIFAR-10 experiments used the same heuristic as in Carmon et al., but how does that relate to the analysis?
(3) The setting of the paper is slightly weird to me as well. (a) model family: they claim to study a two layer network but essentially freeze the top layer weights such that they are ones and there's only one matrix being learnt. I understand that the problem is still non-convex due to the non-linearities but this restriction of all ones seems to be pretty strong to me and its unclear how to generalize to multiple layers. One of the main complications in studying multilayer networks is the interaction between the parameters of each layer which the current version of the paper doesn't deal with. (b) Regression instead of classification: this paper really analyzes regression and not classification (square loss is analyzed both at train and test). I believe this significantly makes the analysis easier since one doesn't have to deal with training only on confident predictions and so on---one single \lambda factor can balance things out. In classification, we typically need some other kind of regularization
(4) Related works: The paper also seems to be missing some important related works  such as
[1] Y. Grandvalet and Y. Bengio. Entropy regularization. In Semi-Supervised Learning, 2005.
[2] P. Rigollet. Generalization error bounds in semi-supervised classification under the cluster
assumption. Journal of Machine Learning Research (JMLR), 8:1369–1392, 2007.
[3] A. Singh, R. Nowak, and J. Zhu. Unlabeled data: Now it helps, now it doesn’t. In Advances in
Neural Information Processing Systems (NeurIPS), 2008.
More generally, self-training has a very rich literature and the current draft doesn't situate this work well in terms of connections to previous work. The main claim of dealing with non-convex two layer networks is not very satisfying to me for reasons above. Are there any other additional insights or takeaways from this work?

(5) In terms of writing, I found the paper generally well-written except for routinely talking about terms that are introduced later. For example, the highlights of the theory section was very difficult to follow because all the main quantities were defined later.



**Summary Of The Paper:**

The paper theoretically analyzes self-training in neural networks with one hidden layer. Self-training is a popular semisupervised learning algorithm where a model uses large unlabeled data by training on pseudolabels generated by a teacher model trained on some small set of labels. The paper studies the generalization performance of self-training on a two-layer network under a Gaussian assumption on the inputs, and has some synthetic and CIFAR-10 experiments to show that the predicted theoretical rates (1/\sqrt{ size unlabeled data}) roughly match empirical performance.

**Summary Of The Review:**

Overall, the paper studies an important problem (self-training on multilayer networks) and tries to connect some theoretical takeaways to practice. However, both in terms of theory and experiments, I found that the paper does not broaden our conceptual understanding of the method, nor does it introduce interesting theoretical tools for analysis. However, I believe there is a lot of scope to build on this work (relaxing assumptions, choosing the right model family and loss functions) to get a more convincing publication.

---

### Decision · Program_Chairs · 2022-01-20

**Decision:**

Accept (Poster)

**Comment:**

The paper studies self-training for a one hidden-layer architecture, showing that with proper initialization self-training will improve over standard supervision. The reviewers appreciated the analysis and thought the results make sense. However, they did comment that the paper does not provide sufficient insight about the effectiveness of self-training and this should be discussed in the final version. There were additionally comments about the architecture choice (e.g., fixed output weights), and the authors should also address this.